# Latent g-Computation for Potential Outcomes Distributional Estimation under Time-Varying Treatments

## Abstract

Estimating individualized potential outcomes (POs) under time-varying treatments is central to fields like medicine, marketing, and public policy, where decisions must account for uncertainty rather than just point forecasts. We introduce a latent g-computation estimator for discrete-time, individualized PO distributions. Under standard longitudinal identification assumptions and a latent factorization/context-sufficiency condition —essentially the usual expressivity assumption for conditional VAEs—, we show that a rollout entirely in latent space targets the same interventional distribution as the classical g-formula, while never autoregressing covariates in data space. We further derive a total-variation error–propagation bound proving that, for a given one-step approximation error, latent rollouts exhibit more favorable long-horizon behavior than data-space autoregressive g-computation. We instantiate this estimator as G-Latent, which replaces G-Net's residual pools (Li et al., 2021) with a conditional VAE that learns history- and treatment-conditioned outcome distributions at each time. To enhance expressivity, we adapt an infinite-mixture asymmetric Laplace (ALD) parameterization (An & Jeon, 2023) to the time-series setting, and we decouple sequence encoding (a transformer over the observed history) from a lightweight GRU latent rollout with selective decoding, enabling fast Monte Carlo sampling over multiple horizons. We evaluate G-Latent in semi-synthetic and real-world datasets, finding that it yields better calibrated and more accurate predictive PO distributions than strong baselines, while reducing inference-time cost.

## 1 Introduction

Estimating individualized potential outcomes under time-varying treatments is central to data-rich domains such as precision medicine, marketing, education, and public policy, where longitudinal records capture detailed sequences of covariates, interventions, and responses. While recent neural approaches address time-dependent confounding and long-range dependencies, most return only point estimates—typically conditional means (Melnychuk et al., 2022; Bouchattaoui et al., 2023)- or consider only *epistemic* (model) uncertainty. Modeling epistemic uncertainty is valuable for flagging low-confidence regions or detecting out-of-distribution inputs; however, it leaves *aleatoric* (data) uncertainty unmodeled, so identical expected outcomes may conceal very different variances, skewness, and tail risks. For risk-sensitive decisions—where clinicians care about adverse-event probabilities, marketers about downside exposure, and policymakers about extreme impacts—ignoring aleatoric uncertainty limits actionable guidance. We therefore advocate moving beyond mean effects and purely epistemic views to full, coherent distributional estimates of individualized potential outcomes across time and variables, enabling transparent, risk-aware decision support.

We introduce G-Latent, a model for distributional individualized POs under time-varying treatments that performs g-computation in latent space. The key idea is a latent rollout: during counterfactual rollouts, we update the temporal representation using VAE latent variables rather than observed covariates, and decode only when needed. This avoids data-space autoregression—reducing accumulation error and making g-computation practical with high-dimensional covariates—while enabling efficient sampling for many treatment sequences and Monte Carlo (MC) draws. G-Latent learns per-step conditional distributions non-parametrically via a conditional VAE on past representations.

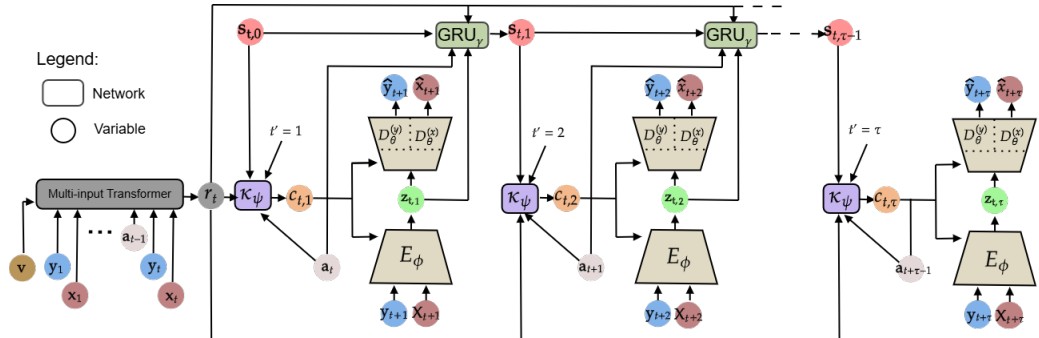

Figure 1: Training-time data flow in G-Latent for a given step $t$. A multi-input transformer encodes history $\mathbf{r}_t$; at each step $t'$ within the projection horizon, a context $\mathbf{c}_{t,t'}$ feeds a shared conditional VAE. The GRU updates the state using latents $\mathbf{z}_{t,t'}$ (not decoded observations). The decoder has outcome (ALD) and covariate (Gaussian) heads.

Following (An & Jeon, 2023) and extending to time series, we parameterize the decoder as an infinite mixture of asymmetric Laplace distributions (ALDs) (Brando et al., 2019), increasing expressivity. In contrast, g-computation baselines such as Li et al. (2021) approximate distributions via mean predictions plus errors from a global residual pool, which can distort individualized distributions, especially under heteroscedasticity. For efficiency, we decouple long-history encoding and short-horizon rollout: a transformer encodes the long prefix once; a lightweight Gated Recurrent Unit (GRU) updates representations across the projection horizon, avoiding repeated transformer passes during sampling. Identifiability follows the g-computation formula under standard assumptions of sequential ignorability, positivity, and consistency.

We summarize our contributions as follows: **1)** We define a novel *latent g-computation estimator* for individualized potential outcome distributions in discrete time under time-varying treatments. Under standard longitudinal identification assumptions and a latent factorization / context-sufficiency condition —essentially the usual expressivity assumption for conditional VAEs— we prove that a rollout entirely in latent space targets the same interventional distribution as the classical g-formula while never autoregressing covariates in data space (Thm. 5.1, Cor. 5.2). To our knowledge, furthermore, ours is the first discrete-time method for individualized distributional POs without global residual pools. **2)** We analyze error propagation for latent vs. data-space implementations of g-computation and derive a total-variation bound showing that, for any fixed one-step approximation error, latent rollouts exhibit more favorable long-horizon behavior than standard autoregressive g-computation (Prop. 5.3), theoretically explaining the improved stability we observe at longer horizons. **3)** We instantiate this estimator as *G-Latent*, a conditional VAE with a transformer history network, a lightweight GRU latent rollout, and an ALD-mixture outcome head adapted from An & Jeon (2023), which together enable flexible individualized outcome distributions and fast Monte Carlo sampling via selective decoding. **4)** We provide an extensive empirical study on semi-synthetic and real-world ICU data, including calibration metrics, runtime comparisons, and an analysis (and correction) of the widely used semi-synthetic MIMIC-III (Melnychuk et al., 2022) benchmark that previously violated positivity. Across datasets, G-Latent improves the quality and calibration of predictive PO distributions relative to strong baselines while reducing inference-time cost.

## 2 RELATED WORK

**Potential outcomes estimation in static settings.** In the static setting, there are several methods for individualized PO estimation. Representative modern examples include Yoon et al. (2018); Vansteelandt & Morzywolek (2023); Shalit et al. (2017); Künzel et al. (2019). Although most static PO methods provide only point estimates, some works estimate distributional POs. For instance, papers like Melnychuk et al. (2023); Kennedy et al. (2023) target population-level distributional POs, whereas Ma et al. (2024) learn individualized distributional POs using diffusion models (Yang et al., 2023).

**Individualized potential outcomes estimation over time.** Traditionally, causal inference has addressed time-varying confounders with Marginal Structural Models (MSMs) (Robins et al., 2000), which rely on inverse probability of treatment weighting (IPTW) (Chesnaye et al., 2022), or G-computation (Taubman et al., 2009). Lim (2018) improve MSMs by employing RNNs in the modeling of outcomes and propensities. Counterfactual Recurrent Network (CRN) (Bica et al.) incorporates adversarial domain training to establish a treatment-invariant representation space using a gradient reversal layer (Ganin & Lempitsky, 2015). G-Net (Li et al., 2021) combines RNNs with G-computation to adjust for confounders and estimate dynamic potential outcomes. Causal Transformer (CT) (Melnychuk et al., 2022) follows the treatment-invariant representation idea from CRN and incorporates transformers to process time series and a Counterfactual Domain Confusion (CDC) loss (Tzeng et al., 2015). Other works that also follow this idea are Wang et al. (2024), which adopts a novel Temporal Integration Predicting strategy and focuses on continuous treatments, and El Bouchattaoui et al. (2024), which introduces an RNN backbone trained with Contrastive Predictive Coding and an InfoMax objective. Wang et al. (2025) use a state-space architecture (Mamba) (Gu & Dao, 2024) that employs covariate-based decorrelation toward selective parameters to reduce confounding bias. Huang et al. (2024) provide an empirical evaluation of balancing strategies. On the other hand, Xiong et al. (2024) use a similar approach to G-Net but processing data with transformers instead of RNNs, and Deng et al. (2024) add model uncertainty to the same approach. Hess et al. (2024) propose a pseudo-outcome regression based on g-formula to obtain individualized POs. Finally, Frauen et al. propose a series of model-agnostic meta-learners for estimating heterogeneous treatment effects over time.

In parallel to the previous works, another line of research has appeared in recent years that models the effects of treatments in continuous-time with neural Ordinary Differential Equations (ODEs). De Brouwer et al. (2022) couples neural ODEs with epistemic uncertainty quantification for continuous-time predictions. Seedat et al. (2022) learn Controlled Differential Equation (CDE) dynamics robust to irregular sampling. Hess et al. present Bayesian Neural CDE (BNCDE), which provides posterior predictive distributions over POs. Finally, Hess & Feuerriegel employ a stabilized continuous-time IPTW formulation to address time-varying confounding.

All the previous works, like ours, assume sequential ignorability (Robins & Hernan, 2008). There is another line of research that tackles violations of this assumption. Among them, papers like Peng et al.; Bouchattaoui et al. (2023) are worth mentioning as, like this work, they use the latent representations of VAEs. However, they do it to infer hidden confounders in settings where they exist. In contrast, our work uses latent representations to adjust for observed confounders following G-computation. Finally, Wang et al. present another VAE-based approach that aims at selecting best treatment sequences by modeling the conditional likelihood of achieving target outcomes.

**Uncertainty Quantification in potential outcomes estimation over time.** Some of the aforementioned time-varying methods include some form of uncertainty quantification. Within the continuous-time works, De Brouwer et al. (2022) handles epistemic uncertainty through variational Bayesian inference . On the other hand, Hess et al. handles both epistemic uncertainty, with Bayesian posterior distributions, and aleatoric uncertainty, with a Gaussian outcome head. However, it does not handle time-varying confounding. Very recently, a new paper appeared (Mu et al., 2025) that employs diffusion models to model distributional potential outcomes with expert models.

As for discrete time models for individualized POs, uncertainty quantification has been mostly ignored. Papers like Melnychuk et al. (2022); Bica et al. handle epistemic uncertainty only through Monte Carlo (MC) dropout. As for aleatoric uncertainty, G-Net (Li et al., 2021) and its transformer extension (Xiong et al., 2024) are, to the best of our knowledge, the only models that handle it. Like our model, G-Net builds on g-computation to generate sequential MC samples. However, its capacity to properly model PO distributions is limited because it only handles homoscedastic data. Furthermore, it tends to underperform in comparison with other methods due to an error compounding problem. Deng et al. (2024) enriches (Transformer) G-Net by adding epistemic uncertainty, but it suffers from the same problems as (Transformer) G-Net. Finally, Wu et al. (2024) combine VAEs and diffusion models with IPTW to obtain distributional POs, and Shirakawa et al. (2024) couple a temporal-difference heterogeneous Transformer with longitudinal Targeted Minimum Loss-based, allowing to estimate POs confidence intervals, but these works handle only population-level POs, so they do not fit our setting.

## 3    PROBLEM FORMULATION

For the variables of our setting, uppercase bold letters (e.g., $\mathbf{X}, \mathbf{A}, \mathbf{Y}$) denote random vectors; lowercase bold (e.g., $\mathbf{x}, \mathbf{a}, \mathbf{y}$) their realizations, and plain letters denote scalars (e.g., $x, y$). For latent vectors and learnable representation vectors, we use bold lowercase.

**Problem Setting.** We adopt the standard setting for estimating counterfactual outcomes over time (Lim, 2018; Bica et al.; Melnychuk et al., 2022; El Bouchattaoui et al., 2024). Let $i$ index patients with trajectories observed at $t = 1, \dots, T^{(i)}$. At each $t$ we observe time-varying covariates $\mathbf{X}_t^{(i)} \in \mathbb{R}^{d_x}$, treatments $\mathbf{A}_t^{(i)}$, and outcomes $\mathbf{Y}_t^{(i)} \in \mathbb{R}^{d_y}$, as well as static covariates $\mathbf{V}^{(i)}$ (e.g., sex, age, risk factors). Unless needed, we omit the patient index $(i)$. We assume i.i.d. observational data $\mathcal{D} = \{(\mathbf{x}_{1:T^{(i)}}^{(i)}, \mathbf{a}_{1:T^{(i)}}^{(i)}, \mathbf{y}_{1:T^{(i)}}^{(i)}, \mathbf{v}^{(i)})\}_{i=1}^N$, with $\mathbf{x}_{1:T^{(i)}}^{(i)} = (\mathbf{x}_1^{(i)}, \dots, \mathbf{x}_{T^{(i)}}^{(i)})$ and analogously for $\mathbf{a}, \mathbf{y}$.

**History and calendar.** We use *start-of-interval* indexing: the treatment $\mathbf{A}_t$ precedes the next measurement $(\mathbf{Y}_{t+1}, \mathbf{X}_{t+1})$. Let the history available *before* choosing $\mathbf{A}_t$ be $\bar{\mathbf{H}}_t = \{\bar{\mathbf{X}}_t, \bar{\mathbf{A}}_{t-1}, \bar{\mathbf{Y}}_t, \mathbf{V}\}$ with $\bar{\mathbf{X}}_t = (\mathbf{X}_1, \dots, \mathbf{X}_t)$, $\bar{\mathbf{Y}}_t = (\mathbf{Y}_1, \dots, \mathbf{Y}_t)$, and $\bar{\mathbf{A}}_{t-1} = (\mathbf{A}_1, \dots, \mathbf{A}_{t-1})$. For compactness we sometimes group outcomes and covariates as $\mathbf{L}_t = (\mathbf{Y}_t, \mathbf{X}_t) \in \mathbb{R}^{d_L}$.

**Targets.** Let $\tau \geq 1$ denote the projection horizon and $\bar{\mathbf{a}}_{t:t+\tau-1} = (\mathbf{a}_t, \dots, \mathbf{a}_{t+\tau-1})$ a given (non-random) treatment intervention. Most previous works in this setting aim to estimate the conditional mean $\mathbb{E}\left[\mathbf{Y}_{t+\tau}[\bar{\mathbf{a}}_{t:t+\tau-1}] \mid \bar{\mathbf{H}}_t\right]$. In contrast, we target the *full conditional distribution*, both at a fixed horizon and jointly across horizons:

$$p^{\bar{\mathbf{a}}}(\mathbf{y}_{t+\tau} \mid \bar{\mathbf{h}}_t), \qquad p^{\bar{\mathbf{a}}}(\mathbf{y}_{t+1:t+\tau} \mid \bar{\mathbf{h}}_t). \tag{1}$$

**Assumptions.** We build upon the potential outcomes framework (Rubin, 2005) and its extension to time-varying treatments (Robins et al., 2000). We assume (1) consistency, (2) sequential ignorability/exchangeability, and (3) sequential overlap/positivity (see App. A).

**Goal.** We design a novel implementation of g-computation that learns flexible per-step conditionals and generates *coherent fast Monte Carlo samples* from $p^{\bar{\mathbf{a}}}(\cdot \mid \bar{\mathbf{h}}_t)$, enabling distributional individualized potential outcomes without data-space autoregression.

**The g-Formula.** Under the assumptions previously specified, for any non-random regime $\bar{\mathbf{a}}_{t:t+\tau-1}$,

$$p^{\bar{\mathbf{a}}_{t:t+\tau-1}}(\mathbf{y}_{t+1:t+\tau} \mid \bar{\mathbf{h}}_t) = \int_{\mathbf{x}_{t+1:t+\tau}} \prod_{s=t}^{t+\tau-1} p(\mathbf{y}_{s+1}, \mathbf{x}_{s+1} \mid \bar{\mathbf{h}}_s, \mathbf{a}_s) \, d\mathbf{x}_{t+1:t+\tau}, \tag{2}$$

where $\bar{\mathbf{h}}_{s+1} := (\bar{\mathbf{h}}_s, \mathbf{a}_s, \mathbf{y}_{s+1}, \mathbf{x}_{s+1})$.

## 4    LATENT G-COMPUTATION

In this section, we first define a *latent g-computation estimator* that implements discrete-time g-computation entirely in latent space (Section 4.1). Under a latent factorization / context-sufficiency condition, we show in Section 4.2 that this estimator targets the same interventional distribution as the classical g-formula, while never autoregressing covariates in data space. We then analyze its error propagation and finally instantiate it as a neural model, G-Latent, based on a transformer history network, a conditional VAE, and GRU updates in latent space.

### 4.1    THE LATENT G-COMPUTATION ESTIMATOR

Consider the g-formula (Eq. 2), which expresses the interventional law under a non-random treatment plan $\bar{\mathbf{a}}_{t:t+\tau-1}$ as an iterated integral over one-step conditionals

$$p^{\star}(\mathbf{y}_{s+1}, \mathbf{x}_{s+1} \mid \bar{\mathbf{h}}_s, \mathbf{a}_s), \qquad s = t, \dots, t+\tau-1. \tag{3}$$

Standard implementations of g-computation approximate these kernels directly in data space and then perform autoregressive rollouts, repeatedly sampling covariates and feeding them back into the model. We instead ask whether g-computation can be implemented entirely in latent space, so that we never autoregress observed covariates while still targeting the same interventional distribution.

---

**Algorithm 1** Latent g-computation estimator (Monte Carlo rollout)

---

1: **Input:** history $\bar{\mathbf{h}}_t$, treatment plan $\bar{\mathbf{a}}_{t:t+\tau-1}$, horizon $\tau$, samples $M$, scope$\in \{\texttt{all}, \texttt{last}\}$
2: $\mathbf{r}_t \leftarrow f_\omega(\bar{\mathbf{h}}_t)$
3: **for** $m = 1$ **to** $M$ **do** ▷ Monte Carlo paths
4:     $\mathbf{s}_{t,0} \leftarrow \mathbf{0}$
5:     **for** $t' = 1$ **to** $\tau$ **do**
6:         $\mathbf{c}_{t,t'} \leftarrow \kappa_\psi(\mathbf{r}_t, \mathbf{s}_{t,t'-1}, \mathbf{a}_{t+t'-1}, t')$
7:         $\mathbf{z}_{t,t'}^{(m)} \sim p_0(\cdot)$ ▷ e.g., $\mathcal{N}(\mathbf{0}, \mathbf{I})$
8:         **if** scope $=$ all **or** $t' = \tau$ **then**
9:             **decode** $\mathbf{y}_{t+t'}^{(m)}, (\mathbf{x}_{t+t'}^{(m)}) \sim p_\theta(\cdot \mid \mathbf{z}_{t,t'}^{(m)}, \mathbf{c}_{t,t'}, \mathbf{a}_{t+t'-1})$
10:         $\mathbf{s}_{t,t'} \leftarrow \Gamma_\gamma(\mathbf{z}_{t,t'}^{(m)}, \mathbf{r}_t, \mathbf{a}_{t+t'-1}, t', \mathbf{s}_{t,t'-1})$
11: **Return:** $\{\mathbf{y}_{t+1:t+\tau}^{(m)}\}_{m=1}^M$ if scope=all, else $\{\mathbf{y}_{t+\tau}^{(m)}\}_{m=1}^M$

---

Fix a time $t$ and a prediction horizon $\tau \geq 1$. Let $\bar{\mathbf{h}}_t$ denote the observed history up to time $t$ and $\bar{\mathbf{a}}_{t:t+\tau-1}$ a treatment plan applied from $t$ to $t+\tau-1$. Our latent estimator uses four components: (i) a *history network* $f_\omega$ that maps the observed history to an embedding $\mathbf{r}_t = f_\omega(\bar{\mathbf{h}}_t)$; (ii) a recurrent *latent state* $\mathbf{s}_{t,t'}$ summarizing the latent trajectory from $t$ to $t+t'$, initialized as $\mathbf{s}_{t,0} = \mathbf{0}$ and updated as

$$\mathbf{s}_{t,t'} = \Gamma_\gamma(\mathbf{z}_{t,t'}, \mathbf{r}_t, \mathbf{a}_{t+t'-1}, t', \mathbf{s}_{t,t'-1}) \qquad t' = 1, \ldots, \tau; \tag{4}$$

(iii) a *context map* $\mathbf{c}_{t,t'} = \kappa_\psi(\mathbf{r}_t, \mathbf{s}_{t,t'-1}, \mathbf{a}_{t+t'-1}, t')$, which collects all information needed by the one-step decoder at step $t'$; and (iv) a conditional decoder $p_\theta$ with fixed latent prior $p_0$ defining one-step kernels

$$p_\theta(\mathbf{l}_{t+t'} \mid \mathbf{z}_{t,t'}, \mathbf{c}_{t,t'}, \mathbf{a}_{t+t'-1}), \qquad \mathbf{l}_{t+t'} = (\mathbf{y}_{t+t'}, \mathbf{x}_{t+t'}), \quad \mathbf{z}_{t,t'} \sim p_0(\cdot). \tag{5}$$

Given these components, we implement g-computation by *ancestral sampling of full latent paths*. For each Monte Carlo replicate, we sample a trajectory of latents $\mathbf{z}_{t,1}, \ldots, \mathbf{z}_{t,\tau}$ under the treatment plan, update the latent state forward in time, and decode outcomes (and optionally covariates) at selected horizons. Crucially, decoded observations are never fed back into the state; all temporal dependence flows through $(\mathbf{r}_t, \mathbf{s}_{t,t'})$. In our concrete instantiation (Section 4.3), $p_\theta$ and $p_0$ arise from a conditional VAE over $(\mathbf{Y}_t, \mathbf{X}_t)$.

With our estimator, one can decode at any subset $S \subseteq \{1, \ldots, \tau\}$ of relative steps. The latent rollout and state updates are identical in all cases; only decoding is selective. We parameterize this choice via an argument scope that specifies at which relative steps we decode outcomes. In this work, we consider two options: scope=all corresponds to decoding at all $t' = 1, \ldots, \tau$, while scope=last corresponds to decoding only at $t' = \tau$. This selective decoding is useful computationally: when we are only interested in $\mathbf{y}_{t+\tau}$, choosing scope=last avoids decoding at the intermediate $\tau - 1$ steps, reducing the decoder cost from $O(\tau M)$ to $O(M)$ for $M$ Monte Carlo paths. More generally, decoding at an arbitrary subset $S$ scales the decoder cost linearly in $|S|$ rather than in $\tau$.

Algorithm 1 defines our latent g-computation estimator: given a history $\bar{\mathbf{h}}_t$ and a treatment plan $\bar{\mathbf{a}}_{t:t+\tau-1}$, it produces Monte Carlo samples from an interventional distribution induced by the one-step conditionals $p_\theta(\mathbf{l}_{t+t'} \mid \mathbf{z}_{t,t'}, \mathbf{c}_{t,t'}, \mathbf{a}_{t+t'-1})$. In Section 4.2, we state conditions under which this estimator is equivalent to the classical g-formula and analyze its error propagation. In Section 4.3, we describe how we instantiate $(f_\omega, \kappa_\psi, p_\theta, \Gamma_\gamma)$ as the neural model G-Latent.

### 4.2 THEORETICAL INSIGHTS

We now provide theoretical guarantees that the latent g-computation estimator implements the same interventional law as the traditional data-space g-formula, and compare its error propagation to a data-space autoregressive rollout. See full proofs and additional discussion in App. E.

**Assumption 4.1.** (Latent factorization and context sufficiency). Fix $t$ and $\tau \geq 1$. Let $\mathbf{r}_t = f_\omega(\bar{\mathbf{h}}_t)$, let the latent state $\mathbf{s}_{t,t'}$ and context $\mathbf{c}_{t,t'}$ be defined as in Section 4.1, and consider the one-step conditional over $\mathbf{l}_{t+t'} = (\mathbf{y}_{t+t'}, \mathbf{x}_{t+t'})$. We assume that the true one-step conditional admits a latent mixture factorization with a fixed prior $p_0$:

$$p^\star(\mathbf{l}_{t+t'} \mid \bar{\mathbf{h}}_{t+t'-1}, \mathbf{a}_{t+t'-1}) = \int p_\theta(\mathbf{l}_{t+t'} \mid \mathbf{z}_{t,t'}, \mathbf{c}_{t,t'}, \mathbf{a}_{t+t'-1}) p_0(\mathbf{z}_{t,t'}) d\mathbf{z}_{t,t'}. \tag{6}$$

(See App. E.1 for the formal statement and further discussion.)

Intuitively, this says that once we condition on a sufficiently informative context $\mathbf{c}_{t,t'}(\bar{\mathbf{h}}_{t+t'-1}, \mathbf{a}_{t+t'-1})$, the decoder family $p_\theta(\cdot \mid \mathbf{z}, \mathbf{c}, \mathbf{a})$ is rich enough to represent the true one-step conditional as a mixture over a fixed prior $p_0$, as in a standard conditional VAE. This is the standard conditional VAE modeling assumption and not an additional causal assumption.

**Theorem 4.2** (Equivalence of latent and data-space $g$-computation). *Under the identification assumptions (App. A) and the latent factorization in Eq. 6, for any treatment plan $\bar{\mathbf{a}}_{t:t+\tau-1}$ and history $\bar{\mathbf{h}}_t$, Algorithm 1 produces i.i.d. MC samples from the interventional laws identified by the g-formula (Eq. 2):*

$$(\textit{full path}) \qquad p^{\bar{\mathbf{a}}}\big(\mathbf{y}_{t+1:t+\tau} \mid \bar{\mathbf{h}}_t\big) = \int \prod_{t'=1}^{\tau} p_\theta\Big(\mathbf{y}_{t+t'} \mid \mathbf{z}_{t,t'}, \mathbf{c}_{t,t'}(\mathbf{z}_{t,1:t'-1}), \mathbf{a}_{t+t'-1}\Big) \prod_{t'=1}^{\tau} p_0(\mathbf{z}_{t,t'}) \, d\mathbf{z}_{t,t'},$$

$$(\textit{fixed horizon}) \quad p^{\bar{\mathbf{a}}}\big(\mathbf{y}_{t+\tau} \mid \bar{\mathbf{h}}_t\big) = \int p_\theta\Big(\mathbf{y}_{t+\tau} \mid \mathbf{z}_{t,\tau}, \mathbf{c}_{t,\tau}(\mathbf{z}_{t,1:\tau-1}), \mathbf{a}_{t+\tau-1}\Big) \prod_{t'=1}^{\tau} p_0(\mathbf{z}_{t,t'}) \, d\mathbf{z}_{t,t'}.$$

$$(7)$$

Proof. *App. E.5.*

**Corollary 4.3** (Selective decoding is coherent). *Decoding only at $t+\tau$ (scope=last) returns i.i.d. samples from $p^{\bar{\mathbf{a}}}(\mathbf{y}_{t+\tau} \mid \bar{\mathbf{h}}_t)$; decoding at any subset $S \subseteq \{1, \ldots, \tau\}$ returns the corresponding marginals $\{p^{\bar{\mathbf{a}}}(\mathbf{y}_{t+t'} \mid \bar{\mathbf{h}}_t)\}_{t' \in S}$.* Proof. *App. E.5.*

**Error propagation: latent vs. data-space g-computation rollouts.** *Takeaway:* the latent rollout (Alg. 1) does not amplify local one-step errors, whereas data-space autoregressive (AR) rollouts can, because they repeatedly decode and re-encode observations.

In latent g-computation, the learned one-step kernel is the decoder-induced latent mixture at context $\mathbf{c}_{t,t'}$, $K_s^{\mathrm{e}}(\cdot \mid \bar{\mathbf{h}}_s, \mathbf{a}_s) = \int p_\theta(\cdot \mid \mathbf{z}, \mathbf{c}_{t,t'}, \mathbf{a}_s) \, p_0(\mathbf{z}) \, d\mathbf{z}$ with $s = t+t'-1$; $\mathbf{c}_{t,t'}$ is defined in Sec. 4.1 and the state is updated *through latents only*. As a comparator, we use a data-space AR rollout that decodes each step and re-feeds (or re-encodes), inducing a single-step Lipschitz AR tail operator with factors $\{1 + \lambda_j\}$. Let $K_s^\star(\cdot \mid \bar{\mathbf{h}}_s, \mathbf{a}_s)$ denote the true one-step conditional and define $\varepsilon_s := \sup_{\bar{\mathbf{h}}_s, \mathbf{a}_s} \mathrm{TV}\big(K_s^\star(\cdot \mid \bar{\mathbf{h}}_s, \mathbf{a}_s), K_s^{\mathrm{e}}(\cdot \mid \bar{\mathbf{h}}_s, \mathbf{a}_s)\big)$, where $\mathrm{TV}(\mu, \nu)$ denotes the total variation distance $\mathrm{TV}(\mu, \nu) := \sup_{A \in \mathcal{A}} |\mu(A) - \nu(A)|$.

**Proposition 4.4** (Propagation-error bound and dominance). *Assume that the single-step AR tail operators are Lipschitz in total variation with factors $(1 + \lambda_j)$ (see Assumption E.7). Let $P^\star$ be the interventional law of $Y_{t+\tau}$ and $P^{\mathrm{lat}}$, $P^{\mathrm{AR}}$ the laws induced by latent and AR rollouts using $\{K_s^{\mathrm{e}}\}$. Then, taking total variation over the marginal of $Y_{t+\tau}$,*

$$\mathrm{TV}(P^\star, P^{\mathrm{lat}}) \leq \sum_{s=t}^{t+\tau-1} \varepsilon_s, \qquad \mathrm{TV}(P^\star, P^{\mathrm{AR}}) \leq \sum_{s=t}^{t+\tau-1} \varepsilon_s \prod_{j=s+1}^{t+\tau-1} (1 + \lambda_j). \qquad (8)$$

Proof. *App. E.10.*

Our model inevitably makes small one-step errors in the conditional distributions. The key difference is how these local errors are propagated. In the latent g-computation rollout, once the factual history is encoded, all future evolution happens in latent space and decoded predictions are never fed back; mathematically, the subsequent latent transitions are Markov and non-expansive in total variation, so each local error contributes at most additively to the final discrepancy. In a data-space autoregressive rollout, every decoded prediction is fed back through a powerful encoder to form the next context, and these encode–decode maps can enlarge discrepancies, so a small local error at a given time step can be amplified at later steps. Proposition 4.4 formalizes exactly this: both approaches share the same local approximation errors, but only the data-space rollout has this additional error-amplification channel, which explains its worse long-horizon behavior.

### 4.3 NEURAL INSTANTIATION: THE G-LATENT MODEL

**Architecture.** We instantiate the abstract components $(f_\omega, \Gamma_\gamma, \kappa_\psi, p_\theta, p_0)$ with a history network, a latent GRU, and a conditional VAE. The *history network $f_\omega$* is a multi-input transformer that maps

the observed history $\bar{\mathbf{h}}_t$ to an embedding $\mathbf{r}_t = f_\omega(\bar{\mathbf{h}}_t)$, following Melnychuk et al. (2022) (three streams for $\bar{\mathbf{x}}_t, \bar{\mathbf{a}}_{t-1}, \bar{\mathbf{y}}_t$ with cross-attention; details in App. B). The latent state update $\Gamma_\gamma$ (Eq. 4) is implemented as a GRU, and the context map as $\mathbf{c}_{t,t'} = \kappa_\psi(\mathbf{r}_t, \mathbf{s}_{t,t'-1}, \mathbf{a}_{t+t'-1}, t')$, so that future contexts depend only on compact latent summaries rather than decoded observations. For the one-step conditionals over $\mathbf{l}_{t+t'} = (\mathbf{y}_{t+t'}, \mathbf{x}_{t+t'})$ we use a single conditional VAE, shared across $t'$, with *VAE encoder* and decoder

$$q_\phi(\mathbf{z}_{t,t'} \mid \mathbf{l}_{t+t'}, \mathbf{c}_{t,t'}), \qquad p_\theta(\mathbf{l}_{t+t'} \mid \mathbf{z}_{t,t'}, \mathbf{c}_{t,t'}, \mathbf{a}_{t+t'-1}),$$

and prior $p_0(\mathbf{z}_{t,t'}) = \mathcal{N}(\mathbf{0}, \mathbf{I})$. Outcomes $\mathbf{y}_{t+t'}$ are modeled with the ALD-mixture parameterization of An & Jeon (2023) (DistVAE), extended here to time series with sequential treatments, while covariates $\mathbf{x}_{t+t'}$ use Gaussian heads; see below.

**Training objective and implementation.** We share one conditional VAE across steps $t' \in \{1, \ldots, \tau\}$ and optimize a joint per-step objective. Given the context $\mathbf{c}_{t,t'}$, the VAE encoder outputs $\mathbf{z}_{t,t'} \sim q_\phi(\mathbf{z}_{t,t'} \mid \mathbf{l}_{t+t'}, \mathbf{c}_{t,t'})$, and we update the latent state via Eq. 4. The decoder $p_\theta(\mathbf{l}_{t+t'} \mid \mathbf{z}_{t,t'}, \mathbf{c}_{t,t'}, \mathbf{a}_{t+t'-1})$ is parameterized by a *shared trunk* $T_\theta$ followed by two heads: an outcome head $D_\theta^{(y)}$ and a covariate head $D_\theta^{(x)}$. Let $\mathbf{w}_{t,t'} = T_\theta(\mathbf{z}_{t,t'}, \mathbf{c}_{t,t'}, \mathbf{a}_{t+t'-1})$ and

$$\widehat{\mathbf{q}}_{\boldsymbol{\alpha},t,t'} = D_\theta^{(y)}(\mathbf{w}_{t,t'}, \boldsymbol{\alpha}), \qquad (\hat{\boldsymbol{\mu}}_{t,t'}, \hat{\boldsymbol{\sigma}}_{t,t'}^2) = D_\theta^{(x)}(\mathbf{w}_{t,t'}),$$

where $\boldsymbol{\alpha} \in (0,1)^{d_y}$ collects per-outcome quantile levels. We implement $D_\theta^{(y)}$ as $d_y$ scalar branches and draw $K$ vectors $\{\boldsymbol{\alpha}^{(k)}\}_{k=1}^K$ with i.i.d. entries $\alpha_j^{(k)} \sim \mathrm{Unif}(0,1)$. The per-step reconstruction loss is

$$\mathcal{L}_{\mathrm{rec}}(t,t') = \sum_{j=1}^{d_y} \frac{1}{K} \sum_{k=1}^K \rho_{\alpha_j^{(k)}}\big(y_{t+t',j} - \widehat{\mathbf{q}}_{\alpha_j^{(k)},t,t',j}\big) + \frac{1}{2}\left\|\frac{\mathbf{x}_{t+t'}-\hat{\boldsymbol{\mu}}_{t,t'}}{\hat{\boldsymbol{\sigma}}_{t,t'}}\right\|_2^2 + \frac{1}{2}\mathbf{1}^\top \log \hat{\boldsymbol{\sigma}}_{t+t'}^2, \quad (9)$$

where $\rho_\alpha(u) = (\alpha - \mathbf{1}\{u < 0\})u$ is the pinball loss and $(\hat{\boldsymbol{\mu}}_{t,t'}, \hat{\boldsymbol{\sigma}}_{t,t'}^2)$ are the Gaussian parameters for $\mathbf{x}_{t+t'}$. The KL term is $\mathcal{L}_{\mathrm{KL}}(t,t') = \mathrm{KL}\big(q_\phi(\mathbf{z}_{t,t'} \mid \cdot) \| \mathcal{N}(\mathbf{0}, \mathbf{I})\big)$. This corresponds to a conditional VAE with an ALD-mixture outcome decoder (An & Jeon, 2023); integrating over $\alpha$ recovers a CRPS-type reconstruction term, which encourages well-calibrated, flexible predictive distributions beyond Gaussian heads (see App. C for details). In our setting, using the ALD mixture for $\mathbf{y}$ improves distributional performance but increases decoder complexity, so we use it only for outcomes and keep a simpler Gaussian head for covariates $\mathbf{x}$, where the additional expressivity does not offset the extra compute. Predictive uncertainty arises both from the sampled latent path (capturing temporal and cross-outcome dependence) and from the outcome head, which plays the role of the likelihood noise model, analogous to decoder noise in a Gaussian VAE.

The history network $f_\omega$ is high-capacity, and the VAE objective alone can be minimized even if $\mathbf{r}_t$ carries little predictive signal (the decoder may partly ignore it). To avoid such degenerate configurations, we add an auxiliary one-step prediction head $\hat{\mathbf{y}}_{t+1} = U_\eta(\mathbf{r}_t, \mathbf{a}_t)$ with MSE loss $\mathcal{L}_{\mathrm{aux}}$ (Eq. 10), used purely as a regularizer to make $\mathbf{r}_t$ predictive of $\mathbf{y}_{t+1}$. The total loss over a mini-batch $\mathcal{B}$ is

$$\mathcal{L} = \frac{1}{|\mathcal{B}|} \sum_{i \in \mathcal{B}} \sum_{t=1}^{T^{(i)}-1} \Big[ \sum_{t'=1}^\tau m_{t,t'}^{(i)} \big( \mathcal{L}_{\mathrm{rec}}^{(i)}(t,t') + \beta\, \mathcal{L}_{\mathrm{KL}}^{(i)}(t,t') \big) + \lambda_{\mathrm{aux}}\, m_{t,1}^{(i)}\, \mathcal{L}_{\mathrm{aux}}^{(i)}(t) \Big], \qquad (10)$$

with masks $m_{t,t'}^{(i)} = \mathbf{1}\{t + t' \le T^{(i)}\}$. In practice, we found it helpful to *warm start* the history network by first optimizing only $\mathcal{L}_{\mathrm{aux}}$ for a small number of epochs, and then training the full objective in Eq. 10. This implementation choice affects how the parameters are learned but does not change the latent g-computation estimator of Section 4.1. We also reweight the two terms in Eq. 9 to give more importance to outcome modeling. Hyperparameters are selected via lightweight tuning on factual-validation sets, guided by distributional metrics and KL–capacity diagnostics; for the transformer we adopt the architecture and base hyperparameters of Melnychuk et al. (2022).

**Inference and sampling cost.** At test time, we apply Algorithm 1 with the learned parameters. For a given anchor time $t$ and treatment plan $\bar{\mathbf{a}}_{t:t+\tau-1}$, we compute the history embedding $\mathbf{r}_t = f_\omega(\bar{\mathbf{h}}_t)$ once, then roll out the latent state and decoder as in Section 4.1. Because decoded observations are never fed back, the inner loop consists only of GRU updates and decodes and vectorizes over

$M$ Monte Carlo paths with a shared $\mathbf{r}_t$. As discussed in Section 4.1 and Corollary 4.3, we can decode at all steps (scope=all) or only at a subset $S \subseteq \{1, \ldots, \tau\}$ (e.g., scope=last for $S = \{\tau\}$) without changing the underlying interventional law, so we pay decoder cost only at horizons of interest. For $M$ MC samples and horizon $\tau$, the cost is $\mathcal{O}\big(\mathrm{cost}(f_\omega) + M[\tau(\mathrm{cost}(\mathrm{GRU}^{(z)}) + \mathrm{cost}(\kappa_\psi)) + |S|\,\mathrm{cost}(D_\theta^{(y)})]\big)$, where $|S| \leq \tau$ is the number of decoded steps and $\mathrm{cost}(f_\omega)$ is paid once. By contrast, a data-space rollout has cost $\mathcal{O}\big(\mathrm{cost}(f_\omega) + M\tau[\mathrm{cost}(\mathrm{GRU}^{(L)}) + \mathrm{cost}(\kappa_\psi) + \mathrm{cost}(D_\theta^{(x,y)})]\big)$, since all steps and both $X$ and $Y$ must be decoded, and a full autoregressive model with decoder scales as $\mathcal{O}\big(M\tau[\mathrm{cost}(f_\omega) + \mathrm{cost}(D_\theta^{(x,y)})]\big)$ (G-Net is of this type, but uses a hold-out error set instead of a decoder). Overall, our model reduces sample cost by (i) computing $f_\omega$ once and reusing it across $M$ and all $\tau$ steps, enabling a high-capacity transformer only for the up-to-$t$ sequence; (ii) decoding selectively so the $D_\theta$ term scales with $|S|$ (e.g., $|S|{=}1$ for last); (iii) decoding only $D_\theta^{(y)}$ and skipping $D_\theta^{(x)}$ at inference; and (iv) updating the GRU in latent space ($\mathrm{GRU}^{(z)}$) instead of data space ($\mathrm{GRU}^{(L)}$), which can yield gains when $d_z \ll d_L$.

## 5 EVALUATION

**Datasets.** Following common practice in benchmarking for POs inference (Bica et al.; Melnychuk et al., 2022), we make use of a semi-synthetic dataset for validating our approach, as it allows to compute ground truth POs. Additionally, we also use a real-world dataset to demonstrate the practical applicability of our approach. These datasets were selected because they have a considerable number of covariates to adjust for, which is the type of setting for which our model can be more useful. *Semi-synthetic:* from ICU data (Johnson et al., 2016), we generate high-dimensional, long-range trajectories with treatment effects and endogenous/exogenous dependencies following Melnychuk et al. (2022); Schulam & Saria (2017); confounding is controllable and ground-truth POs are known. We detected violations of the positivity assumption in the original form of this dataset, presented in Melnychuk et al. (2022). Despite having become a standard benchmark, the aforementioned positivity violations make it unsuitable for evaluation of methods with the standard causal assumptions. For this reason, we make several modifications to avoid this problem. We detail the detected problems in the original form of the dataset and the changes we make in F. *Real-world:* a fully observational benchmark from MIMIC-III using the same cohort definition and preprocessing as the semi-synthetic setup (sampling grid, variable definitions, imputation, and discrete action categories per Melnychuk et al., 2022); lacking ground-truth counterfactuals, evaluation targets predictive quality of observational next steps. Variables include standard ICU vitals/labs and intervention-derived action indicators. We refer to App. F for more details about both datasets.

**Baselines.** To evaluate our model, we use several baselines that handle aleatoric uncertainty and deliver distributional estimates. We use G-Net (Li et al., 2021) as an alternative implementation of the g-formula and, for better comparability, its extension Transformer G-Net (Xiong et al., 2024), which we implement with the same multi-input transformer architecture used in G-Latent. To the best of our knowledge, these are the only previous works that estimate aleatoric uncertainty of individualized POs in a discrete setting. We also compare with Causal Transformer (CT) (Melnychuk et al., 2022): in its original form for point estimate metrics, and with two distributional adaptations: CT-Gaussian, with a Gaussian head, and CT-CRPS, with a CRPS head, analogous to G-Latent decoder. Among the non-distributional models for individualized POs, we chose to adapt CT as it is a strong baseline and G-Latent shares its transformer-based processing of history data. As for our model, we present three variants apart from the one described in 4.3: G-Latent with a full Gaussian reconstruction, and two variants that perform the rollout in the data space: one with CRPS decoder and another one with full Gaussian decoder. We call these variants G-VAE, and D.S. accounts for data space. We specify the details in App. G. In continuous settings, we are aware of two works that estimate data distributions: Hess et al. and Mu et al. (2025). We exclude the former because it introduces a heavy machinery for epistemic uncertainty and continuous time processing that makes it very expensive to train, while its way to handle aleatoric uncertainty is a Gaussian head, which is already covered by CT-Gaussian. As for the latter, we exclude it because it addresses a slightly different setting (expert models) and because it was released over one month before the submission of this work, without available code.

Table 1: Results at selected steps $t' \in \{3, 5, 8, 11\}$ for the (new) semi-synthetic dataset. Metrics: Energy Score (ES ↓) (per step and across steps), KDE-Loglikelihood (KDE-LL ↑), RMSE ↓, Calibration MAE ↓.

| Model | $t' = 3$ | | | $t' = 5$ | | | $t' = 8$ | | | $t' = 11$ | | | Global | |
| | ES ↓ | KDE-LL ↑ | RMSE ↓ | ES ↓ | KDE-LL ↑ | RMSE ↓ | ES ↓ | KDE-LL ↑ | RMSE ↓ | ES ↓ | KDE-LL ↑ | RMSE ↓ | ES ↓ | Cal. MAE ↓ |
|---|---|---|---|---|---|---|---|---|---|---|---|---|---|---|
| G-Net | 0.39±0.04 | −1.27±0.17 | 0.64±0.07 | 0.51±0.05 | −1.74±0.21 | 0.81±0.09 | 0.63±0.07 | −2.18±0.25 | 0.98±0.11 | 0.70±0.08 | −2.45±0.28 | 1.09±0.12 | 1.85±0.20 | 6.29±1.35 |
| Transformer G-Net | 0.40±0.05 | −1.35±0.21 | 0.66±0.08 | 0.50±0.07 | −1.69±0.31 | 0.80±0.13 | 0.58±0.11 | −2.01±0.45 | 0.92±0.19 | 0.64±0.14 | −2.24±0.56 | 1.00±0.23 | 1.71±0.11 | 6.97±2.06 |
| CT (CRPS) | 0.32±0.07 | −1.00±0.30 | 0.66±0.08 | 0.41±0.07 | −1.40±0.34 | 0.71±0.10 | 0.50±0.11 | −1.87±0.35 | 0.84±0.10 | 0.57±0.07 | −2.22±0.36 | 0.92±0.10 | 1.52±0.23 | 13.14±2.55 |
| CT (Gaussian) | 0.30±0.07 | −0.91±0.31 | 0.54±0.13 | 0.37±0.08 | −1.17±0.35 | 0.64±0.14 | 0.44±0.09 | −1.44±0.38 | 0.74±0.14 | 0.49±0.09 | −1.64±0.38 | 0.81±0.14 | 1.35±0.29 | 7.88±1.76 |
| CT | ... | ... | **0.43±0.10** | ... | ... | **0.53±0.12** | ... | ... | **0.60±0.13** | ... | ... | **0.65±0.13** | ... | ... |
| D.S. G-VAE (Gaussian) | 0.49±0.04 | −2.26±0.12 | 0.54±0.09 | 0.58±0.06 | −2.56±0.16 | 0.66±0.11 | 0.64±0.07 | −2.72±0.18 | 0.76±0.13 | 0.67±0.07 | −2.78±0.18 | 0.83±0.13 | 2.01±0.20 | 14.99±0.86 |
| D.S. G-VAE (CRPS) | **0.28±0.05** | **−0.89±0.25** | 0.49±0.10 | **0.35±0.06** | **−1.14±0.29** | 0.59±0.12 | 0.42±0.07 | −1.40±0.29 | 0.69±0.12 | 0.47±0.06 | −1.58±0.26 | 0.76±0.12 | 1.28±0.21 | 5.48±3.08 |
| G-Latent (Gaussian) | 0.38±0.04 | −1.70±0.14 | 0.53±0.09 | 0.42±0.05 | −1.80±0.16 | 0.61±0.11 | 0.46±0.06 | −1.90±0.18 | 0.69±0.12 | 0.48±0.06 | −1.95±0.18 | 0.73±0.12 | 1.51±0.18 | 10.14±1.36 |
| G-Latent (CRPS) | 0.29±0.05 | −0.95±0.21 | 0.51±0.10 | **0.35±0.06** | −1.18±0.26 | 0.60±0.12 | **0.40±0.07** | **−1.37±0.29** | 0.68±0.13 | **0.43±0.08** | **−1.50±0.29** | 0.73±0.13 | **1.25±0.23** | **2.95±1.37** |

Table 2: Results at selected steps $t' \in \{2, 3, 5, 6\}$ for the real-world dataset. Metrics: Energy Score (ES ↓) (per step and across steps), KDE-Loglikelihood (KDE-LL ↑), and RMSE ↓.

| Model | $t' = 2$ | | | $t' = 3$ | | | $t' = 5$ | | | $t' = 6$ | | | Global |
| | ES ↓ | KDE-LL ↑ | RMSE ↓ | ES ↓ | KDE-LL ↑ | RMSE ↓ | ES ↓ | KDE-LL ↑ | RMSE ↓ | ES ↓ | KDE-LL ↑ | RMSE ↓ | ES ↓ |
|---|---|---|---|---|---|---|---|---|---|---|---|---|---|
| G-Net | 5.32±0.08 | −3.92±0.05 | 11.84±0.24 | 5.82±0.08 | −4.11±0.05 | 12.83±0.29 | 6.98±0.08 | −4.55±0.07 | 14.05±0.30 | 7.44±0.11 | −4.83±0.04 | 14.23±0.29 | 18.35±0.33 |
| Transformer G-Net | 5.28±0.06 | −3.89±0.06 | 10.90±0.30 | 5.84±0.08 | −4.06±0.08 | 11.67±0.26 | 6.47±0.08 | −4.30±0.06 | 12.96±0.32 | 6.90±0.08 | −4.48±0.04 | 13.21±0.29 | 16.70±0.23 |
| CT (CRPS) | 4.92±0.06 | −3.81±0.06 | 10.10±0.29 | 5.39±0.08 | −3.94±0.06 | 10.53±0.26 | 5.77±0.08 | −4.08±0.04 | 10.75±0.29 | 5.86±0.07 | −4.19±0.06 | 10.91±0.28 | 14.61±0.27 |
| CT (Gaussian) | 5.25±0.06 | −3.92±0.06 | 10.41±0.29 | 5.71±0.08 | −4.04±0.07 | 10.74±0.29 | 6.15±0.07 | −4.18±0.06 | 11.01±0.34 | 6.34±0.08 | −4.24±0.07 | 11.25±0.30 | 15.55±0.23 |
| CT | ... | ... | **9.00±0.23** | ... | ... | **9.57±0.24** | ... | ... | **10.16±0.27** | ... | ... | **10.35±0.31** | ... |
| D.S. G-VAE (Gaussian) | 5.51±0.08 | −3.90±0.06 | 9.58±0.25 | 5.99±0.08 | −3.98±0.06 | 10.29±0.22 | 6.34±0.06 | −4.03±0.05 | 10.88±0.26 | 6.44±0.07 | −4.04±0.05 | 11.04±0.29 | 15.98±0.23 |
| D.S. G-VAE (CRPS) | 4.89±0.08 | −3.82±0.06 | 9.40±0.22 | 5.36±0.08 | −3.92±0.05 | 10.09±0.25 | 5.70±0.07 | −3.99±0.06 | 10.63±0.29 | 5.82±0.06 | −4.04±0.06 | 10.79±0.30 | 14.38±0.19 |
| G-Latent (Gaussian) | 5.27±0.06 | −3.85±0.06 | 9.42±0.23 | 5.64±0.08 | −3.89±0.06 | 10.09±0.23 | 5.96±0.07 | **−3.94±0.04** | 10.64±0.19 | 6.07±0.07 | **−3.95±0.06** | 10.80±0.25 | 15.21±0.26 |
| G-Latent (CRPS) | **4.85±0.05** | **−3.79±0.06** | 9.23±0.20 | **5.25±0.08** | **−3.88±0.05** | 9.79±0.24 | **5.60±0.09** | **−3.94±0.05** | 10.36±0.29 | **5.72±0.06** | −3.96±0.06 | 10.55±0.28 | **14.23±0.23** |

**Metrics.** Our model produces MC samples at each prediction step. We evaluate with: *RMSE of the predictive mean*, computed from the average of MC samples at each step (lower is better); *Energy Score (ES)*, a strictly proper multivariate scoring rule that reduces to CRPS in the univariate case and assesses distributional fit. We report it per step and over the full trajectory to capture temporal coherence (lower is better); and *KDE log-likelihood (KDE-LL*, the log-likelihood of the observed outcome under a Gaussian kernel density estimate fit to the model's samples, reflecting density fit (higher is better). After trying over ten bandwidths for each dataset and baseline, we selected the one with general better results to report here. For the semi-synthetic dataset, we report results for two additional bandwidth (see App. J). In general, the best bandwidths provided better results consistently across models. For the semi-synthetic dataset, We also assess *calibration* via *quantile coverage*: for $q \in \{0.1, \ldots, 0.9\}$ we compute, per step and per outcome dimension (and aggregated across steps), the fraction of test outcomes below the MC-estimated $q$-quantile (ideal coverage equals $q$). As a scalar summary we report *Calibration MAE*, the mean absolute gap between empirical and nominal coverage averaged over quantiles, dimensions, and steps (lower is better). To obtain the metrics, we used 50 and 40 MC samples for the semi-synthetic and the real-world dataset, respectively. See App. H for more details on the metrics.

**Results.** We ran all experiments in AWS SageMaker on an ml.g5.4xlarge instance (A10G GPU, 24 GiB VRAM). We report selected steps in Table 1 (semi-synthetic, modified) and Table 2 (real-world), with full results—and the original semi-synthetic benchmark—in App. J. Semi-synthetic runs use five random seeds; real-world runs use four; intervals denote standard deviations.[1] Across both datasets, **G-Latent** attains the strongest *distributional* performance, especially at larger horizons. On semi-synthetic data, *G-VAE–CRPS* remains competitive with *G-Latent–CRPS*—showing small ES gaps overall and occasional wins at short horizons—whereas among the Gaussian variants the gap between *G-Latent* and *G-VAE* is pronounced: Gaussian heads are more error-prone, and the latent rollout reduces accumulation error. KDE log-likelihood consistently favors **G-Latent** at large steps (across all tested bandwidths). On the real-world cohort, *G-Latent–CRPS* is best at every reported step and globally. For calibration on the semi-synthetic benchmark, *G-Latent–CRPS* achieves the lowest Calibration MAE by a clear margin, while Gaussian variants fare markedly worse. In App. J we show extensive quantile coverage tables. Regarding other baselines, CT with Gaussian/CRPS heads trails the latent models on distributional metrics, while the point-estimate CT attains the lowest RMSE (as expected for a point forecaster); G-Net and Transformer G-Net lag further behind on ES and KDE-LL. Overall, *G-Latent–CRPS* provides the best distributional metrics at long horizons while remaining competitive on point accuracy, and it clearly outperforms prior g-computation–based models.

---

[1]See App. J for complete tables and diagnostics.

We measure end-to-end test-set inference time on the semi-synthetic dataset (50 MC samples; 11 projection-horizon steps). Table 3 reports the results: decoding all steps with G-Latent–CRPS takes 00:19:27 (1,167 s $\pm$ 12 s), while decoding only the last step takes 00:07:11 (431 s $\pm$ 5 s)—an $\approx 63\%$ reduction that is valuable when only a few horizons are needed, since non-latent rollouts must decode every step. For G-VAE–CRPS, inference time is 00:25:42 (1,542 s $\pm$ 12 s), about 32% slower than G-Latent–CRPS (all steps). This gap stems from our decoupled decoder, which allows G-Latent-CRPS to decode outcomes without covariates. In our implementation, the outcome and covariate decoders share three layers (App. D); further decoupling could yield additional gains.

Table 3: Test-set inference time on the semi-synthetic dataset (50 MC samples; 11 projection-horizon steps) (hh:mm:ss).

| Method | hh:mm:ss |
|---|---|
| G-Latent (CRPS) [all] | 00:19:27 $\pm$ 12s |
| G-Latent (CRPS) [last] | 00:07:11 $\pm$ 05s |
| G-Latent (Gaussian) [all] | 00:20:16 $\pm$ 15s |
| G-Latent (Gaussian) [last] | 00:07:26 $\pm$ 08s |
| G-VAE (CRPS) | 00:25:42 $\pm$ 12s |
| G-VAE (Gaussian) | 00:20:05 $\pm$ 11s |
| Transformer G-Net | 01:03:21 $\pm$ 36s |
| G-Net | 00:05:45 $\pm$ 05s |
| CT–CRPS | 00:59:25 $\pm$ 29s |
| CT–Gaussian | 00:53:08 $\pm$ 19s |

The Gaussian head yields similar wall-clock for G-Latent—00:20:16 (1,216 s $\pm$ 15 s) for all steps and 00:07:26 (446 s $\pm$ 8 s) for last-step decoding—and 00:20:05 (1,205 s $\pm$ 11 s) for G-VAE (there is no covariate decoupling in the Gaussian head models). Among other baselines, Transformer G-Net and CT–CRPS/CT–Gaussian are substantially slower at 01:03:21 (3,801 s $\pm$ 36 s), 00:59:25 (3,565 s $\pm$ 29 s), and 00:53:08 (3,188 s $\pm$ 19 s), respectively, while G-Net is faster at 00:05:45 (345 s $\pm$ 5 s). For all the baselines, we fully tensorize and cache recurrent state (e.g., Transformer hidden states in Transformer G-Net and CT–CRPS/Gaussian), so each step only processes the last MC prediction rather than recomputing the entire history. In summary, all full transformer–based models exceed 50 minutes per test set, whereas G-Latent (and its variants) substantially reduces inference time by using the transformer only to encode the history up-to-$t'$, then updating the representation during the MC rollout with a lightweight GRU. Our tensorized and cached implementation of G-Net achieves very low inference times because it uses a lightweight RNN to process data and, unlike G-Latent, has no decoder—it injects residual noise. However, this reduces its expressivity and adaptability to particular data distributions.

## 6    Conclusions and Limitations

In this work, we introduce G-Latent, a novel method for distributional estimation of individualized POs under time-varying treatment effects for discrete settings, with identifiability guarantees through g-computation in the latent space. We demonstrate the general efficacy of our approach, both theoretically and experimentally. Also, we show that our method is efficient at sampling compared with other variants that perform g-computation in the data-space. We identify two potential limitations: the first is related to the latent factorization in eq. 6, fundamental for G-Latent. This assumption would be violated, for example, under posterior collapse (Lucas et al., 2019), which is relatively common in VAE training and prevents latent representations from properly representing data. We did not observe this problem in the experiments, but it is important to be careful with that. On the other hand, another potential limitation comes from the CRPS decoder; as An & Jeon (2023) discuss, the ALD-decoder assumes that the different elements of $\mathbf{Y}_{t+t'}$ (if multivariate) are independent given $\mathbf{z}$. If the assumption fails, cross-dimensional dependence may remain unmodeled. However, neither DistVAE nor us empirically observe this problem (G-Latent has strong ES metrics). Finally, our focus in this work is aleatoric uncertainty; epistemic uncertainty is orthogonal and can be added with MC dropout or deep ensembles, or more formally via Bayesian priors.

We restrict attention to g-computation–based estimators rather than IPTW/MSM-style generative baselines (e.g., Wu et al., 2024). In principle, IPTW could be adapted to our conditional, trajectory-level estimands, but would require high-dimensional propensity models (or conditional treatment densities for continuous treatments) and weighted conditional density estimation, which can lead to unstable importance weights in long-horizon, high-dimensional settings. Designing and evaluating IPTW/MSM-style generative models for individualized distributional potential outcomes remains an interesting direction for future work.

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

## A   ASSUMPTIONS FOR CAUSAL IDENTIFICATION

We work within the potential outcomes paradigm (Rubin, 2005) and its extension to temporal treatments and outcomes (Robins et al., 2000), a setup also adopted by prior sequence models for treatment effect inference (e.g., Lim, 2018; Bica et al.). In this framework, identification of counterfactual distributions over time (and, in particular, the $\tau$-step conditional mean from Eq. (1)) relies on three standard conditions on the data-generating process.

**Assumption A.1 (Consistency).**   For any fixed treatment history $\bar{\mathbf{a}}_t$, if the realized actions satisfy $\bar{\mathbf{A}}_t = \bar{\mathbf{a}}_t$, then
$$\mathbf{Y}_{t+1}[\bar{\mathbf{a}}_t] \;=\; \mathbf{Y}_{t+1}.$$
That is, under the actually received treatment sequence, the relevant potential outcome coincides with the observed one.

**Assumption A.2 (Sequential Overlap/Positivity).**   For any history value $\bar{\mathbf{h}}_t$ in the support of $\bar{\mathbf{H}}_t$, each admissible action has positive probability:
$$0 \;<\; p(\mathbf{A}_t = \mathbf{a}_t \mid \bar{\mathbf{H}}_t = \bar{\mathbf{h}}_t) \;<\; 1 \quad \text{whenever} \quad p(\bar{\mathbf{H}}_t = \bar{\mathbf{h}}_t) > 0.$$

**Assumption A.3 (Sequential Ignorability / No Unmeasured Confounding).**   Conditioning on the observed history renders the current action as-if randomized with respect to the next-step potential outcome:
$$\forall\, t \text{ and } \forall\, \bar{\mathbf{a}}_{t:t+\tau-1} : \; \mathbf{A}_t \;\perp\!\!\!\perp\; \big(\bar{\mathbf{L}}_{t+1:t+\tau}[\bar{\mathbf{a}}_{t:t+\tau-1}],\; \bar{\mathbf{Y}}_{t+1:t+\tau}[\bar{\mathbf{a}}_{t:t+\tau-1}]\big) \,\big|\, \bar{\mathbf{H}}_t.$$

**Corollary A.4 (g-computation; Robins & Hernan, 2008).**   Under A.1–A.3, the $\tau$-step-ahead conditional mean under a fixed intervention path $\bar{\mathbf{a}}_{t:t+\tau-1}$ is identified by the longitudinal $g$-formula.

## B  MULTI-INPUT TRANSFORMER

**Scope.**  This appendix details the *encoder* we use to compute the history embedding $\mathbf{r}_t = f_\omega(\bar{\mathbf{h}}_t)$ from the factual history $\bar{\mathbf{h}}_t = \{\bar{\mathbf{X}}_t, \bar{\mathbf{A}}_{t-1}, \bar{\mathbf{Y}}_t, \mathbf{V}\}$. It follows the multi-input transformer design of Melnychuk et al. (2022) (three streams with cross-attention and shared relative positional encodings), but we do *not* use their balancing loss and we *never* feed model predictions back into the transformer. The output of this encoder is a single fused representation $\mathbf{r}_t$ that our model uses downstream (Sec. 4.3).

### B.1  INPUTS AND TOKENIZATION

Let $b = 1, \dots, B$ index transformer blocks and $d_h$ the model width. For the first block, we map each sequence to hidden states via time-shared linear layers:

$$\mathbf{A}_{1:t}^0 = \mathrm{Linear}_A(\bar{\mathbf{A}}_{t-1}), \qquad \mathbf{X}_{1:t}^0 = \mathrm{Linear}_X(\bar{\mathbf{X}}_t), \qquad \mathbf{Y}_{1:t}^0 = \mathrm{Linear}_Y(\bar{\mathbf{Y}}_t), \qquad \tilde{\mathbf{V}} = \mathrm{Linear}_V(\mathbf{V}),$$

where $\bar{\mathbf{A}}_{t-1} = (\mathbf{A}_1, \dots, \mathbf{A}_{t-1}, \mathbf{0})$ is a left-shifted treatment stream aligned with our start-of-interval indexing (decision $\mathbf{A}_j$ precedes $(\mathbf{Y}_{j+1}, \mathbf{X}_{j+1})$). Subsequent blocks receive the previous block's outputs.

We denote the stream-specific hidden sequences at block $b$ by $\mathbf{A}_{1:t}^b$, $\mathbf{X}_{1:t}^b$, and $\mathbf{Y}_{1:t}^b$ ($\in \mathbb{R}^{t \times d_h}$).

### B.2  MASKED SELF-ATTENTION WITH RELATIVE POSITIONAL ENCODINGS

Each stream applies masked multi-head self-attention (causal mask so a position $i$ only attends to $j \le i$) with *relative* positional encodings (RPE). For head dimension $d_{qk}$, attention at position $i$ is

$$\mathrm{Attn}_i(Q, K, V) = \sum_{j=1}^{t} \alpha_{ij}(V_j + a_{ij}^V), \qquad \alpha_{ij} = \mathrm{softmax}_j\left(\frac{Q_i^\top(K_j + a_{ij}^K)}{\sqrt{d_{qk}}}\right), \tag{11}$$

$$a_{ij}^V = w_{\mathrm{clip}(j-i,\ell_{\max})}^V, \quad a_{ij}^K = w_{\mathrm{clip}(j-i,\ell_{\max})}^K, \quad \mathrm{clip}(x, \ell_{\max}) = \max\{-\ell_{\max}, \min\{\ell_{\max}, x\}\},$$

with trainable $w_\ell^V, w_\ell^K \in \mathbb{R}^{d_{qk}}$ for $\ell \in \{-\ell_{\max}, \dots, 0\}$. These Toeplitz-structured encodings depend only on relative distance and are shared across blocks and streams. Layer normalization and residual connections wrap the attention sublayer, and a position-wise feed-forward network $\mathrm{FF}(h) = \mathrm{Linear}(\mathrm{ReLU}(\mathrm{Linear}(h)))$ follows, again with residual+LN.

### B.3  CROSS-ATTENTION BETWEEN STREAMS AND STATIC COVARIATES

To couple signals across modalities, each block augments self-attention with *cross-attentions* between the three streams. Using tildes for post-self-attention states and writing $\mathrm{MHA}(Q, K, V)$ for multi-head attention,

$$\tilde{\mathbf{A}}_X^{b-1} = \mathrm{LN}\Big(\mathrm{MHA}\big(Q(\tilde{\mathbf{A}}^{b-1}), K(\mathbf{X}^{b-1}), V(\mathbf{X}^{b-1})\big) + \tilde{\mathbf{A}}^{b-1}\Big), \tag{12}$$

$$\tilde{\mathbf{A}}_Y^{b-1} = \mathrm{LN}\Big(\mathrm{MHA}\big(Q(\tilde{\mathbf{A}}^{b-1}), K(\mathbf{Y}^{b-1}), V(\mathbf{Y}^{b-1})\big) + \tilde{\mathbf{A}}^{b-1}\Big), \tag{13}$$

$$\tilde{\mathbf{X}}_A^{b-1} = \mathrm{LN}\Big(\mathrm{MHA}\big(Q(\tilde{\mathbf{X}}^{b-1}), K(\mathbf{A}^{b-1}), V(\mathbf{A}^{b-1})\big) + \tilde{\mathbf{X}}^{b-1}\Big), \tag{14}$$

$$\tilde{\mathbf{X}}_Y^{b-1} = \mathrm{LN}\Big(\mathrm{MHA}\big(Q(\tilde{\mathbf{X}}^{b-1}), K(\mathbf{Y}^{b-1}), V(\mathbf{Y}^{b-1})\big) + \tilde{\mathbf{X}}^{b-1}\Big), \tag{15}$$

$$\tilde{\mathbf{Y}}_X^{b-1} = \mathrm{LN}\Big(\mathrm{MHA}\big(Q(\tilde{\mathbf{Y}}^{b-1}), K(\mathbf{X}^{b-1}), V(\mathbf{X}^{b-1})\big) + \tilde{\mathbf{Y}}^{b-1}\Big), \tag{16}$$

$$\tilde{\mathbf{Y}}_A^{b-1} = \mathrm{LN}\Big(\mathrm{MHA}\big(Q(\tilde{\mathbf{Y}}^{b-1}), K(\mathbf{A}^{b-1}), V(\mathbf{A}^{b-1})\big) + \tilde{\mathbf{Y}}^{b-1}\Big). \tag{17}$$

We then pool the two cross-attended views per stream and inject static covariates at every time step:

$$\breve{\mathbf{A}}^{b-1} = \tilde{\mathbf{A}}_X^{b-1} + \tilde{\mathbf{A}}_Y^{b-1} + \mathbf{1}\tilde{\mathbf{V}}^\top, \quad \breve{\mathbf{X}}^{b-1} = \tilde{\mathbf{X}}_A^{b-1} + \tilde{\mathbf{X}}_Y^{b-1} + \mathbf{1}\tilde{\mathbf{V}}^\top, \tag{18}$$

$$\breve{\mathbf{Y}}^{b-1} = \tilde{\mathbf{Y}}_X^{b-1} + \tilde{\mathbf{Y}}_A^{b-1} + \mathbf{1}\tilde{\mathbf{V}}^\top, \tag{19}$$

followed by parallel FF+residual+LN sublayers to yield $\mathbf{A}^b, \mathbf{X}^b, \mathbf{Y}^b$. Treatments remain left-shifted throughout (so treatment token at index $i$ aligns with covariate/outcome tokens at $i+1$).

### B.4 FUSION TO A SINGLE HISTORY EMBEDDING $\mathbf{r}_t$

After the final block $B$, we fuse the three streams by element-wise averaging at each time $i \leq t$, then project with a linear layer and ELU:

$$\tilde{\boldsymbol{\Phi}}_i = \frac{1}{3}\big(\mathbf{A}_{i-1}^B + \mathbf{X}_i^B + \mathbf{Y}_i^B\big), \qquad \boldsymbol{\Phi}_i = \mathrm{ELU}\big(\mathrm{Linear}(\tilde{\boldsymbol{\Phi}}_i)\big), \quad \mathbf{r}_t := \boldsymbol{\Phi}_t \in \mathbb{R}^{d_r}.$$

We use only the factual $\{\mathbf{X}_{1:t}, \mathbf{A}_{1:t-1}, \mathbf{Y}_{1:t}\}$ to build $\mathbf{r}_t$; predicted outcomes are *never* fed back into the encoder.

**Remarks.** (i) All attention modules use the causal mask and the same RPE as in Eq. 11. (ii) Static covariates $\mathbf{V}$ are injected at every block/time step via $\tilde{\mathbf{V}}$. (iii) Dropout is applied after linear layers in attention and feed-forward sublayers.

## C DISTVAE-STYLE LOSS: DERIVATION AND DISCUSSION

We adapt the continuous-variable objective of An & Jeon (2023) to our setting (ignoring categorical variables). Let $x = (x_1, \ldots, x_p)$ denote continuous observations (here, $x \equiv \mathbf{y}$) and $z$ the latent. DistVAE assumes an *ALD* (asymmetric Laplace) decoder *mixed* over a quantile level $\alpha \in (0,1)$:

$$p(x; \theta, \beta) = \iint p(x \mid z, \alpha; \theta, \beta)\, p(z)\, p(\alpha)\, d\alpha\, dz, \qquad p(x \mid z, \alpha; \theta, \beta) = \prod_{j=1}^{p} p(x_j \mid z, \alpha; \theta_j, \beta), \tag{20}$$

where, for each coordinate,

$$p(x_j \mid z, \alpha; \theta_j, \beta) = \frac{\alpha(1-\alpha)}{\beta} \exp\left(-\rho_\alpha\Big(\frac{x_j - D_j(\alpha, z; \theta_j)}{\beta}\Big)\right), \qquad \rho_\alpha(u) = (\alpha - \mathbb{I}\{u < 0\})\, u. \tag{21}$$

Here $D_j(\alpha, z; \theta_j)$ is the conditional *quantile function* (ALD location)[2], $\beta > 0$ is a scale constant, and $\rho_\alpha$ is the pinball loss.

**Assumption 1 (DistVAE).** (i) $\{x_j\}$ are conditionally independent given $z$; (ii) (discrete variables independent of $\alpha$; not used here); (iii) $\alpha \perp z$. Item (i) is the usual VAE factorization; (iii) treats $\alpha$ as a prior (no $q(\alpha \mid x)$), which is key to the proper-scoring-rule objective below.

### C.1 FINITE-$K$ NEGATIVE ELBO (COMPOSITE QUANTILE)

Approximate the $\alpha$-integral by a uniform grid $\alpha_k = \frac{k}{K}$, $k = 1, \ldots, K$, with $p(\alpha_k) = \frac{1}{K}$, and introduce $q_\phi(z \mid x)$. A Jensen step yields, up to additive constants independent of $(\theta, \phi)$,

$$-\mathrm{ELBO}_K(\theta, \phi) = \mathbb{E}_{q_\phi(z|x)}\left[\frac{1}{K}\sum_{k=1}^{K}\sum_{j=1}^{p} \rho_{\alpha_k}\big(x_j - D_j(\alpha_k, z; \theta_j)\big)\right] + \beta\, \mathrm{KL}\big(q_\phi(z \mid x) \,\|\, p(z)\big) + C_K, \tag{22}$$

so the reconstruction is a *composite quantile* (average ALD NLL across $\{\alpha_k\}$).

### C.2 LIMIT $K \to \infty$: CRPS OBJECTIVE AND DISTVAE LOSS

Under mild integrability/continuity in $\alpha$,

$$\lim_{K \to \infty} \frac{1}{K}\sum_{k=1}^{K} \rho_{\alpha_k}\big(x_j - D_j(\alpha_k, z; \theta_j)\big) = \int_0^1 \rho_\alpha\big(x_j - D_j(\alpha, z; \theta_j)\big)\, d\alpha, \tag{23}$$

$$\lim_{K \to \infty} \frac{1}{K}\sum_{k=1}^{K} \log \alpha_k(1 - \alpha_k) = \int_0^1 \log \alpha(1 - \alpha)\, d\alpha. \tag{24}$$

---

[2]An & Jeon (2023) enforce $D_j(\cdot, z)$ to be monotone in $\alpha$ (to avoid quantile crossing) via an isotonic-spline parameterization. We do not impose this constraint: it adds architectural restrictions and, in our experiments, occasional finite-$K$ crossings had negligible effect on CRPS or downstream rollouts.

Hence $-\mathrm{ELBO}_K$ converges to

$$\mathcal{L}_{\mathrm{DistVAE}}(\theta, \phi) = \mathbb{E}_{q_\phi(z|x)}\left[\sum_{j=1}^{p} \int_0^1 \rho_\alpha\big(x_j - D_j(\alpha, z; \theta_j)\big)\, d\alpha\right] \;+\; \beta\, \mathrm{KL}\big(q_\phi(z \mid x) \,\|\, p(z)\big) \;+\; C,$$

(25)

where $\int_0^1 \rho_\alpha(\cdot)\, d\alpha$ equals the *Continuous Ranked Probability Score* (CRPS) for the model CDF. In practice we estimate it by Monte Carlo over $\alpha \sim \mathrm{Unif}(0, 1)$. Thus the "ALD NLL (MC–CRPS)" reconstruction is the $K \to \infty$ limit of a valid ELBO (not a heuristic).

### C.3 Why this helps vs. Gaussian decoding

**Distributional capacity.** Gaussian decoders impose symmetry and typically homoscedastic noise, and in practice often compensate for mean misspecification by *inflating the predicted variance*, yielding over-dispersed (underconfident) forecasts. ALD/quantile decoding directly captures *skewness* and *heteroscedasticity* across $\alpha$ while preserving VAE advantages: (i) a likelihood-derived proper scoring rule (CRPS) for reconstruction, (ii) simple sampling via inverse transform ($u \sim \mathrm{Unif}(0, 1)$ then $x_j = D_j(u, z)$), (iii) a tractable latent KL. By focusing the loss on quantile locations across $\alpha$, the ALD/CRPS objective discourages variance inflation and typically yields sharper predictive distributions under non-Gaussian data.

### C.4 Our objective (continuous head) in DistVAE form

Identifying $x \equiv \mathbf{y}$ (continuous outcomes), our training loss for the outcome head is

$$\mathcal{L}_{\mathrm{cont}} = \mathbb{E}_{q_\phi(z|\cdot)}\left[\frac{1}{K}\sum_{k=1}^{K}\sum_{j=1}^{d_y} \rho_{\alpha^{(k)}}\big(y_j - D_j(\alpha^{(k)}, z; \theta_j)\big)\right] \;+\; \beta\, \mathrm{KL}\big(q_\phi(z \mid \cdot) \,\|\, p(z)\big), \qquad \alpha^{(k)} \overset{\text{i.i.d.}}{\sim} \mathrm{Unif}(0, 1).$$

(26)

This is exactly the *ALD NLL (MC–CRPS)* plus KL, i.e., the continuous-variable DistVAE objective specialized to our architecture (temporal and cross-outcome dependence are mediated by the latent path; the quantile head supplies the likelihood noise, analogous to a Gaussian decoder's noise).

## D  G-Latent Architecture: Encoder, Temporal Core, and Decoder

**Scope.** This appendix specifies the *network architecture* of G-Latent: the history network $f_\omega$, the temporal core ($\kappa_\psi$ and $\mathrm{GRU}_\gamma$), and the shared conditional VAE ($E_\phi, D_\theta$) reused at every relative step. Training objectives and identification assumptions are described elsewhere.

### D.1 Notation and Shapes

Let $\mathbf{X}_t \in \mathbb{R}^{d_x}$, $\mathbf{Y}_t \in \mathbb{R}^{d_y}$, and $\mathbf{L}_t = (\mathbf{Y}_t, \mathbf{X}_t) \in \mathbb{R}^{d_L}$ with $d_L = d_x + d_y$; treatments $\mathbf{A}_t \in \mathbb{R}^{d_a}$; and static covariates $\mathbf{V} \in \mathbb{R}^{d_v}$. The history network outputs $\mathbf{r}_t \in \mathbb{R}^{d_r}$. At relative step $t' \in \{1, \dots, \tau\}$, the latent is $\mathbf{z}_{t,t'} \in \mathbb{R}^{d_z}$, the temporal state is $\mathbf{s}_{t,t'} \in \mathbb{R}^{d_s}$, and the step context is $\mathbf{c}_{t,t'} \in \mathbb{R}^{d_c}$.

### D.2 History Network $f_\omega$

We use the multi–input transformer of Melnychuk et al. (2022) (full details in App. B). Briefly:

- **Inputs.** Three factual streams up to anchor time $t$: $\bar{\mathbf{X}}_t$, $\bar{\mathbf{Y}}_t$, and left–shifted $\bar{\mathbf{A}}_{t-1}$ (start–of–interval indexing), plus static $\mathbf{V}$. Each stream is linearly projected to the model width; $\mathbf{V}$ is injected at every time step.
- **Blocks.** Each block applies masked multi–head self–attention with shared relative positional encodings per stream, cross–attentions between streams, and a positionwise feed–forward network. All sublayers use residual connections, layer normalization, and dropout.

- **Fusion.** The final per–time states of the three streams are averaged and linearly projected with ELU to yield $\mathbf{r}_t = f_\omega(\bar{\mathbf{h}}_t) \in \mathbb{R}^{d_r}$. No model predictions are fed back into the encoder.

### D.3 TEMPORAL CORE: CONTEXT COMBINER AND LATENT–DRIVEN STATE UPDATE

Given $\mathbf{r}_t$, previous state $\mathbf{s}_{t,t'-1}$, current action $\mathbf{a}_{t+t'-1}$, and relative index $t'$, we form a dense context and update the recurrent state.

**Context combiner.** We concatenate the inputs and project to $d_c$ with a single linear layer:

$$\tilde{\mathbf{c}}_{t,t'} = \left[\, \mathbf{r}_t \,;\, \mathbf{s}_{t,t'-1} \,;\, \mathbf{a}_{t+t'-1} \,;\, t' \,\right] \in \mathbb{R}^{d_r+d_s+d_a+1}, \qquad \mathbf{c}_{t,t'} = \kappa_\psi(\tilde{\mathbf{c}}_{t,t'}) \in \mathbb{R}^{d_c}. \tag{27}$$

**State update (latents only).** A GRUCell updates the temporal state using the latent, the frozen history embedding, the current action, and the step index:

$$\mathbf{s}_{t,t'} = \mathrm{GRU}_\gamma\left(\left[\, \mathbf{z}_{t,t'} \,;\, \mathbf{r}_t \,;\, \mathbf{a}_{t+t'-1} \,;\, t' \,\right], \mathbf{s}_{t,t'-1}\right), \qquad \mathbf{s}_{t,0} = \mathbf{0}. \tag{28}$$

GRU weights are orthogonally initialized and biases are zero–initialized. A data–space variant (not used in our main model) replaces $\mathbf{z}_{t,t'}$ with $\mathbf{l}_{t+t'}$.

### D.4 SHARED CONDITIONAL VAE ($E_\phi, D_\theta$)

A single conditional VAE is reused across steps. Encoder $E_\phi$ outputs a Gaussian posterior over $\mathbf{z}_{t,t'}$, and decoder $D_\theta$ maps $[\mathbf{z}_{t,t'}; \mathbf{c}_{t,t'}; \mathbf{a}_{t+t'-1}]$ to the reconstruction heads. The decoder uses *dense skip concatenation*: after every hidden block, $[\mathbf{z}; \mathbf{c}; \mathbf{a}]$ is re–concatenated to the block output before the next block.

#### D.4.1 ENCODER $E_\phi$

The encoder is an MLP applied to $[\mathbf{l}_{t+t'}; \mathbf{c}_{t,t'}]$ with repeated blocks Linear $\rightarrow$ BatchNorm $\rightarrow$ ReLU $\rightarrow$ Dropout, followed by two linear heads for mean and log–variance:

$$(\boldsymbol{\mu}_{t,t'}, \log \boldsymbol{\sigma}_{t,t'}^2) = E_\phi\left([\mathbf{l}_{t+t'}; \mathbf{c}_{t,t'}]\right) \in \mathbb{R}^{d_z} \times \mathbb{R}^{d_z}, \qquad \mathbf{z}_{t,t'} = \boldsymbol{\mu}_{t,t'} + \boldsymbol{\sigma}_{t,t'} \odot \boldsymbol{\epsilon}, \ \boldsymbol{\epsilon} \sim \mathcal{N}(\mathbf{0}, \mathbf{I}). \tag{29}$$

#### D.4.2 DECODER TRUNK $T_\theta$ WITH DENSE SKIPS

Starting from $\mathbf{h}_0 = [\mathbf{z}_{t,t'}; \mathbf{c}_{t,t'}; \mathbf{a}_{t+t'-1}]$, the trunk applies repeated blocks Linear $\rightarrow$ ReLU $\rightarrow$ Dropout; after each block with output $\mathbf{h}$, we set

$$\mathbf{h} \leftarrow \left[\, \mathbf{h} \,;\, \mathbf{z}_{t,t'} \,;\, \mathbf{c}_{t,t'} \,;\, \mathbf{a}_{t+t'-1} \,\right] \tag{30}$$

before entering the next block. The trunk output $\mathbf{w}_{t,t'}$ feeds the heads below.

**Gaussian (heteroscedastic) decoding path.** When using a purely Gaussian decoder for all $d_L$ coordinates, two linear heads produce mean and positive scale (via softplus):

$$\hat{\boldsymbol{\mu}}_{t,t'} = W_\mu \mathbf{w}_{t,t'} + b_\mu, \qquad \hat{\boldsymbol{\sigma}}_{t,t'} = \mathrm{softplus}(W_\sigma \mathbf{w}_{t,t'} + b_\sigma), \tag{31}$$

yielding a diagonal Gaussian on $\mathbf{L}_{t+t'}$. Optional clamping can be applied to designated coordinates (e.g., nonnegativity of specific outputs) by shifting the corresponding mean channels.

**CRPS / random–quantile outcome path.** When using the distributional outcome head, the decoder splits into:

1. **Outcome quantile head (per outcome, per quantile).** Let $\boldsymbol{\alpha} \in (0,1)^{d_y}$ collect per–outcome quantile levels and draw $A$ i.i.d. samples per outcome. From $\mathbf{w}_{t,t'}$ (optionally after a small shared sub–trunk), each outcome coordinate $j \in \{1, \ldots, d_y\}$ has a dedicated MLP that *re–concatenates* $[\mathbf{z}_{t,t'}; \mathbf{c}_{t,t'}; \mathbf{a}_{t+t'-1}; \alpha_j]$ at every hidden layer and outputs a scalar quantile $\hat{q}_{\alpha_j,t,t',j}$. Stacking across $A$ samples yields $\widehat{\mathbf{Q}}_{t,t'} \in \mathbb{R}^{d_y \times A}$.

2. **Remaining coordinates (Gaussian head).** If $d_L > d_y$, a separate trunk (fed by $[\mathbf{w}_{t,t'}; \mathbf{z}_{t,t'}; \mathbf{c}_{t,t'}; \mathbf{a}_{t+t'-1}]$) outputs $(\hat{\boldsymbol{\mu}}_{\mathrm{rem}}, \log \hat{\boldsymbol{\sigma}}_{\mathrm{rem}}^2)$ for the remaining $d_L - d_y$ coordinates.

This realizes the outcome–specific $\alpha$–aware branches while keeping non–outcome channels Gaussian.

## D.5 PER–STEP FLOW (TRAINING AND INFERENCE INTERFACE)

At each step $t'$:

1. Build the context:
$$\mathbf{c}_{t,t'} = \kappa_\psi\left(\left[\ \mathbf{r}_t\ ;\ \mathbf{s}_{t,t'-1}\ ;\ \mathbf{a}_{t+t'-1}\ ;\ t'\ \right]\right). \tag{32}$$

2. *Training:* encode $[\mathbf{l}_{t+t'}; \mathbf{c}_{t,t'}]$ to obtain $(\boldsymbol{\mu}_{t,t'}, \log\boldsymbol{\sigma}^2_{t,t'})$ and sample $\mathbf{z}_{t,t'}$.

3. Decode with either the Gaussian head to obtain $(\hat{\boldsymbol{\mu}}_{t,t'}, \hat{\boldsymbol{\sigma}}_{t,t'})$ for all coordinates, or the quantile outcome head to obtain $\widehat{\mathbf{Q}}_{t,t'}$ (and Gaussian parameters for any remaining coordinates).

4. Update the state:
$$\mathbf{s}_{t,t'} = \text{GRU}_\gamma\left(\left[\ \mathbf{z}_{t,t'}\ ;\ \mathbf{r}_t\ ;\ \mathbf{a}_{t+t'-1}\ ;\ t'\ \right], \mathbf{s}_{t,t'-1}\right). \tag{33}$$

At inference, $\mathbf{z}_{t,t'} \sim \mathcal{N}(\mathbf{0}, \mathbf{I})$ is sampled independently across steps and Monte Carlo draws; by default only outcomes $\mathbf{Y}$ are decoded, and decoding can be restricted to any subset of steps $S \subseteq \{1, \ldots, \tau\}$.

## D.6 DESIGN NOTES

- **Treatment sensitivity.** Actions enter both the context combiner and *every* decoder block via dense re–concatenation, preserving a short path from treatment to outputs.
- **Relative step embedding.** The scalar index $t'$ (or a small positional code) is concatenated in $\kappa_\psi$ and the GRU input to inform the horizon position without per–step parameters.
- **Normalization and positivity.** BatchNorm is used only in the VAE encoder. Decoder scales are enforced positive with softplus.
- **Parameter sharing.** A single $(E_\phi, D_\theta, \kappa_\psi, \text{GRU}_\gamma)$ instance is reused across all $t'$, improving data efficiency and keeping semantics consistent across horizons.

## D.7 MODULE I/O SUMMARY

| Module | Signature |
|---|---|
| History network $f_\omega$ | $\bar{\mathbf{h}}_t \mapsto \mathbf{r}_t \in \mathbb{R}^{d_r}$ |
| Context combiner $\kappa_\psi$ | $[\mathbf{r}_t; \mathbf{s}_{t,t'-1}; \mathbf{a}_{t+t'-1}; t'] \mapsto \mathbf{c}_{t,t'} \in \mathbb{R}^{d_c}$ |
| Encoder $E_\phi$ | $[\mathbf{l}_{t+t'}; \mathbf{c}_{t,t'}] \mapsto (\boldsymbol{\mu}_{t,t'}, \log\boldsymbol{\sigma}^2_{t,t'}) \in \mathbb{R}^{d_z} \times \mathbb{R}^{d_z}$ |
| Decoder trunk $T_\theta$ | $[\mathbf{z}_{t,t'}; \mathbf{c}_{t,t'}; \mathbf{a}_{t+t'-1}] \mapsto \mathbf{w}_{t,t'}$ (dense skips) |
| Outcome head $D_\theta^{(y)}$ | (CRPS) $[\mathbf{w}_{t,t'}; \alpha] \mapsto \hat{q}_\alpha \in \mathbb{R}$ (per outcome, per $\alpha$) |
| Covariate head $D_\theta^{(x)}$ | (Gaussian) $\mathbf{w}_{t,t'} \mapsto (\hat{\boldsymbol{\mu}}_{\text{rem}}, \log\hat{\boldsymbol{\sigma}}^2_{\text{rem}})$ |
| State update $\text{GRU}_\gamma$ | $[\mathbf{z}_{t,t'}; \mathbf{r}_t; \mathbf{a}_{t+t'-1}; t'], \mathbf{s}_{t,t'-1} \mapsto \mathbf{s}_{t,t'}$ |

# E THEORETICAL INSIGHTS

## E.1 EQUIVALENCE OF LATENT AND DATA-SPACE G-COMPUTATION

We first formalize when sampling *only in latent space* (Alg. 1) is sufficient to recover the interventional laws identified by the sequential g-formula.

**Standing causal assumptions.** We assume the usual conditions for identification by the g-formula: (i) consistency, (ii) sequential ignorability/exchangeability, and (iii) sequential overlap/positivity (cf. App. A).

**Assumption E.1** (Latent factorization and context sufficiency)**.** Let $p_0$ be a fixed prior density on latents (e.g., $\mathcal{N}(\mathbf{0}, \mathbf{I})$). Fix an anchor time $t$ and let $\mathbf{r}_t = f_\omega(\bar{\mathbf{h}}_t)$ denote the history embedding computed at $t$. For each relative step $t' \in \{1, \ldots, \tau\}$ define the latent-state update recursively from $\mathbf{s}_{t,0} = \mathbf{0}$ by
$$\mathbf{s}_{t,t'} = \text{GRU}_\gamma\left(\left[\mathbf{z}_{t,t'}, \mathbf{r}_t, \mathbf{a}_{t+t'-1}, t'\right], \mathbf{s}_{t,t'-1}\right).$$

Assume that for every $t' \in \{1, \ldots, \tau\}$ and every history $\bar{\mathbf{h}}_{t+t'-1}$ the true one-step conditional distribution of $\mathbf{L}_{t+t'} = (\mathbf{Y}_{t+t'}, \mathbf{X}_{t+t'})$ admits the factorization

$$p^\star(\mathbf{l}_{t+t'} \mid \bar{\mathbf{h}}_{t+t'-1}, \mathbf{a}_{t+t'-1}) \;=\; \int p_\theta\big(\mathbf{l}_{t+t'} \mid \mathbf{z}_{t,t'}, \mathbf{c}_{t,t'}, \mathbf{a}_{t+t'-1}\big) \, p_0(\mathbf{z}_{t,t'}) \, d\mathbf{z}_{t,t'},$$

where $\mathbf{c}_{t,t'} = \kappa_\psi(\mathbf{r}_t, \mathbf{s}_{t,t'-1}, \mathbf{a}_{t+t'-1}, t')$. Moreover, for each fixed $(\mathbf{c}, \mathbf{a})$ the map $\mathbf{z} \mapsto p_\theta(\cdot \mid \mathbf{z}, \mathbf{c}, \mathbf{a})$ is a probability-kernel in $\mathbf{L}$ measurable in $(\mathbf{z}, \mathbf{c}, \mathbf{a})$.

Assumption E.1 states that $(\mathbf{r}_t, \mathbf{s}_{t,t'-1})$ is a *sufficient statistic* of $\bar{\mathbf{H}}_{t+t'-1}$ for predicting $\mathbf{L}_{t+t'}$, and that the true stepwise conditional factors through a latent with fixed prior density $p_0$.

*Remark* E.2 (Relation to training). Assumption E.1 is a modeling/realizability statement: it postulates that the one-step conditionals factor through a latent with prior $p_0$ given the context $(\mathbf{r}_t, \mathbf{s}_{t,t'-1}, \mathbf{a}_{t+t'-1}, t')$. Our conditional-VAE training (Sec. 4.3) is the estimation procedure we use to realize this factorization in practice by maximizing the (conditional) ELBO, i.e., approximately minimizing the negative log-likelihood of $p_\theta(\mathbf{L}_{t+t'} \mid \mathbf{z}_{t,t'}, \mathbf{c}_{t,t'}, \mathbf{a}_{t+t'-1})$ under $p_0$. All results that require the assumption hold exactly; with finite data and imperfect training, they hold approximately with the local errors $\{\varepsilon_{t'}\}$ used in Prop. E.10.

*Remark* E.3 (State update uses latent representations). The recurrent state is updated *through latents only* $\big(\mathbf{s}_{t,t'} = \mathrm{GRU}_\gamma([\mathbf{z}_{t,t'}, \mathbf{r}_t, \mathbf{a}_{t+t'-1}, t'], \mathbf{s}_{t,t'-1})\big)$. Thus all predictive information that propagates forward from step $t'$ enters via $\mathbf{z}_{t,t'}$ and the context $\mathbf{c}_{t,t'} = \kappa_\psi(\mathbf{r}_t, \mathbf{s}_{t,t'-1}, \mathbf{a}_{t+t'-1}, t')$. When $\mathbf{z}_{t,t'}$ is a *good representation* of $\mathbf{L}_{t+t'}$ (e.g., the decoder $p_\theta(\mathbf{l}_{t+t'} \mid \mathbf{z}_{t,t'}, \mathbf{c}_{t,t'}, \mathbf{a}_{t+t'-1})$ is highly expressive and, ideally, injective in $\mathbf{z}_{t,t'}$ for a.e. $(\mathbf{c}_{t,t'}, \mathbf{a}_{t+t'-1})$), the pair $(\mathbf{r}_t, \mathbf{s}_{t,t'-1})$ approaches a sufficient statistic of $\bar{\mathbf{H}}_{t+t'-1}$ for predicting $\mathbf{L}_{t+t'}$. In VAEs the mapping is not exactly invertible, but training to maximize the conditional ELBO encourages $\mathbf{z}_{t,t'}$ to retain information about $\mathbf{L}_{t+t'}$ that is relevant for prediction; higher-fidelity decoders (e.g., with flows) make this approximation tighter.

**Lemma E.4** (Representation sufficiency implies context sufficiency). *Fix the embedding $\mathbf{r}_t$ and suppose Assumption E.1 holds. Assume that for Lebesgue-a.e. $(\mathbf{c}, \mathbf{a})$ the mapping $\mathbf{z} \mapsto p_\theta(\cdot \mid \mathbf{z}, \mathbf{c}, \mathbf{a})$ is injective as a map into $\mathcal{P}(\mathcal{L})$ (i.e., distinct $\mathbf{z}$ induce distinct conditional laws). Assume also that $\mathbf{s}_{t,t'-1}$ is a deterministic, measurable function of $(\mathbf{z}_{t,1:t'-1}, \mathbf{r}_t, \mathbf{a}_{t:t+t'-2}, 1{:}t'-1)$. Then for almost every $(\mathbf{r}_t, \mathbf{s}_{t,t'-1}, \mathbf{a}_{t+t'-1}, t')$ we have the conditional independence*

$$\mathbf{L}_{t+t'} \;\perp\!\!\!\perp\; \bar{\mathbf{H}}_{t+t'-1} \;\mid\; (\mathbf{r}_t, \mathbf{s}_{t,t'-1}, \mathbf{a}_{t+t'-1}, t'),$$

*i.e., $(\mathbf{r}_t, \mathbf{s}_{t,t'-1})$ is sufficient (with $\mathbf{a}_{t+t'-1}, t'$) for predicting $\mathbf{L}_{t+t'}$.*

*Proof.* Given $(\mathbf{r}_t, \mathbf{s}_{t,t'-1}, \mathbf{a}_{t+t'-1}, t')$, the next context $\mathbf{c}_{t,t'}$ is fixed and $\mathbf{z}_{t,t'} \sim p_0$ is independent of $\bar{\mathbf{H}}_{t+t'-1}$. The conditional density of $\mathbf{L}_{t+t'}$ factors as $p(\mathbf{l}_{t+t'} \mid \bar{\mathbf{h}}_{t+t'-1}, \mathbf{a}_{t+t'-1}) = \int p_\theta(\mathbf{l}_{t+t'} \mid \mathbf{z}, \mathbf{c}_{t,t'}, \mathbf{a}_{t+t'-1}) p_0(\mathbf{z}) \, d\mathbf{z}$ by Assumption E.1. Because $\mathbf{s}_{t,t'-1}$ is a deterministic function of past latents, any dependence on $\bar{\mathbf{H}}_{t+t'-1}$ enters only through $(\mathbf{r}_t, \mathbf{s}_{t,t'-1})$. Injectivity in $\mathbf{z}$ rules out aliasing of predictive distributions conditioned on $\mathbf{c}_{t,t'}$, so conditioning on $(\mathbf{r}_t, \mathbf{s}_{t,t'-1}, \mathbf{a}_{t+t'-1}, t')$ screens off the past. $\qquad\square$

**Notation.** When we write $\mathbf{c}_{t,t'}(\mathbf{z}_{t,1:t'-1})$ we suppress fixed arguments $(\mathbf{r}_t, \mathbf{a}_{t+t'-1}, t')$ and emphasize the indirect dependence via $\mathbf{s}_{t,t'-1}$; explicitly, $\mathbf{c}_{t,t'} = \kappa_\psi(\mathbf{r}_t, \mathbf{s}_{t,t'-1}(\mathbf{z}_{t,1:t'-1}), \mathbf{a}_{t+t'-1}, t')$.

**Theorem E.5** (Equivalence of latent and data-space g-computation). *Fix a time $t$, a horizon $\tau \geq 1$, a treatment plan $\bar{\mathbf{a}}_{t:t+\tau-1}$, and a history $\bar{\mathbf{h}}_t$. Under the standing causal assumptions and Assumption E.1, the interventional law identified by the sequential g-formula equals the law induced by latent rollout (Alg. 1):*

(i) **(Fixed-horizon marginal)** *For the last-step outcome,*

$$p^{\bar{\mathbf{a}}}(\mathbf{y}_{t+\tau} \mid \bar{\mathbf{h}}_t) \;=\; \int p_\theta\Big(\mathbf{y}_{t+\tau} \,\Big|\, \mathbf{z}_{t,\tau}, \mathbf{c}_{t,\tau}(\mathbf{z}_{t,1:\tau-1}), \mathbf{a}_{t+\tau-1}\Big) \prod_{t'=1}^{\tau} p_0(\mathbf{z}_{t,t'}) \, d\mathbf{z}_{t,t'}.$$

*Here $\mathbf{c}_{t,\tau}(\mathbf{z}_{t,1:\tau-1})$ is the deterministic context produced by the latent-state recursion driven by $\mathbf{z}_{t,1:\tau-1}$.*

*(ii) (**Full-path law**) For the joint path,*

$$p^{\bar{\mathbf{a}}}\big(\mathbf{y}_{t+1:t+\tau} \mid \bar{\mathbf{h}}_t\big) \;=\; \int \prod_{t'=1}^{\tau} p_\theta\Big(\mathbf{y}_{t+t'} \,\Big|\, \mathbf{z}_{t,t'},\, \mathbf{c}_{t,t'}(\mathbf{z}_{t,1:t'-1}),\, \mathbf{a}_{t+t'-1}\Big) \; \prod_{t'=1}^{\tau} p_0(\mathbf{z}_{t,t'}) \, d\mathbf{z}_{t,t'} \,,$$

*where, if desired, the covariates $\{\mathbf{x}_{t+t'}\}$ are integrated out.*

*Consequently, the Monte Carlo samples produced by Alg. 1 (with* `scope=last` *or* `all`*) are i.i.d. draws from the respective interventional laws.*

*Proof.* By identification, the last-step interventional density is

$$p^{\bar{\mathbf{a}}}\big(\mathbf{y}_{t+\tau} \mid \bar{\mathbf{h}}_t\big) = \int_{\mathbf{l}_{t+1:t+\tau-1}} \left[ \prod_{t'=1}^{\tau-1} p(\mathbf{l}_{t+t'} \mid \bar{\mathbf{h}}_{t+t'-1}, \mathbf{a}_{t+t'-1}) \right] p(\mathbf{y}_{t+\tau} \mid \bar{\mathbf{h}}_{t+\tau-1}, \mathbf{a}_{t+\tau-1}) \, d\mathbf{l}_{t+1:t+\tau-1}.$$

Insert Assumption E.1 at each step (including the last) to obtain

$$\int \left\{ \int_{\mathbf{l}_{t+1:t+\tau-1}} \prod_{t'=1}^{\tau-1} p_\theta(\mathbf{l}_{t+t'} \mid \mathbf{z}_{t,t'}, \mathbf{c}_{t,t'}, \mathbf{a}_{t+t'-1}) \, d\mathbf{l}_{t+1:t+\tau-1} \right\} p_\theta(\mathbf{y}_{t+\tau} \mid \mathbf{z}_{t,\tau}, \mathbf{c}_{t,\tau}, \mathbf{a}_{t+\tau-1}) \prod_{t'=1}^{\tau} p_0(\mathbf{z}_{t,t'}) \, d\mathbf{z}_{t,t'}.$$

Using Tonelli/Fubini (all integrands are nonnegative densities), we can swap integration order, and since $p_\theta(\mathbf{l}_{t+t'} \mid \mathbf{z}_{t,t'}, \mathbf{c}_{t,t'}, \mathbf{a}_{t+t'-1})$ is a normalized conditional density with $\mathbf{c}_{t,t'}$ independent of decoded $\mathbf{L}$, we have $\int p_\theta(\mathbf{l}_{t+t'} \mid \cdot) \, d\mathbf{l}_{t+t'} = 1$ for $t' = 1, \ldots, \tau - 1$ (the remaining integrals are over the latent path $\mathbf{z}_{t:t+\tau}$ and the terminal outcome, i.e., we are integrating out all intermediate variables). This yields the first result (i).

For the full-path law, repeat the same steps but keep the outcome components $\mathbf{y}_{t+t'}$ unintegrated (integrate only the covariates $\mathbf{x}_{t+t'}$ if desired). The product form in item (ii) follows because $\mathbf{c}_{t,t'}$ depends only on $(\mathbf{r}_t, \mathbf{a}_{t:t+t'-1}, \mathbf{z}_{t,1:t'-1})$, never on decoded $\mathbf{L}$. Finally, Alg. 1 draws $\{\mathbf{z}_{t,t'}\}$ i.i.d. from $p_0$ and applies the same deterministic maps and decoder conditional densities as above, so its outputs are i.i.d. from these laws. $\qquad\square$

**Corollary E.6** (Selective decoding (`scope`) is coherent). *Under the conditions of Thm. E.5, decoding only at $t+\tau$ (`scope=last`) returns i.i.d. samples from $p^{\bar{\mathbf{a}}}(\mathbf{y}_{t+\tau} \mid \bar{\mathbf{h}}_t)$. More generally, decoding at any subset $S \subseteq \{1, \ldots, \tau\}$ returns i.i.d. samples from the corresponding marginal over $\{\mathbf{Y}_{t+t'}\}_{t' \in S}$.*

**Sketch / intuition.** The sequential g-formula integrates over *future observations*. Assumption E.1 lets each one-step conditional be written as a mixture over a *latent noise* $\mathbf{z}_{t,t'}$ whose context depends only on $(\mathbf{r}_t, \mathbf{s}_{t,t'-1}, \mathbf{a}_{t+t'-1}, t')$. Because future contexts never use decoded $\mathbf{L}$, all intermediate integrals over $\mathbf{L}$ collapse to 1: only the latent-driven contexts matter. Thus, sampling latents and decoding where desired reproduces the same interventional law.

E.2  PROPAGATION ERROR: LATENT VS. DATA-SPACE G-COMPUTATION ROLLOUT

We now provide theoretical justification for the empirical superiority of latent rollouts over autoregressive rollouts. Let $\{K_s^\star\}_{s=t}^{t+\tau-1}$ denote the true one-step transition kernels and $\{K_s^{\mathrm{e}}\}_{s=t}^{t+\tau-1}$ the learned approximations. For each step $s$, define the local one-step approximation error

$$\varepsilon_s := \sup_{\bar{h}_s, a_s} \mathrm{TV}\Big(K_s^\star(\cdot \mid \bar{h}_s, a_s), K_s^{\mathrm{e}}(\cdot \mid \bar{h}_s, a_s)\Big),$$

where TV denotes total variation distance.

**Tail operators.** For $s \in \{t, \ldots, t+\tau-1\}$, let $T_{s+1:t+\tau}^{\bullet}$ denote the *tail operator* that maps a law on $L_{s+1}$ to the induced law of $Y_{t+\tau}$ obtained by propagating forward under rollout type $\bullet \in \{\mathrm{lat}, \mathrm{AR}\}$.

It is standard that pushing forward measures by a fixed Markov kernel is nonexpansive in total variation, hence

$$\mathrm{TV}\big(T_{s+1:t+\tau}^{\mathrm{lat}}[\mu], T_{s+1:t+\tau}^{\mathrm{lat}}[\nu]\big) \;\leq\; \mathrm{TV}(\mu, \nu). \tag{34}$$

The property is a standard result for Markov kernels, often referred to as the Data Processing Inequality. For autoregressive rollouts, the re-encoding step introduces sensitivity to the input measure. We assume:

**Assumption E.7** (Single-step AR operator Lipschitz property). For each index $j \in \{t+1, \ldots, t+\tau - 1\}$ define the *single-step AR tail operator*

$$\mathcal{T}^{\mathrm{AR}}_{j \to j+1} : \mathcal{P}(\mathcal{L}_j) \longrightarrow \mathcal{P}(\mathcal{L}_{j+1}),$$

which maps a law on $L_j$ (the predicted/decoded quantity at time $j$) to the induced law of the next-step quantity under the autoregressive re-encoding and decoding procedure. Assume there exist constants $\lambda_j \geq 0$ such that, for all probability measures $\mu, \nu$ on $\mathcal{L}_j$,

$$\mathrm{TV}\big(\mathcal{T}^{\mathrm{AR}}_{j \to j+1}[\mu], \ \mathcal{T}^{\mathrm{AR}}_{j \to j+1}[\nu]\big) \ \leq \ (1 + \lambda_j)\,\mathrm{TV}(\mu, \nu).$$

The assumption E.7 is justified because the autoregressive operator, as a finite composition of linear layers and Lipschitz-continuous activation functions, is itself guaranteed to be Lipschitz-continuous on any bounded domain.

**Lemma E.8** (Composition amplification). *Under Assumption E.7, the composed AR tail operator* $T^{\mathrm{AR}}_{s+1:t+\tau} = \mathcal{T}^{\mathrm{AR}}_{t+\tau-1 \to t+\tau} \circ \cdots \circ \mathcal{T}^{\mathrm{AR}}_{s+1 \to s+2}$ *satisfies, for any* $\mu, \nu$ *on* $\mathcal{L}_{s+1}$,

$$\mathrm{TV}\big(T^{\mathrm{AR}}_{s+1:t+\tau}[\mu], \ T^{\mathrm{AR}}_{s+1:t+\tau}[\nu]\big) \ \leq \ \prod_{j=s+1}^{t+\tau-1} (1 + \lambda_j)\,\mathrm{TV}(\mu, \nu).$$

*Proof.* Apply the single-step bound (S) iteratively. For brevity write $\mu_{s+1} = \mu$, $\nu_{s+1} = \nu$ and define $\mu_{j+1} = \mathcal{T}^{\mathrm{AR}}_{j \to j+1}[\mu_j]$, $\nu_{j+1} = \mathcal{T}^{\mathrm{AR}}_{j \to j+1}[\nu_j]$. Then

$$\mathrm{TV}(\mu_{j+1}, \nu_{j+1}) \leq (1 + \lambda_j)\,\mathrm{TV}(\mu_j, \nu_j).$$

Chaining these inequalities for $j = s+1, \ldots, t+\tau-1$ yields

$$\mathrm{TV}(\mu_{t+\tau}, \nu_{t+\tau}) \leq \Big( \prod_{j=s+1}^{t+\tau-1} (1 + \lambda_j) \Big) \mathrm{TV}(\mu_{s+1}, \nu_{s+1}),$$

which is the claimed bound. $\square$

*Remark* E.9 (A sufficient bound for $\lambda_j$). A convenient sufficient condition for Assumption E.7 is obtained by decomposing the single-step AR operator into (i) a *re-encoding map* $\Xi_j : \mathcal{P}(\mathcal{L}_j) \to \mathcal{C}_j$ that maps a predicted law on $L_j$ to a context in $\mathcal{C}_j$, and (ii) a decoder-induced kernel family $\{K_j^c\}_{c \in \mathcal{C}_j}$ that maps a context to a next-step kernel.

Concretely, suppose that for each $j$:

1. $\Xi_j$ is $L_{\Xi,j}$-Lipschitz in total variation, i.e.

$$\mathrm{TV}\big(\Xi_j[\mu], \Xi_j[\nu]\big) \leq L_{\Xi,j}\,\mathrm{TV}(\mu, \nu) \quad \text{for all } \mu, \nu \in \mathcal{P}(\mathcal{L}_j);$$

2. the decoder-induced kernel family is $L_{K,j}$-Lipschitz in context, i.e.

$$\sup_{c,c'} \mathrm{TV}\big(K_j^c, K_j^{c'}\big) \leq L_{K,j}\,\|c - c'\|.$$

Then for any two input measures $\mu, \nu$ on $\mathcal{L}_j$ we have

$$\mathrm{TV}\big(\mathcal{T}^{\mathrm{AR}}_{j \to j+1}[\mu], \mathcal{T}^{\mathrm{AR}}_{j \to j+1}[\nu]\big) \leq \sup_{c,c'} \mathrm{TV}\big(K_j^c, K_j^{c'}\big) \leq L_{K,j}\,\|\Xi_j[\mu] - \Xi_j[\nu]\| \leq L_{K,j} L_{\Xi,j}\,\mathrm{TV}(\mu, \nu).$$

Hence one may take

$$\lambda_j \leq L_{K,j}\,L_{\Xi,j},$$

and the product amplification in Proposition E.10 follows by composing these single-step bounds (cf. Lemma E.8).

**Main result.** Let $P^\star$ denote the true marginal law of $Y_{t+\tau}$, $P^{\text{lat}}$ the law induced by the latent rollout, and $P^{\text{AR}}$ the law induced by the autoregressive rollout. Then:

**Proposition E.10** (Propagation-error bound and dominance). *Let $t, \tau, a_{t:t+\tau-1}, \bar{h}_t$ be fixed, and let $P^\star$ denote the true interventional law of $Y_{t+\tau}$. Let $P^{\text{lat}}$ and $P^{\text{AR}}$ denote the learned laws produced by the latent and autoregressive/data-space rollouts, respectively, when both use the same per-step approximations $\{K_s^{\text{e}}\}_{s=t}^{t+\tau-1}$. Under Assumption E.7 (single-step AR operator Lipschitz property) we have*

$$\text{TV}(P^\star, P^{\text{lat}}) \leq \sum_{s=t}^{t+\tau-1} \varepsilon_s, \tag{35}$$

$$\text{TV}(P^\star, P^{\text{AR}}) \leq \sum_{s=t}^{t+\tau-1} \varepsilon_s \prod_{j=s+1}^{t+\tau-1} (1 + \lambda_j), \tag{36}$$

*where $\varepsilon_s := \sup_{\bar{h}_s, a_s} \text{TV}\big(K_s^\star(\cdot \mid \bar{h}_s, a_s), K_s^{\text{e}}(\cdot \mid \bar{h}_s, a_s)\big)$. In particular, if some $\lambda_j > 0$ then the bound equation 36 dominates equation 35, so the latent rollout attains a uniformly tighter (or equal) upper bound on the final-step discrepancy.*

*Proof.* For the latent rollout, a standard telescoping decomposition across steps combined with the non-expansive property of Markov kernels in Equation 34 yields the bound:

$$\text{TV}(P^\star, P^{\text{lat}}) \leq \sum_{s=t}^{t+\tau-1} \varepsilon_s.$$

For the autoregressive rollout, we define a sequence of hybrid distributions $P_s$ for $s = t, \ldots, t+\tau$, where $P_s$ is the law generated by using the true kernels $K^\star$ up to step $s-1$ and the learned kernels $K^e$ from step $s$ onwards. This gives $P_{t+\tau} = P^\star$ and $P_t = P^{\text{AR}}$.

By the triangle inequality, the total error is bounded by the sum of one-step differences:

$$\text{TV}(P^\star, P^{\text{AR}}) = \text{TV}(P_{t+\tau}, P_t) \leq \sum_{s=t}^{t+\tau-1} \text{TV}(P_{s+1}, P_s).$$

The difference between $P_{s+1}$ and $P_s$ arises only from the kernel used at step $s$. The error introduced at this step, at most $\varepsilon_s$, is then propagated forward by the autoregressive tail operator $T_{s+1:t+\tau}^{\text{AR}}$. Using the amplification bound from Lemma E.8, the contribution from step $s$ is:

$$\text{TV}(P_{s+1}, P_s) \leq \varepsilon_s \prod_{j=s+1}^{t+\tau-1} (1 + \lambda_j).$$

Summing these terms from $s = t$ to $t + \tau - 1$ yields the bound in Equation 36. Since each factor $(1 + \lambda_j) \geq 1$, the bound in Equation 36 is uniformly greater than or equal to the bound in Equation 35, completing the proof. $\square$

# F  DATASETS

## F.1  DETAILS ON EXPERIMENTS WITH SEMI-SYNTHETIC DATA (ORIGINAL SETTING)

Following Melnychuk et al. (2022), we build on MIMIC-EXTRACT (Wang et al., 2020)—a standardized preprocessing pipeline for MIMIC-III (Johnson et al., 2016)—which provides ICU time series aggregated at an hourly cadence. Missing values are imputed using forward and backward filling, and all continuous time-varying variables are standardized.

From this resource we retain 25 vital signs as time-varying covariates and three static covariates (gender, ethnicity, age). The complete feature list is provided in the accompanying code repository for reproducibility. Static covariates are one-hot encoded and later reused to modulate noise terms. In total, this yields a $d_v = 44$ dimensional covariate vector.

**High-level simulator design.** Following the basic idea of Schulam & Saria (2017), we first synthesize *untreated* outcome trajectories under endogenous and exogenous dependencies, and then apply treatments sequentially. We assume sparsity: each outcome depends on only a small subset of covariates and treatments; treatment assignment likewise depends on a limited subset of recent outcomes and covariates.

**Cohort selection.** We sample 1,000 patients whose ICU stays last at least 20 hours. Stays longer than 100 hours are clipped, so for patient $i$ we have $T^{(i)} \in [20, 100]$.

**Untreated outcomes.** For each patient $i$ and each outcome dimension $j = 1, \ldots, d_y$, we construct an untreated signal $Z_{j,t}^{(i)}$ by combining (i) a global trend, (ii) a patient-specific smooth component, (iii) an exogenous effect of current covariates, and (iv) noise:

$$Z_{j,t}^{(i)} = \underbrace{\alpha_j^S \, \text{B-spline}(t) + \alpha_j^g \, g_j^{(i)}(t)}_{\text{endogenous}} + \underbrace{\alpha_j^f \, f_j^Z\big(X_t^{(i)}\big)}_{\text{exogenous}} + \underbrace{\varepsilon_t}_{\text{noise}}, \qquad \varepsilon_t \sim \mathcal{N}(0, \, 0.005^2). \quad (37)$$

Here, $\text{B-spline}(t)$ is drawn from a mixture of three cubic splines (rapid decline, mild decline, stable) over the ICU stay; $g_j^{(i)}(t)$ is an independent Gaussian process with a Matérn kernel; and $f_j^Z(\cdot)$ is sampled via a random Fourier features (RFF) approximation to a Gaussian process (**?**), which avoids repeated Cholesky factorizations when sampling at many points in $\mathbb{R}^{d_x}$. The weights $\alpha_j^S, \alpha_j^g, \alpha_j^f$ control the relative contributions.

**Treatment assignment.** We then generate $d_a$ binary treatments $\{A_t^l\}_{l=1}^{d_a}$ sequentially, introducing confounding through (a) a function of current covariates and (b) recent outcome history. For treatment $l$ at time $t$ we define

$$p_{l,t}^A = \sigma\big(\gamma_l^A \, \overline{A}_{T_l}(\overline{Y}_{t-1}) + \gamma_l^X \, f_l^Y(X_t) + b_l\big), \quad (38)$$

$$A_t^l \sim \text{Bernoulli}\big(p_{l,t}^A\big), \quad (39)$$

where $\sigma(\cdot)$ is the logistic function; $\overline{A}_{T_l}(\overline{Y}_{t-1})$ denotes the average over a selected subset of the previous $T_l$ treated outcomes using the history $\overline{Y}_{t-1}$; $f_l^Y(\cdot)$ is sampled via an RFF GP (analogous to $f_j^Z$); and $\gamma_l^A, \gamma_l^X$ together with bias $b_l$ govern the strength of confounding.

**Treatment effects.** We set $Y_{j,1} = Z_{j,1}$ and endow each treatment $l$ with a long-lasting additive effect on outcome $j$ that is maximal immediately after administration and decays as an inverse square of elapsed time within a window of length $w_l$. Effects are scaled by the assignment probability $p_{l,i}^A$. When multiple treatments are active, we aggregate their contributions conservatively by taking the minimum at each elapsed time. Let $\varphi_l(\Delta) = (\Delta + 1)^{-2} \, \mathbf{1}\{0 \leq \Delta \leq w_l\}$. Then

$$E_j(t) = \sum_{i=1}^{t} \min_{l=1,\ldots,d_a} \left\{ \mathbf{1}\{A_i^l = 1\} \, p_{l,i}^A \, \beta_{lj} \, \varphi_l(t - i) \right\}, \quad (40)$$

where $\beta_{lj}$ is the maximum (immediate) effect size of treatment $l$ on outcome $j$ (either a constant or zero if treatment $l$ does not act on $j$).

**Observed outcomes.** The observed process adds treatment effects to the untreated signal:

$$Y_{j,t} = Z_{j,t} + E_j(t). \quad (41)$$

**Dataset construction and evaluation.** Unless stated otherwise, exact simulator hyperparameters are provided in the code. In our main setting we use $d_a = 3$ synthetic binary treatments and $d_y = 2$ outcomes. The 1,000 patients are split into train/validation/test using a $60\%/20\%/20\%$ split. For one-step-ahead evaluation we enumerate all $2^3 = 8$ counterfactuals. For multi-step rollouts with $\tau_{\max} = 10$, we sample 10 random treatment trajectories per patient and time step.

### F.1.1 VIOLATIONS OF THE POSITIVITY ASSUMPTION

We observed violations of the positivity (overlap) assumption in several instantiations of the semi-synthetic dataset generated with the parameters proposed by Melnychuk et al. (2022) and closely followed by several other works like (El Bouchattaoui et al., 2024; Wang et al., 2025). Concretely, for some random initializations almost all realized treatments are 0; for others, the distribution is heavily skewed toward 1. Inspecting the individual (per-arm) propensities $p_{\ell,t}^A \in (0,1)$ defined by Eq. 38 reveals that a large fraction of values are effectively *degenerate*. For one seed, for example, 95.6% of per-arm propensities are $< 1\%$, 76.5% are $< 0.1\%$, 42.9% are $< 0.01\%$, 15.8% are $< 0.001\%$, and 2.9% are $< 0.0001\%$; only 28 out of 101,031 valid treatment decisions have propensity $> 50\%$. For another seed, the mass concentrates near 1: 8.7% of propensities exceed 99% and 3.2% exceed 99.99%.

While the positivity assumption requires $0 < \Pr(A_t = a \mid H_t) < 1$ almost surely, in practice causal estimators become unstable when a substantial mass of propensities lies outside $[\epsilon, 1-\epsilon]$ for a small $\epsilon$ (e.g., $10^{-3}$). The extreme values above arise because the *logit* in Eq. 38 (a linear combination of recent outcomes and covariate features) can be very large in magnitude for some seeds, pushing $\sigma(\cdot)$ close to 0 or 1. In the next subsection we describe a minimally invasive modification that ensures overlap while preserving sequential confounding structure.

## F.2 OUR VERSION OF THE SEMI-SYNTHETIC DATASET

**Positivity via a monotone floor/ceiling.** To guarantee per-arm overlap we apply a monotone remapping to the *final* probability:

$$\tilde{p}_{\ell,t}^A = q + (1 - 2q)\,\sigma\big(b_\ell + z_{\ell,t}\big), \qquad q \in (0, 0.5), \tag{42}$$

which forces $\tilde{p}_{\ell,t}^A \in [q, 1-q]$. We use $q = 0.15$.

**Preserving confounding via logit normalization.** A naive floor alone avoids practical violations of positivity assumption but can still yield weak dependence on confounders if the *logit* distribution collapses (e.g., is almost always very large or very small). We therefore re-scale the *pre-bias* logit using train-set statistics so that the sigmoid operates on a stable range:

$$r_{\ell,t} = \gamma_\ell^Y\,\overline{Y}_{t-1} + \gamma_\ell^X\,f_X^{(\ell)}(X_t), \tag{43}$$

$$z_{\ell,t} = \frac{r_{\ell,t} - \mu_\ell}{\sigma_\ell + \varepsilon}, \qquad \varepsilon > 0, \tag{44}$$

where $(\mu_\ell, \sigma_\ell)$ are the mean and standard deviation of $r_{\ell,t}$ estimated *on the training split only*. The final propensity is then given by Eq. 42. This is an affine, monotone transformation of the original logit and therefore preserves the ordering of $r_{\ell,t}$ with respect to the history $H_t$.

**Two-pass generation to avoid leakage.** We use a standard two-pass protocol:

1. **Pass 1 (train only, original policy).** We run the generator once using Eq. 38 and record $r_{\ell,t}$ from Eq. 43 for every $(\ell, t)$ on the training split. We compute $(\mu_\ell, \sigma_\ell)$ per arm via an online (Welford) estimator. The trajectories from this pass are discarded; only $(\mu_\ell, \sigma_\ell)$ are kept.

2. **Pass 2 (train/val/test, overlap-calibrated).** We regenerate all splits from scratch. At each step we recompute $r_{\ell,t}$ from the *current* pass's history, apply the z-score in Eq. 44, then compute $\tilde{p}_{\ell,t}^A$ via Eq. 42 and sample treatments. Thus, sequential dependence on past outcomes/treatments remains intact; the first pass only provides $(\mu_\ell, \sigma_\ell)$, analogous to feature normalization. The magnitude of the utilized bias term is sufficiently small to not make logit magnitudes too large.

Pass 2 recomputes the logit from the *realized* past outcomes and treatments of the same pass; pass 1 probabilities are never used for sampling. Since z-scoring is affine and the final mapping is monotone, the confounding signal (how $H_t$ shifts treatment odds) is preserved, while the floor prevents near-degenerate propensities that destabilize estimation and calibration.

For one random instantiation of our new dataset, we have that the minimal probability of an individual treatment is 15.7%, and the maximum probability is 84.6%: apart from avoiding values too close to 0% or to 100%, the sigmoid does not get completely saturated, which would produce minimal or maximal values exactly in the floor. Apart from that, 86.7% of per-arm propensities are $> 25\%$, 14.3% are $> 50\%$, and 1.1% are $> 75\%$. For another seed, we have that the minimum per-arm propensity score is 15.8% and the highest one is 84.9%. Also, we have 89.1% of per-arm propensities $> 25\%$, 40.7% $> 50\%$, and 5.2% $> 75\%$.

### F.3 DETAILS ON EXPERIMENTS WITH REAL-WORLD DATA

In line with the semi-synthetic setup (App. F.1), we rely on MIMIC-EXTRACT (Wang et al., 2020), a standardized preprocessing pipeline for ICU time series (hourly resolution). Missing values are imputed using forward and backward filling, and all continuous time-varying variables are standardized. We use the same set of $d_x = 25$ vital signs and the same three static attributes (gender, ethnicity, age), one-hot encoded, yielding $d_v = 44$ static features. Both the time-varying covariates and static features are treated as potential confounders.

We consider $d_a = 2$ binary interventions: vasopressors and mechanical ventilation. The factual outcome is diastolic blood pressure ($d_y = 1$). Clinically, both interventions can increase or decrease blood pressure depending on context, motivating counterfactual trajectory analysis under alternative treatment choices.

**Cohort and splits.** We select 5,000 patients with ICU stays of at least 30 hours; stays are truncated at 60 hours. The cohort is divided into train/validation/test sets with a $70\%/15\%/15\%$ split.

## G BASELINES

### G.1 CAUSAL TRANSFORMER

#### G.1.1 BASE CAUSAL TRANSFORMER

We implement the Causal Transformer (CT) of Melnychuk et al. (2022) as a strong baseline for estimating

$$\mathbb{E}\big[ Y_{t+\tau}\big[\bar{a}_{t:t+\tau-1}\big] \,\big|\, \bar{H}_t\big] \tag{45}$$

under a treatment plan $\bar{a}_{t:t+\tau-1}$. To avoid duplication, we reuse the multi-input transformer encoder in App. B and highlight only CT-specific pieces (projection inputs, balanced-representation learning, and stabilizers).

**Inputs and autoregressive conditioning.** CT consumes three factual streams up to anchor time $t$: covariates $\bar{X}_t$, outcomes $\bar{Y}_t$, and *left-shifted* treatments $\bar{A}_{t-1}$, plus static covariates $V$. For a projection horizon $\tau$, CT concatenates the factual histories with the (non-random) *future* intervention sequence on the treatment stream and with *autoregressively fed predictions* on the outcome stream:

$$\bar{A}_{t-1} \,\|\, \bar{a}_{t:t+\tau-1}, \tag{46}$$

$$\bar{Y}_t \,\|\, \hat{\bar{Y}}_{t+1:t+\tau-1}. \tag{47}$$

Teacher forcing is used during training for multi-step prediction; at evaluation time, the model feeds back its own predictions autoregressively. Static covariates $V$ are injected in all subnetworks.

**Architecture (encoder blocks, cross-attention, pooling).** CT follows the multi-input transformer pattern in App. B: masked self-attention per stream, cross-attention between streams, position-wise feed-forward layers, and LN+residual connections, with trainable relative positional encodings and attentional dropout. After the last block, the three stream states are *averaged* and passed through a Linear+ELU to obtain a balanced representation $\mathbf{\Phi}_t \in \mathbb{R}^{d_r}$:

$$\mathbf{\Phi}_t \;=\; \mathrm{ELU}\big(\mathrm{Linear}\big(\tfrac{1}{3}\big(\mathbf{A}_{t-1}^B + \mathbf{X}_t^B + \mathbf{Y}_t^B\big)\big)\big). \tag{48}$$

(Implementation note: CT omits the final output projection after concatenating attention heads to reduce overfitting.)

**Balanced-representation training.** CT trains $\boldsymbol{\Phi}_t$ to be (i) predictive of the one-step factual outcome while (ii) *non-predictive* of the current treatment with a Counterfactual Domain Confusion (CDC) loss. Two light heads are attached to $\boldsymbol{\Phi}_t$: an outcome head $G_Y$ and a treatment classifier $G_A$. Let $d_a$ be the number of treatment categories. The losses are

$$\mathcal{L}_{GA}(\theta_A, \theta_R) \; = \; - \sum_{j=1}^{d_a} \mathbf{1}\{A_t = a_j\} \, \log\big(G_A(\boldsymbol{\Phi}_t(\theta_R); \theta_A)_j\big), \tag{49}$$

$$\mathcal{L}_{\mathrm{conf}}(\theta_A, \theta_R) \; = \; - \sum_{j=1}^{d_a} \frac{1}{d_a} \, \log\big(G_A(\boldsymbol{\Phi}_t(\theta_R); \theta_A)_j\big), \tag{50}$$

and the alternating min–min scheme is

$$(\hat{\theta}_Y, \hat{\theta}_R) \; = \; \arg\min_{\theta_Y, \theta_R} \; \mathcal{L}_{GY}(\theta_Y, \theta_R) \; + \; \alpha \, \mathcal{L}_{\mathrm{conf}}(\hat{\theta}_A, \theta_R), \tag{51}$$

$$\hat{\theta}_A \; = \; \arg\min_{\theta_A} \; \alpha \, \mathcal{L}_{GA}(\theta_A, \hat{\theta}_R), \tag{52}$$

with $\alpha > 0$ the domain-confusion weight and $\mathcal{L}_{GY}$ defined by the chosen outcome head (see below).

**Training stabilizers and augmentation.** We follow CT practice: (i) an *exponential moving average* (EMA) of parameters across trainable modules; (ii) *attentional dropout*; and (iii) mini-batch augmentation that duplicates samples and randomly *masks* the last $t_s$ covariate steps in the duplicate (to reflect unavailable future covariates for $\tau \ge 2$).

**Point-estimator CT (original).** The original CT uses a point head $G_Y$ with squared error:

$$\mathcal{L}_{GY}^{(\mathrm{point})}(\theta_Y, \theta_R) \; = \; \big\| \, \mathbf{y}_{t+1} - G_Y\big(\boldsymbol{\Phi}_t(\theta_R), \mathbf{a}_t; \theta_Y\big) \, \big\|_2^2. \tag{53}$$

### G.1.2 DISTRIBUTIONAL VARIANTS

We additionally evaluate two *distributional* adaptations of CT that replace the outcome head/loss, keeping architecture and CDC unchanged.

**CT–Gaussian head (heteroscedastic NLL).** The Gaussian head predicts per-dimension mean and variance $(\hat{\boldsymbol{\mu}}_{t+1}, \hat{\boldsymbol{\sigma}}_{t+1}^2) = G_Y^{\mathcal{N}}(\boldsymbol{\Phi}_t, \mathbf{a}_t)$, and minimizes the Gaussian negative log-likelihood (diagonal covariance):

$$\mathcal{L}_{GY}^{\mathcal{N}}(\theta_Y, \theta_R) \; = \; \tfrac{1}{2} \left\| \frac{\mathbf{y}_{t+1} - \hat{\boldsymbol{\mu}}_{t+1}}{\hat{\boldsymbol{\sigma}}_{t+1}} \right\|_2^2 \; + \; \tfrac{1}{2} \, \mathbf{1}^\top \log \hat{\boldsymbol{\sigma}}_{t+1}^2. \tag{54}$$

**CT–CRPS / random-quantile head.** The random-quantile head predicts outcome quantiles given $\alpha \in (0,1)^{d_y}$. Let $\hat{\mathrm{q}}_{\alpha,j} = d_j(\boldsymbol{\Phi}_t, \mathbf{a}_t, \alpha_j)$ denote the predicted $\alpha_j$-quantile of $Y_{t+1,j}$ for branch $j$. Drawing $K$ i.i.d. vectors $\{\boldsymbol{\alpha}^{(k)}\}_{k=1}^K$ with entries $\alpha_j^{(k)} \sim \mathrm{Unif}(0,1)$, we use the Monte Carlo CRPS objective:

$$\mathcal{L}_{GY}^{\mathrm{CRPS}}(\theta_Y, \theta_R) \; = \; \sum_{j=1}^{d_y} \frac{1}{K} \sum_{k=1}^{K} \rho_{\alpha_j^{(k)}}\Big(y_{t+1,j} - \hat{\mathrm{q}}_{\alpha_j^{(k)}, j}\Big), \tag{55}$$

with the pinball loss

$$\rho_\alpha(u) \; = \; (\alpha - \mathbf{1}\{u < 0\}) \, u. \tag{56}$$

This is the same random-quantile reconstruction used for $\mathbf{Y}$ in G-Latent, providing a proper scoring rule (CRPS) and capturing predictive uncertainty through the quantile function.

## G.2   G-NET

G-Net implements g-computation in two steps. First, it estimates the conditional *expectations* of within-time components of $\mathbf{L}_{t+1} = (\mathbf{Y}_{t+1}, \mathbf{X}_{t+1})$ given history and action. Concretely, for an ordered decomposition $\mathbf{L}_{t+1}^{(0)}, \ldots, \mathbf{L}_{t+1}^{(p-1)}$, we learn

$$\mathbb{E}\big[\mathbf{L}_{t+1}^{(j)} \mid \bar{\mathbf{H}}_t, \mathbf{A}_t, \mathbf{L}_{t+1}^{(0:j-1)}\big] \tag{57}$$

with a two-layer LSTM. Samples from the corresponding conditionals are obtained by adding residuals drawn from an empirical error distribution built on a $10\%$ holdout split (residual bootstrap). Training uses teacher forcing and an MSE loss.

Second, counterfactual trajectories under a treatment plan $\bar{\mathbf{a}}_{t:t+\tau-1}$ are generated by Monte Carlo, rolling the learned conditionals forward across steps.

We follow the same architecture class reported alongside CT: one–two layered LSTMs, a linear representation layer, and a small feed-forward head on top. At evaluation, we simulate under $\bar{\mathbf{a}}$ with start-of-interval indexing (action $\mathbf{A}_s$ precedes $(\mathbf{Y}_{s+1}, \mathbf{X}_{s+1})$), using the residual-bootstrap sampler.

## G.3   TRANSFORMER G-NET

Transformer G-Net follows the same two-step pipeline but replaces the recurrent modules with the multi-input transformer encoder of App. B. The transformer encodes the factual history before action into a fused state $\mathbf{r}_t$ (respecting start-of-interval indexing). For an ordered within-time decomposition $\mathbf{L}_{t+1}^{(0)}, \ldots, \mathbf{L}_{t+1}^{(p-1)}$, each conditional expectation is predicted by a small MLP head conditioned on $\mathbf{r}_t$, $\mathbf{A}_t$, and previously generated groups; training uses teacher forcing and an MSE objective. During rollout we inject residual noise via the same $10\%$ holdout bootstrap and obtain the *distribution* at horizon $t+\tau$ as the empirical measure over $M$ Monte Carlo trajectories (again $M{=}50$), without any balanced-representation objective.

## H   METRICS

Our model outputs i.i.d. Monte Carlo (MC) samples $\{\mathbf{y}_{t+s}^{(m,i)}\}_{m=1}^{M}$ from the interventional law $p^{\bar{\mathbf{a}}}\big(\mathbf{y}_{t+s} \mid \bar{\mathbf{h}}_t^{(i)}\big)$ at each relative step $s \in \{1, \ldots, \tau\}$, given history $\bar{\mathbf{h}}_t^{(i)}$ and a treatment plan $\bar{\mathbf{a}}_{t:t+\tau-1}$. All metrics are computed *per step* and averaged over $n$ test patients; when relevant we also report a trajectory-level score aggregating all steps.

**RMSE of the predictive mean (point accuracy).**   Let the per-step predictive mean for patient $i$ be

$$\hat{\boldsymbol{\mu}}_{t+s}^{(i)} = \frac{1}{M} \sum_{m=1}^{M} \mathbf{y}_{t+s}^{(m,i)}. \tag{58}$$

For a $d_y$-dimensional outcome we report

$$\mathrm{RMSE}_s = \sqrt{\frac{1}{n\,d_y} \sum_{i=1}^{n} \big\| \hat{\boldsymbol{\mu}}_{t+s}^{(i)} - \mathbf{y}_{t+s}^{(i)} \big\|_2^2}, \tag{59}$$

which summarizes point accuracy of the posterior mean implied by the predictive distribution (lower is better).

**KDE log-likelihood (density fit).**   We estimate the patient-specific predictive density at relative step $t'$ with an isotropic Gaussian KDE using a single global bandwidth $h > 0$:

$$\hat{f}_{t'}^{(i)}(\mathbf{y}) = \frac{1}{M} \sum_{m=1}^{M} \mathcal{N}\big(\mathbf{y}; \mathbf{y}_{t+t'}^{(m,i)}, h^2 \mathbf{I}_{d_y}\big), \tag{60}$$

where $\mathbf{I}_{d_y}$ is the $d_y \times d_y$ identity matrix and $h$ is fixed across all $t'$ and all patients. The metric is the average log-likelihood:

$$\mathrm{KDE\text{-}LL}_{t'} = \frac{1}{n} \sum_{i=1}^{n} \log \hat{f}_{t'}^{(i)}\big(\mathbf{y}_{t+t'}^{(i)}\big). \tag{61}$$

**Energy score (multivariate proper scoring rule).** For $d_y \geq 1$, the energy score (ES) for a predictive distribution $F_s$ and realization $\mathbf{y}_{t+s}$ is

$$\mathrm{ES}_s(F_s, \mathbf{y}_{t+s}) \;=\; \mathbb{E}\big[\|\mathbf{X} - \mathbf{y}_{t+s}\|_2\big] - \tfrac{1}{2}\,\mathbb{E}\big[\|\mathbf{X} - \mathbf{X}'\|_2\big], \tag{62}$$

with $\mathbf{X}, \mathbf{X}' \sim F_s$ i.i.d. Using MC samples, we estimate

$$\widehat{\mathrm{ES}}_s \;=\; \frac{1}{n}\sum_{i=1}^{n}\left\{ \frac{1}{M}\sum_{m=1}^{M} \big\|\mathbf{y}_{t+s}^{(m,i)} - \mathbf{y}_{t+s}^{(i)}\big\|_2 - \frac{1}{2M(M-1)}\sum_{m\neq m'}\big\|\mathbf{y}_{t+s}^{(m,i)} - \mathbf{y}_{t+s}^{(m',i)}\big\|_2 \right\}, \tag{63}$$

which is strictly proper and *sensitive to cross-dimensional dependence* (lower is better). In the univariate case ($d_y=1$) ES equals the continuous ranked probability score (CRPS).

**Global (pathwise) energy score (temporal coherence).** To assess coherence across *all* output dimensions and steps, we compute ES on the concatenated outcome vector $\tilde{\mathbf{y}}_{t+1:t+\tau} \in \mathbb{R}^{\tau d_y}$, where $\tilde{\mathbf{y}}_{t+1:t+\tau}^{(m,i)} := [\mathbf{y}_{t+1}^{(m,i)}; \ldots; \mathbf{y}_{t+\tau}^{(m,i)}]$ and $\tilde{\mathbf{y}}_{t+1:t+\tau}^{(i)} := [\mathbf{y}_{t+1}^{(i)}; \ldots; \mathbf{y}_{t+\tau}^{(i)}]$:

$$\mathrm{GES} \;=\; \frac{1}{n}\sum_{i=1}^{n}\left\{ \frac{1}{M}\sum_{m=1}^{M} \big\|\tilde{\mathbf{y}}_{t+1:t+\tau}^{(m,i)} - \tilde{\mathbf{y}}_{t+1:t+\tau}^{(i)}\big\|_2 - \frac{1}{2M(M-1)}\sum_{m\neq m'}\big\|\tilde{\mathbf{y}}_{t+1:t+\tau}^{(m,i)} - \tilde{\mathbf{y}}_{t+1:t+\tau}^{(m',i)}\big\|_2 \right\}. \tag{64}$$

This whole-trajectory ES rewards correct temporal correlations and cross-step consistency of the joint predictive law (lower is better).

**Quantile coverage (calibration).** For quantile levels $\mathcal{Q} = \{0.1, 0.2, \ldots, 0.9\}$, we compare each realized outcome component to the MC-estimated predictive quantile of that component. Let $(\cdot)_j$ denote the $j$-th component. Define

$$\hat{Q}_{s,j}^{(i)}(q) \;:=\; \mathrm{quantile}_q\big(\{\,(\mathbf{y}_{t+s}^{(m,i)})_j\,\}_{m=1}^{M}\big). \tag{65}$$

Per step and per dimension, the empirical $q$-coverage is

$$\widehat{\mathrm{Cov}}_{s,j}(q) \;=\; \frac{1}{n}\sum_{i=1}^{n}\mathbb{I}\Big\{\,(\mathbf{y}_{t+s}^{(i)})_j \leq \hat{Q}_{s,j}^{(i)}(q)\,\Big\}, \tag{66}$$

which should match the nominal level $q$ for a calibrated model (higher/lower than $q$ indicates over-/under-coverage). We use "$\leq$" to break ties; quantiles are computed from MC samples per $(i,s,j)$ with a fixed interpolation rule.

We also use aggregations across steps:

$$\widehat{\mathrm{Cov}}_{j}^{\mathrm{steps}}(q) = \frac{1}{n\,\tau}\sum_{s=1}^{\tau}\sum_{i=1}^{n}\mathbb{I}\Big\{\,(\mathbf{y}_{t+s}^{(i)})_j \leq \hat{Q}_{s,j}^{(i)}(q)\,\Big\}, \tag{67}$$

**Calibration summary (MAE).** A scalar summary is the mean absolute calibration error, averaged over quantiles, dimensions, and steps:

$$\mathrm{CalMAE} \;=\; \frac{1}{|\mathcal{Q}|\,d_y\,\tau}\sum_{q\in\mathcal{Q}}\sum_{j=1}^{d_y}\sum_{s=1}^{\tau}\big|\,\widehat{\mathrm{Cov}}_{s,j}(q) - q\,\big|. \tag{68}$$

Lower is better; per-dimension or per-step variants follow by omitting the corresponding averages.

# I HYPERPARAMETERS

## I.1 MULTI-INPUT TRANSFORMER

For better comparability, we used the same multi-input transformer hyperparameters for all the models that use transformer processing (CT, CT-CRPS, Transformer G-Net, CT-Gaussian and G-Latent). We used the same hyperparameters as Melnychuk et al. (2022), as additional tuning on our specific models did not provide significant improvements. We list these hyperparameters in table 4, and define them next:

Table 4: Architectural hyperparameters for the multi-input transformer.

| Hyperparameter | Semi-synthetic | Real-world |
|---|---|---|
| Transformer units | 24 | 24 |
| Representation size | 44 | 22 |
| Fully connected hidden units | 22 | 22 |
| Dropout rate | 0.1 | 0.2 |
| Transformer blocks | 1 | 2 |
| Attention heads | 2 | 3 |
| Max relative position | 20 | 30 |

- Transformer units: model width per stream (token and attention projection size; per-head dimension roughly Transformer units divided by Attention heads).

- Representation size: fused history embedding dimension used downstream.

- Fully connected hidden units: inner width of the position-wise feed-forward sublayer.

- Dropout rate: probability used after linear layers in attention and feed-forward sublayers.

- Transformer blocks: number of stacked encoder blocks.

- Attention heads: number of heads in multi-head attention.

- Max relative position: clipping radius for relative positional encodings shared across blocks and streams.

## I.2 CAUSAL TRANSFORMER

We report the specific training hyperparameters of CT in table 5.

Table 5: Training hyperparameters for the multi-input transformer.

| Hyperparameter | Semi-synthetic | Real-world |
|---|---|---|
| Learning rate | 0.01 | 0.0001 |
| Batch size | 64 | 64 |
| Max epochs | 400 | 300 |

For the distributional versions of CT, we used the same hyperparameters. For CT-CRPS, we used a number of $\alpha$ quantile MC samples $K = 5$ for both semi-synthetic and real-world dataset. This value is the same we used for G-Latent.

## I.3 G-NET

For G-Net, we used the hyperparameters configuration from the implementation in Melnychuk et al. (2022). We report it in table 6, and define them as:

- Recurrent layers: number of stacked recurrent layers.

- Sequence hidden units: hidden size per recurrent layer.

- Fully connected hidden units: width of the feed-forward head.

- Dropout rate: dropout probability in recurrent/feed-forward parts.

- Representation size: size of the intermediate representation.

- Learning rate: optimizer step size.

- Batch size: examples per minibatch.

- Max epochs: maximum training epochs.

## I.4 G-LATENT

As previously mentioned, the multi-input transformer we used in G-Latent has the hyperparameters shared with other baselines and defined in I.1. As for the rest of hyperparameters, after an optimization process based on factual validation datasets, we selected the ones shown in table 7. We defined next:

Table 6: Architectural and training hyperparameters for G-Net.

| Hyperparameter | Semi-synthetic | Real-world |
|---|---|---|
| Recurrent layers | 1 | 2 |
| Sequence hidden units | 148 | 144 |
| Fully connected hidden units | 74 | 72 |
| Dropout rate | 0.1 | 0.1 |
| Representation size | 74 | 72 |
| Learning rate | 0.01 | 0.001 |
| Batch size | 256 | 256 |
| Max epochs | 200 | 200 |

- Learning rate: optimizer step size.

- KL weight: coefficient on the KL divergence term in the ELBO.

- Latent dimension: dimensionality of the VAE latent variable $z$.

- Auxiliar loss weight ($\lambda_{\text{aux}}$): weight on the auxiliary one-step prediction loss.

- Max epochs: maximum number of training epochs.

- Reconstruction weights (outcome, covariates): multipliers for outcome and covariate reconstruction terms. The fact that, in both datasets, covariates have much higher coefficients than outcomes makes the model give balanced weight to both of them. Weights are selected in such a way that the sum of products of each weight with each dimensionality gives one.

- MC $\alpha$ samples ($K$): number of quantile levels sampled per step for the CRPS/quantile head.

- Batch size: number of examples per minibatch.

- Context dimension: size of the context vector fed to the VAE.

- Encoder hidden sizes: layer widths of the encoder MLP $q_\phi(z \mid x, c)$.

- Decoder hidden sizes: layer widths of the shared decoder trunk $T_\theta$.

- Quantile-branch hidden sizes: layer widths in the per-outcome, $\alpha$-aware branches.

- Shared decoder layers: count of initial decoder layers shared by the $\alpha$-aware and mean/log-variance branches.

- Warm-up epochs (auxiliar loss only): epochs optimizing only the auxiliary loss before enabling VAE terms.

- GRU hidden size: hidden width of the temporal GRU cell used in latent rollouts.

Table 7: Architectural and training hyperparameters for the RNN+Conditional VAE (G-Latent) stack.

| Hyperparameter | Semi-synthetic | Real-world |
|---|---|---|
| Learning rate | 0.0001 | 0.0003 |
| KL weight | 1.0 | 1.0 |
| Latent dimension | 6 | 6 |
| Auxiliar loss weight ($\lambda_{aux}$) | 0.1 | 0.1 |
| Max epochs | 70 | 110 |
| Reconstruction weights (outcome, covariates) | [6.67, 0.32] | [18.0, 0.32] |
| MC $\alpha$ samples ($K$) | 5 | 5 |
| Batch size | 8 | 8 |
| Context dimension | 256 | 256 |
| Encoder hidden sizes | [256, 256, 256, 256, 256] | [256, 256, 256, 256, 256] |
| Decoder hidden sizes | [256, 256, 256, 256, 256] | [256, 256, 256, 256, 256] |
| Quantile-branch hidden sizes | [64, 64] | [128, 128] |
| Shared decoder layers | 3 | 3 |
| Warm-up epochs (auxiliar loss only) | 20 | 30 |
| GRU hidden size | 64 | 64 |

## J ADDITIONAL RESULTS

### J.1 SEMI-SYNTHETIC DATASET (OUR NEW VERSION)

In table 8, we show the Energy Scores for our new modified semi-synthetic dataset. In tables 9, 10 and 11 we show the KDE-LL for bandwidths 0.2, 0.3 and 0.4, respectively. In table 12, we show the RMSE metrics. Finally, in tables 13 and 14 we show the empirical quantile coverage for all the steps (1 to 6 in the first table, 7 to 11 in the second one, plus across step coverage), the dimensions, and several quantiles from 0.1 to 0.9, The bolded results are the ones closest to the expected coverage percentage, i.e., for quantile 0.1, 10%, for quantile 0.2, 20%, etc.

Table 8: Energy Score per step $t'$ on semi-synthetic dataset (corrected benchmark). Rightmost column reports the Global Energy Score across steps. Best per column in **bold**.

| Model | $t'=1$ | $t'=2$ | $t'=3$ | $t'=4$ | $t'=5$ | $t'=6$ | $t'=7$ | $t'=8$ | $t'=9$ | $t'=10$ | $t'=11$ | Global |
|---|---|---|---|---|---|---|---|---|---|---|---|---|
| G-Net | $0.17 \pm 0.00$ | $0.30 \pm 0.03$ | $0.39 \pm 0.04$ | $0.45 \pm 0.04$ | $0.51 \pm 0.05$ | $0.55 \pm 0.06$ | $0.59 \pm 0.06$ | $0.63 \pm 0.07$ | $0.65 \pm 0.07$ | $0.68 \pm 0.07$ | $0.70 \pm 0.08$ | $1.85 \pm 0.20$ |
| Transformer G-Net | $0.37 \pm 0.04$ | $0.34 \pm 0.04$ | $0.40 \pm 0.05$ | $0.46 \pm 0.06$ | $0.50 \pm 0.07$ | $0.53 \pm 0.08$ | $0.56 \pm 0.10$ | $0.58 \pm 0.11$ | $0.60 \pm 0.12$ | $0.62 \pm 0.13$ | $0.64 \pm 0.14$ | $1.71 \pm 0.11$ |
| CT-CRPS | $\mathbf{0.09 \pm 0.01}$ | $0.26 \pm 0.06$ | $0.32 \pm 0.07$ | $0.37 \pm 0.07$ | $0.41 \pm 0.07$ | $0.45 \pm 0.08$ | $0.48 \pm 0.07$ | $0.50 \pm 0.07$ | $0.53 \pm 0.07$ | $0.55 \pm 0.07$ | $0.57 \pm 0.07$ | $1.52 \pm 0.23$ |
| CT-Gaussian | $\mathbf{0.09 \pm 0.01}$ | $0.25 \pm 0.06$ | $0.30 \pm 0.07$ | $0.34 \pm 0.08$ | $0.37 \pm 0.08$ | $0.40 \pm 0.09$ | $0.42 \pm 0.09$ | $0.44 \pm 0.09$ | $0.46 \pm 0.09$ | $0.48 \pm 0.09$ | $0.49 \pm 0.09$ | $1.35 \pm 0.29$ |
| D.S. G-VAE (Gaussian) | $0.28 \pm 0.01$ | $0.40 \pm 0.02$ | $0.49 \pm 0.04$ | $0.54 \pm 0.05$ | $0.58 \pm 0.06$ | $0.60 \pm 0.06$ | $0.62 \pm 0.07$ | $0.64 \pm 0.07$ | $0.65 \pm 0.07$ | $0.66 \pm 0.07$ | $0.67 \pm 0.07$ | $2.01 \pm 0.20$ |
| D.S. G-VAE (CRPS) | $0.13 \pm 0.00$ | $\mathbf{0.23 \pm 0.04}$ | $\mathbf{0.28 \pm 0.05}$ | $\mathbf{0.32 \pm 0.06}$ | $\mathbf{0.35 \pm 0.06}$ | $0.38 \pm 0.06$ | $0.40 \pm 0.07$ | $0.42 \pm 0.07$ | $0.44 \pm 0.06$ | $0.45 \pm 0.06$ | $0.47 \pm 0.06$ | $1.28 \pm 0.21$ |
| G-Latent (Gaussian) | $0.31 \pm 0.02$ | $0.35 \pm 0.03$ | $0.38 \pm 0.04$ | $0.40 \pm 0.05$ | $0.42 \pm 0.05$ | $0.44 \pm 0.06$ | $0.45 \pm 0.06$ | $0.46 \pm 0.06$ | $0.47 \pm 0.06$ | $0.48 \pm 0.06$ | $0.48 \pm 0.06$ | $1.51 \pm 0.18$ |
| G-Latent (CRPS) | $0.19 \pm 0.02$ | $0.25 \pm 0.04$ | $0.29 \pm 0.05$ | $0.33 \pm 0.06$ | $\mathbf{0.35 \pm 0.06}$ | $\mathbf{0.37 \pm 0.07}$ | $\mathbf{0.39 \pm 0.07}$ | $\mathbf{0.40 \pm 0.07}$ | $\mathbf{0.42 \pm 0.07}$ | $\mathbf{0.42 \pm 0.08}$ | $\mathbf{0.43 \pm 0.08}$ | $1.25 \pm 0.23$ |

Table 9: KDE Loglikelihood per step $t'$ on semi-synthetic dataset with bandwidth 0.2. Best per column in **bold**.

| Model | $t'=1$ | $t'=2$ | $t'=3$ | $t'=4$ | $t'=5$ | $t'=6$ | $t'=7$ | $t'=8$ | $t'=9$ | $t'=10$ | $t'=11$ |
|---|---|---|---|---|---|---|---|---|---|---|---|
| G-Net | $0.30 \pm 0.05$ | $-0.85 \pm 0.20$ | $-1.48 \pm 0.25$ | $-1.91 \pm 0.28$ | $-2.21 \pm 0.31$ | $-2.47 \pm 0.33$ | $-2.70 \pm 0.36$ | $-2.89 \pm 0.39$ | $-3.04 \pm 0.42$ | $-3.17 \pm 0.45$ | $-3.29 \pm 0.48$ |
| Transformer G-Net | $-1.34 \pm 0.20$ | $-1.07 \pm 0.24$ | $-1.52 \pm 0.27$ | $-1.86 \pm 0.38$ | $-2.12 \pm 0.49$ | $-2.33 \pm 0.59$ | $-2.52 \pm 0.69$ | $-2.70 \pm 0.79$ | $-2.86 \pm 0.90$ | $-3.00 \pm 0.98$ | $-3.14 \pm 1.06$ |
| CT-CRPS | $0.99 \pm 0.07$ | $-0.88 \pm 0.71$ | $-1.56 \pm 0.78$ | $-2.16 \pm 0.82$ | $-2.68 \pm 0.79$ | $-3.17 \pm 0.81$ | $-3.63 \pm 0.81$ | $-4.06 \pm 0.79$ | $-4.43 \pm 0.81$ | $-4.78 \pm 0.83$ | $-5.08 \pm 0.86$ |
| CT-Gaussian | $\mathbf{1.00 \pm 0.05}$ | $-0.40 \pm 0.49$ | $-0.73 \pm 0.54$ | $-0.99 \pm 0.57$ | $-1.20 \pm 0.57$ | $-1.39 \pm 0.57$ | $-1.56 \pm 0.57$ | $-1.72 \pm 0.57$ | $-1.86 \pm 0.57$ | $-2.01 \pm 0.57$ | $-2.14 \pm 0.57$ |
| D.S. G-VAE (Gaussian) | $-1.21 \pm 0.08$ | $-1.87 \pm 0.10$ | $-2.25 \pm 0.14$ | $-2.45 \pm 0.17$ | $-2.57 \pm 0.18$ | $-2.65 \pm 0.19$ | $-2.70 \pm 0.20$ | $-2.74 \pm 0.20$ | $-2.76 \pm 0.20$ | $-2.79 \pm 0.20$ | $-2.80 \pm 0.20$ |
| D.S. G-VAE (CRPS) | $0.45 \pm 0.07$ | $\mathbf{-0.32 \pm 0.33}$ | $\mathbf{-0.66 \pm 0.37}$ | $\mathbf{-0.89 \pm 0.39}$ | $\mathbf{-1.06 \pm 0.39}$ | $\mathbf{-1.22 \pm 0.40}$ | $\mathbf{-1.34 \pm 0.39}$ | $-1.45 \pm 0.38$ | $-1.54 \pm 0.36$ | $-1.62 \pm 0.33$ | $-1.69 \pm 0.31$ |
| G-Latent (Gaussian) | $-1.36 \pm 0.11$ | $-1.52 \pm 0.16$ | $-1.62 \pm 0.18$ | $-1.69 \pm 0.20$ | $-1.74 \pm 0.21$ | $-1.79 \pm 0.22$ | $-1.83 \pm 0.23$ | $-1.86 \pm 0.23$ | $-1.88 \pm 0.23$ | $-1.90 \pm 0.23$ | $-1.92 \pm 0.23$ |
| G-Latent (CRPS) | $-0.01 \pm 0.17$ | $-0.50 \pm 0.30$ | $-0.78 \pm 0.35$ | $-0.98 \pm 0.39$ | $-1.12 \pm 0.41$ | $-1.24 \pm 0.42$ | $\mathbf{-1.34 \pm 0.44}$ | $\mathbf{-1.42 \pm 0.44}$ | $\mathbf{-1.48 \pm 0.44}$ | $\mathbf{-1.53 \pm 0.43}$ | $\mathbf{-1.59 \pm 0.44}$ |

Table 10: KDE Loglikelihood per step $t'$ on semi-synthetic dataset with bandwidth 0.3. Best per column in **bold**.

| Model | $t'=1$ | $t'=2$ | $t'=3$ | $t'=4$ | $t'=5$ | $t'=6$ | $t'=7$ | $t'=8$ | $t'=9$ | $t'=10$ | $t'=11$ |
|---|---|---|---|---|---|---|---|---|---|---|---|
| G-Net | $-0.02 \pm 0.02$ | $-0.79 \pm 0.15$ | $-1.26 \pm 0.20$ | $-1.59 \pm 0.23$ | $-1.84 \pm 0.24$ | $-2.04 \pm 0.26$ | $-2.22 \pm 0.27$ | $-2.36 \pm 0.29$ | $-2.48 \pm 0.30$ | $-2.58 \pm 0.31$ | $-2.67 \pm 0.33$ |
| Transformer G-Net | $-1.09 \pm 0.19$ | $-1.00 \pm 0.22$ | $-1.34 \pm 0.23$ | $-1.58 \pm 0.30$ | $-1.77 \pm 0.36$ | $-1.93 \pm 0.42$ | $-2.06 \pm 0.49$ | $-2.18 \pm 0.54$ | $-2.29 \pm 0.60$ | $-2.38 \pm 0.65$ | $-2.47 \pm 0.69$ |
| CT-CRPS | $\mathbf{0.38 \pm 0.04}$ | $-0.63 \pm 0.40$ | $-1.02 \pm 0.46$ | $-1.35 \pm 0.49$ | $-1.64 \pm 0.48$ | $-1.90 \pm 0.49$ | $-2.15 \pm 0.48$ | $-2.38 \pm 0.48$ | $-2.57 \pm 0.48$ | $-2.76 \pm 0.49$ | $-2.92 \pm 0.49$ |
| CT-Gaussian | $0.37 \pm 0.03$ | $-0.51 \pm 0.35$ | $-0.75 \pm 0.40$ | $-0.94 \pm 0.43$ | $-1.09 \pm 0.44$ | $-1.23 \pm 0.45$ | $-1.34 \pm 0.46$ | $-1.45 \pm 0.46$ | $-1.55 \pm 0.46$ | $-1.64 \pm 0.46$ | $-1.73 \pm 0.46$ |
| D.S. G-VAE (Gaussian) | $-1.29 \pm 0.07$ | $-1.88 \pm 0.08$ | $-2.23 \pm 0.12$ | $-2.43 \pm 0.15$ | $-2.54 \pm 0.17$ | $-2.62 \pm 0.18$ | $-2.67 \pm 0.18$ | $-2.71 \pm 0.19$ | $-2.73 \pm 0.19$ | $-2.75 \pm 0.19$ | $-2.77 \pm 0.19$ |
| D.S. G-VAE (CRPS) | $0.08 \pm 0.04$ | $\mathbf{-0.47 \pm 0.27}$ | $\mathbf{-0.72 \pm 0.31}$ | $\mathbf{-0.90 \pm 0.33}$ | $\mathbf{-1.04 \pm 0.34}$ | $\mathbf{-1.16 \pm 0.34}$ | $-1.27 \pm 0.34$ | $-1.35 \pm 0.33$ | $-1.43 \pm 0.32$ | $-1.50 \pm 0.30$ | $-1.56 \pm 0.29$ |
| G-Latent (Gaussian) | $-1.41 \pm 0.09$ | $-1.54 \pm 0.13$ | $-1.63 \pm 0.15$ | $-1.69 \pm 0.17$ | $-1.74 \pm 0.18$ | $-1.79 \pm 0.19$ | $-1.82 \pm 0.20$ | $-1.85 \pm 0.20$ | $-1.87 \pm 0.20$ | $-1.89 \pm 0.20$ | $-1.91 \pm 0.20$ |
| G-Latent (CRPS) | $-0.24 \pm 0.12$ | $-0.59 \pm 0.22$ | $-0.80 \pm 0.27$ | $-0.96 \pm 0.30$ | $-1.08 \pm 0.32$ | $-1.18 \pm 0.33$ | $\mathbf{-1.26 \pm 0.34}$ | $\mathbf{-1.32 \pm 0.34}$ | $\mathbf{-1.38 \pm 0.34}$ | $\mathbf{-1.42 \pm 0.34}$ | $\mathbf{-1.47 \pm 0.34}$ |

Table 11: KDE Loglikelihood per step $t'$ on semi-synthetic dataset with bandwidth 0.4. Best per column in **bold**.

| Model | $t'=1$ | $t'=2$ | $t'=3$ | $t'=4$ | $t'=5$ | $t'=6$ | $t'=7$ | $t'=8$ | $t'=9$ | $t'=10$ | $t'=11$ |
|---|---|---|---|---|---|---|---|---|---|---|---|
| G-Net | $-0.37 \pm 0.01$ | $-0.91 \pm 0.12$ | $-1.27 \pm 0.17$ | $-1.53 \pm 0.19$ | $-1.74 \pm 0.21$ | $-1.91 \pm 0.22$ | $-2.06 \pm 0.24$ | $-2.18 \pm 0.25$ | $-2.28 \pm 0.26$ | $-2.37 \pm 0.27$ | $-2.45 \pm 0.28$ |
| Transformer G-Net | $-1.12 \pm 0.17$ | $-1.10 \pm 0.19$ | $-1.35 \pm 0.21$ | $-1.54 \pm 0.26$ | $-1.69 \pm 0.31$ | $-1.84 \pm 0.33$ | $-2.01 \pm 0.45$ | $-2.10 \pm 0.49$ | $-2.17 \pm 0.53$ | $-2.24 \pm 0.56$ | |
| CT-CRPS | $\mathbf{-0.12 \pm 0.02}$ | $-0.75 \pm 0.26$ | $-1.00 \pm 0.30$ | $-1.22 \pm 0.33$ | $-1.40 \pm 0.34$ | $-1.57 \pm 0.35$ | $-1.73 \pm 0.35$ | $-1.87 \pm 0.35$ | $-2.00 \pm 0.35$ | $-2.11 \pm 0.36$ | $-2.22 \pm 0.36$ |
| CT-Gaussian | $-0.13 \pm 0.02$ | $-0.73 \pm 0.26$ | $-0.91 \pm 0.31$ | $-1.05 \pm 0.34$ | $-1.17 \pm 0.35$ | $-1.27 \pm 0.37$ | $-1.36 \pm 0.37$ | $-1.44 \pm 0.38$ | $-1.51 \pm 0.38$ | $-1.58 \pm 0.38$ | $-1.64 \pm 0.38$ |
| D.S. G-VAE (Gaussian) | $-1.40 \pm 0.06$ | $-1.93 \pm 0.07$ | $-2.26 \pm 0.12$ | $-2.45 \pm 0.14$ | $-2.56 \pm 0.16$ | $-2.63 \pm 0.17$ | $-2.68 \pm 0.17$ | $-2.72 \pm 0.18$ | $-2.74 \pm 0.18$ | $-2.76 \pm 0.18$ | $-2.78 \pm 0.18$ |
| D.S. G-VAE (CRPS) | $-0.30 \pm 0.03$ | $\mathbf{-0.70 \pm 0.21}$ | $\mathbf{-0.89 \pm 0.25}$ | $\mathbf{-1.03 \pm 0.29}$ | $\mathbf{-1.14 \pm 0.29}$ | $\mathbf{-1.24 \pm 0.29}$ | $-1.33 \pm 0.29$ | $-1.40 \pm 0.29$ | $-1.46 \pm 0.28$ | $-1.52 \pm 0.27$ | $-1.58 \pm 0.26$ |
| G-Latent (Gaussian) | $-1.51 \pm 0.08$ | $-1.62 \pm 0.11$ | $-1.70 \pm 0.14$ | $-1.76 \pm 0.15$ | $-1.80 \pm 0.16$ | $-1.84 \pm 0.17$ | $-1.87 \pm 0.18$ | $-1.90 \pm 0.18$ | $-1.92 \pm 0.18$ | $-1.94 \pm 0.18$ | $-1.95 \pm 0.18$ |
| G-Latent (CRPS) | $-0.53 \pm 0.09$ | $-0.78 \pm 0.17$ | $-0.95 \pm 0.21$ | $-1.08 \pm 0.24$ | $-1.18 \pm 0.26$ | $-1.26 \pm 0.27$ | $\mathbf{-1.32 \pm 0.28}$ | $\mathbf{-1.37 \pm 0.29}$ | $\mathbf{-1.42 \pm 0.29}$ | $\mathbf{-1.46 \pm 0.29}$ | $\mathbf{-1.50 \pm 0.29}$ |

Table 12: RMSE per step $t'$ on semi-synthetic dataset (corrected benchmark). Best per column in **bold**.

| Model | $t'=1$ | $t'=2$ | $t'=3$ | $t'=4$ | $t'=5$ | $t'=6$ | $t'=7$ | $t'=8$ | $t'=9$ | $t'=10$ | $t'=11$ |
|---|---|---|---|---|---|---|---|---|---|---|---|
| G-Net | $0.28 \pm 0.01$ | $0.51 \pm 0.05$ | $0.64 \pm 0.07$ | $0.74 \pm 0.08$ | $0.81 \pm 0.09$ | $0.88 \pm 0.09$ | $0.94 \pm 0.10$ | $0.98 \pm 0.11$ | $1.02 \pm 0.11$ | $1.06 \pm 0.12$ | $1.09 \pm 0.12$ |
| Transformer G-Net | $0.60 \pm 0.06$ | $0.56 \pm 0.06$ | $0.66 \pm 0.08$ | $0.74 \pm 0.10$ | $0.80 \pm 0.13$ | $0.84 \pm 0.15$ | $0.89 \pm 0.17$ | $0.92 \pm 0.19$ | $0.95 \pm 0.21$ | $0.98 \pm 0.22$ | $1.00 \pm 0.23$ |
| CT-Gaussian | $0.17 \pm 0.02$ | $0.46 \pm 0.11$ | $0.54 \pm 0.13$ | $0.60 \pm 0.14$ | $0.64 \pm 0.14$ | $0.68 \pm 0.14$ | $0.71 \pm 0.14$ | $0.74 \pm 0.14$ | $0.76 \pm 0.14$ | $0.79 \pm 0.14$ | $0.81 \pm 0.14$ |
| CT-CRPS | $0.16 \pm 0.02$ | $0.48 \pm 0.10$ | $0.58 \pm 0.11$ | $0.65 \pm 0.11$ | $0.71 \pm 0.10$ | $0.76 \pm 0.10$ | $0.80 \pm 0.10$ | $0.84 \pm 0.10$ | $0.87 \pm 0.10$ | $0.89 \pm 0.10$ | $0.92 \pm 0.10$ |
| CT | $\mathbf{0.14 \pm 0.01}$ | $\mathbf{0.34 \pm 0.07}$ | $\mathbf{0.43 \pm 0.10}$ | $\mathbf{0.49 \pm 0.11}$ | $\mathbf{0.53 \pm 0.12}$ | $\mathbf{0.56 \pm 0.13}$ | $\mathbf{0.58 \pm 0.13}$ | $\mathbf{0.60 \pm 0.13}$ | $\mathbf{0.62 \pm 0.13}$ | $\mathbf{0.64 \pm 0.13}$ | $\mathbf{0.65 \pm 0.13}$ |
| D.S. G-VAE (Gaussian) | $0.26 \pm 0.01$ | $0.44 \pm 0.06$ | $0.54 \pm 0.09$ | $0.61 \pm 0.10$ | $0.66 \pm 0.11$ | $0.70 \pm 0.12$ | $0.73 \pm 0.12$ | $0.76 \pm 0.13$ | $0.79 \pm 0.13$ | $0.81 \pm 0.13$ | $0.83 \pm 0.13$ |
| D.S. G-VAE (CRPS) | $0.23 \pm 0.01$ | $0.40 \pm 0.07$ | $0.49 \pm 0.10$ | $0.55 \pm 0.11$ | $0.59 \pm 0.12$ | $0.63 \pm 0.12$ | $0.66 \pm 0.12$ | $0.69 \pm 0.12$ | $0.72 \pm 0.12$ | $0.74 \pm 0.12$ | $0.76 \pm 0.12$ |
| G-Latent (Gaussian) | $0.35 \pm 0.03$ | $0.46 \pm 0.06$ | $0.53 \pm 0.09$ | $0.58 \pm 0.10$ | $0.61 \pm 0.11$ | $0.64 \pm 0.12$ | $0.67 \pm 0.12$ | $0.69 \pm 0.12$ | $0.71 \pm 0.12$ | $0.72 \pm 0.12$ | $0.73 \pm 0.12$ |
| G-Latent (CRPS) | $0.33 \pm 0.04$ | $0.44 \pm 0.07$ | $0.51 \pm 0.10$ | $0.56 \pm 0.11$ | $0.60 \pm 0.12$ | $0.63 \pm 0.12$ | $0.66 \pm 0.13$ | $0.68 \pm 0.13$ | $0.70 \pm 0.13$ | $0.71 \pm 0.13$ | $0.73 \pm 0.13$ |

Table 13: Empirical coverage (%) by step and dimension for each quantile $q$. Steps $t' \in \{1, \ldots, 6\}$, two outcome dimensions.

| Model | Step 1 Dim 1 | Step 1 Dim 2 | Step 2 Dim 1 | Step 2 Dim 2 | Step 3 Dim 1 | Step 3 Dim 2 | Step 4 Dim 1 | Step 4 Dim 2 | Step 5 Dim 1 | Step 5 Dim 2 | Step 6 Dim 1 | Step 6 Dim 2 |
|---|---|---|---|---|---|---|---|---|---|---|---|---|
| **Quantile $q = 0.1$** | | | | | | | | | | | | |
| G-Net | $11.54 \pm 3.61$ | $12.61 \pm 3.20$ | $14.04 \pm 3.53$ | $15.62 \pm 4.43$ | $15.33 \pm 4.63$ | $17.32 \pm 4.86$ | $16.19 \pm 5.38$ | $18.38 \pm 5.14$ | $16.86 \pm 5.73$ | $19.17 \pm 5.29$ | $17.33 \pm 5.97$ | $19.82 \pm 5.32$ |
| Transformer G-Net | $23.52 \pm 9.91$ | $24.92 \pm 5.68$ | $11.51 \pm 6.44$ | $16.22 \pm 4.97$ | $11.87 \pm 7.14$ | $18.83 \pm 4.85$ | $12.50 \pm 7.51$ | $20.69 \pm 4.54$ | $13.21 \pm 7.90$ | $22.26 \pm 4.31$ | $13.88 \pm 8.18$ | $23.55 \pm 4.20$ |
| CT-CRPS | $8.78 \pm 2.77$ | $15.43 \pm 6.74$ | $16.47 \pm 1.73$ | $21.24 \pm 7.52$ | $17.94 \pm 2.02$ | $25.14 \pm 8.93$ | $19.28 \pm 2.47$ | $28.42 \pm 9.75$ | $20.30 \pm 2.60$ | $30.79 \pm 10.73$ | $21.08 \pm 2.77$ | $32.92 \pm 11.52$ |
| CT-Gaussian | $13.91 \pm 6.81$ | $15.34 \pm 5.09$ | $9.59 \pm 2.86$ | $13.11 \pm 2.40$ | $11.08 \pm 3.35$ | $16.69 \pm 3.56$ | $12.44 \pm 4.19$ | $19.45 \pm 4.15$ | $13.45 \pm 4.95$ | $21.59 \pm 4.36$ | $14.09 \pm 5.50$ | $23.30 \pm 4.57$ |
| D.S. G-VAE (Gaussian) | $0.19 \pm 0.03$ | $0.07 \pm 0.05$ | $0.72 \pm 0.38$ | $0.19 \pm 0.19$ | $0.71 \pm 0.40$ | $0.16 \pm 0.17$ | $0.67 \pm 0.38$ | $0.14 \pm 0.15$ | $0.66 \pm 0.40$ | $0.13 \pm 0.14$ | $0.68 \pm 0.41$ | $0.14 \pm 0.14$ |
| D.S. G-VAE (CRPS) | $6.87 \pm 2.35$ | $5.07 \pm 1.70$ | $8.77 \pm 4.42$ | $7.00 \pm 1.62$ | $10.97 \pm 4.77$ | $8.70 \pm 1.72$ | $12.62 \pm 4.49$ | $\mathbf{10.10 \pm 2.20}$ | $13.89 \pm 4.11$ | $\mathbf{10.96 \pm 2.77}$ | $14.79 \pm 3.84$ | $\mathbf{11.54 \pm 3.19}$ |
| G-Latent (Gaussian) | $0.47 \pm 0.07$ | $0.11 \pm 0.10$ | $1.38 \pm 0.51$ | $0.41 \pm 0.42$ | $1.99 \pm 0.82$ | $0.64 \pm 0.61$ | $2.40 \pm 1.00$ | $0.82 \pm 0.75$ | $2.70 \pm 1.13$ | $1.00 \pm 0.88$ | $2.97 \pm 1.23$ | $1.13 \pm 0.95$ |
| G-Latent (CRPS) | $\mathbf{8.89 \pm 1.89}$ | $\mathbf{9.48 \pm 3.75}$ | $\mathbf{9.88 \pm 1.77}$ | $\mathbf{10.23 \pm 2.93}$ | $\mathbf{10.37 \pm 1.88}$ | $\mathbf{10.86 \pm 2.97}$ | $\mathbf{10.79 \pm 1.94}$ | $11.46 \pm 3.35$ | $\mathbf{11.17 \pm 1.95}$ | $11.96 \pm 3.70$ | $\mathbf{11.45 \pm 2.04}$ | $12.64 \pm 3.88$ |
| **Quantile $q = 0.2$** | | | | | | | | | | | | |
| G-Net | $\mathbf{20.30 \pm 4.30}$ | $\mathbf{21.89 \pm 4.67}$ | $22.66 \pm 4.04$ | $24.36 \pm 5.21$ | $23.72 \pm 5.00$ | $25.89 \pm 5.56$ | $24.55 \pm 5.55$ | $26.78 \pm 5.77$ | $25.25 \pm 5.82$ | $27.51 \pm 5.98$ | $25.63 \pm 6.07$ | $28.08 \pm 6.03$ |
| Transformer G-Net | $33.95 \pm 9.60$ | $34.79 \pm 5.36$ | $\mathbf{20.21 \pm 6.78}$ | $25.21 \pm 5.47$ | $\mathbf{20.03 \pm 7.62}$ | $27.75 \pm 4.93$ | $20.55 \pm 4.18$ | $29.73 \pm 4.58$ | $21.33 \pm 8.57$ | $31.35 \pm 4.33$ | $22.08 \pm 8.83$ | $32.68 \pm 4.41$ |
| CT-CRPS | $15.22 \pm 4.00$ | $23.94 \pm 9.75$ | $23.21 \pm 1.82$ | $29.47 \pm 10.16$ | $24.37 \pm 2.21$ | $32.93 \pm 11.42$ | $26.14 \pm 12.25$ | $36.14 \pm 12.25$ | $26.62 \pm 2.80$ | $38.33 \pm 13.10$ | $27.46 \pm 3.06$ | $40.18 \pm 13.70$ |
| CT-Gaussian | $23.29 \pm 9.12$ | $25.21 \pm 5.91$ | $16.84 \pm 4.69$ | $22.56 \pm 3.60$ | $18.65 \pm 5.30$ | $26.19 \pm 4.67$ | $\mathbf{20.27 \pm 6.21}$ | $28.84 \pm 4.76$ | $21.30 \pm 6.92$ | $30.93 \pm 4.85$ | $22.14 \pm 7.56$ | $32.63 \pm 4.84$ |
| D.S. G-VAE (Gaussian) | $0.76 \pm 0.15$ | $0.34 \pm 0.18$ | $2.21 \pm 0.75$ | $0.70 \pm 0.61$ | $2.33 \pm 0.83$ | $0.69 \pm 0.62$ | $2.51 \pm 0.88$ | $0.73 \pm 0.63$ | $2.65 \pm 0.92$ | $0.75 \pm 0.63$ | $2.81 \pm 0.99$ | $0.80 \pm 0.66$ |
| D.S. G-VAE (CRPS) | $12.87 \pm 3.62$ | $9.69 \pm 2.25$ | $16.85 \pm 6.39$ | $14.61 \pm 2.69$ | $20.08 \pm 6.02$ | $16.99 \pm 3.08$ | $22.47 \pm 5.44$ | $18.73 \pm 3.79$ | $24.08 \pm 4.96$ | $\mathbf{19.83 \pm 4.57}$ | $25.22 \pm 4.80$ | $\mathbf{20.38 \pm 5.22}$ |
| G-Latent (Gaussian) | $1.98 \pm 0.43$ | $0.69 \pm 0.41$ | $4.28 \pm 1.06$ | $1.68 \pm 1.21$ | $5.63 \pm 1.40$ | $2.41 \pm 1.59$ | $6.62 \pm 1.66$ | $3.08 \pm 1.90$ | $7.40 \pm 1.84$ | $3.64 \pm 2.06$ | $8.09 \pm 1.97$ | $4.23 \pm 2.20$ |
| G-Latent (CRPS) | $17.69 \pm 2.88$ | $17.16 \pm 5.17$ | $18.63 \pm 2.75$ | $\mathbf{18.70 \pm 3.85}$ | $19.03 \pm 2.78$ | $\mathbf{19.54 \pm 3.65}$ | $19.45 \pm 2.94$ | $\mathbf{20.38 \pm 3.91}$ | $\mathbf{19.80 \pm 2.99}$ | $20.98 \pm 4.09$ | $\mathbf{20.22 \pm 3.16}$ | $21.66 \pm 4.31$ |
| **Quantile $q = 0.3$** | | | | | | | | | | | | |
| G-Net | $\mathbf{29.49 \pm 4.64}$ | $\mathbf{30.87 \pm 5.44}$ | $31.03 \pm 4.05$ | $32.74 \pm 5.34$ | $\mathbf{31.83 \pm 4.80}$ | $33.78 \pm 5.76$ | $32.44 \pm 5.30$ | $34.53 \pm 6.03$ | $32.79 \pm 5.58$ | $34.99 \pm 6.12$ | $33.10 \pm 5.82$ | $35.39 \pm 6.13$ |
| Transformer G-Net | $42.26 \pm 8.68$ | $42.59 \pm 4.71$ | $29.37 \pm 6.17$ | $33.79 \pm 4.98$ | $28.68 \pm 7.22$ | $35.88 \pm 4.54$ | $28.95 \pm 7.90$ | $37.62 \pm 4.32$ | $\mathbf{29.54 \pm 8.34}$ | $39.11 \pm 4.39$ | $\mathbf{30.19 \pm 8.69}$ | $40.30 \pm 4.69$ |
| CT-CRPS | $23.56 \pm 4.87$ | $34.25 \pm 13.32$ | $\mathbf{30.13 \pm 1.90}$ | $36.98 \pm 11.97$ | $30.70 \pm 2.32$ | $39.75 \pm 13.28$ | $31.64 \pm 2.85$ | $42.56 \pm 13.93$ | $32.45 \pm 2.77$ | $44.47 \pm 14.82$ | $33.22 \pm 3.08$ | $46.00 \pm 15.33$ |
| CT-Gaussian | $31.98 \pm 10.51$ | $34.65 \pm 6.29$ | $25.03 \pm 6.30$ | $32.23 \pm 4.57$ | $26.86 \pm 6.79$ | $35.19 \pm 5.31$ | $28.40 \pm 7.72$ | $37.57 \pm 5.50$ | $29.52 \pm 8.36$ | $39.35 \pm 5.53$ | $30.23 \pm 9.01$ | $40.89 \pm 5.62$ |
| D.S. G-VAE (Gaussian) | $3.76 \pm 1.18$ | $2.64 \pm 0.99$ | $6.90 \pm 1.51$ | $3.47 \pm 1.64$ | $7.43 \pm 1.49$ | $3.63 \pm 1.75$ | $7.98 \pm 1.62$ | $3.94 \pm 1.81$ | $8.53 \pm 1.71$ | $4.24 \pm 1.84$ | $8.97 \pm 1.81$ | $4.53 \pm 2.06$ |
| D.S. G-VAE (CRPS) | $22.96 \pm 4.69$ | $18.36 \pm 2.97$ | $27.09 \pm 7.03$ | $24.62 \pm 4.17$ | $\mathbf{30.46 \pm 6.15}$ | $26.69 \pm 4.69$ | $32.77 \pm 5.57$ | $28.19 \pm 5.54$ | $34.29 \pm 5.12$ | $29.09 \pm 6.40$ | $35.35 \pm 5.03$ | $\mathbf{29.55 \pm 7.24}$ |
| G-Latent (Gaussian) | $7.37 \pm 1.45$ | $4.11 \pm 1.18$ | $11.29 \pm 1.96$ | $6.54 \pm 2.34$ | $13.44 \pm 1.99$ | $8.21 \pm 2.76$ | $15.00 \pm 2.16$ | $9.67 \pm 3.01$ | $16.13 \pm 2.24$ | $10.93 \pm 3.14$ | $17.07 \pm 2.28$ | $12.23 \pm 3.24$ |
| G-Latent (CRPS) | $28.14 \pm 3.58$ | $26.52 \pm 6.23$ | $28.71 \pm 3.42$ | $\mathbf{28.72 \pm 4.76}$ | $28.97 \pm 3.39$ | $\mathbf{29.51 \pm 4.49}$ | $\mathbf{29.09 \pm 3.49}$ | $\mathbf{30.30 \pm 4.61}$ | $29.24 \pm 3.73$ | $\mathbf{30.84 \pm 4.67}$ | $29.50 \pm 4.00$ | $31.37 \pm 4.71$ |
| **Quantile $q = 0.4$** | | | | | | | | | | | | |
| G-Net | $39.30 \pm 4.56$ | $\mathbf{39.96 \pm 5.65}$ | $39.36 \pm 3.81$ | $40.98 \pm 5.16$ | $\mathbf{39.75 \pm 4.48}$ | $41.53 \pm 5.58$ | $\mathbf{40.05 \pm 4.83}$ | $41.90 \pm 5.77$ | $\mathbf{40.08 \pm 5.16}$ | $42.23 \pm 5.96$ | $\mathbf{40.25 \pm 5.45}$ | $42.47 \pm 5.87$ |
| Transformer G-Net | $49.42 \pm 7.43$ | $49.15 \pm 3.94$ | $38.83 \pm 5.24$ | $42.16 \pm 4.20$ | $37.70 \pm 6.49$ | $43.56 \pm 4.07$ | $37.60 \pm 7.25$ | $44.91 \pm 4.17$ | $37.86 \pm 7.79$ | $46.19 \pm 4.58$ | $38.36 \pm 8.16$ | $47.16 \pm 5.12$ |
| CT-CRPS | $33.40 \pm 5.09$ | $45.33 \pm 16.41$ | $37.10 \pm 1.93$ | $44.14 \pm 13.20$ | $37.17 \pm 2.27$ | $46.22 \pm 14.48$ | $37.54 \pm 2.81$ | $48.42 \pm 15.22$ | $38.19 \pm 2.68$ | $49.88 \pm 16.13$ | $38.81 \pm 2.96$ | $51.04 \pm 16.60$ |
| CT-Gaussian | $40.47 \pm 11.36$ | $44.10 \pm 6.51$ | $34.61 \pm 7.28$ | $42.02 \pm 5.43$ | $36.02 \pm 7.71$ | $44.15 \pm 6.19$ | $37.24 \pm 8.59$ | $45.92 \pm 6.50$ | $38.19 \pm 9.10$ | $47.40 \pm 6.68$ | $38.72 \pm 9.68$ | $48.63 \pm 6.89$ |
| D.S. G-VAE (Gaussian) | $16.97 \pm 4.62$ | $16.07 \pm 4.34$ | $20.70 \pm 4.03$ | $16.98 \pm 3.41$ | $21.27 \pm 3.99$ | $17.06 \pm 3.17$ | $22.05 \pm 4.26$ | $17.60 \pm 3.00$ | $22.62 \pm 4.37$ | $18.10 \pm 3.13$ | $23.21 \pm 4.61$ | $18.63 \pm 3.41$ |
| D.S. G-VAE (CRPS) | $37.24 \pm 5.25$ | $31.72 \pm 3.78$ | $39.28 \pm 6.34$ | $36.33 \pm 5.70$ | $41.53 \pm 5.29$ | $37.27 \pm 6.26$ | $43.15 \pm 4.92$ | $38.11 \pm 7.05$ | $44.32 \pm 4.88$ | $38.69 \pm 8.00$ | $45.29 \pm 4.99$ | $\mathbf{38.91 \pm 8.96}$ |
| G-Latent (Gaussian) | $22.09 \pm 3.15$ | $17.47 \pm 1.78$ | $25.71 \pm 2.68$ | $21.07 \pm 2.86$ | $27.68 \pm 2.06$ | $22.93 \pm 3.07$ | $28.94 \pm 1.81$ | $24.61 \pm 3.27$ | $29.92 \pm 1.85$ | $26.05 \pm 3.56$ | $30.64 \pm 1.96$ | $27.42 \pm 3.54$ |
| G-Latent (CRPS) | $\mathbf{39.69 \pm 3.84}$ | $37.54 \pm 7.02$ | $\mathbf{39.71 \pm 3.67}$ | $\mathbf{39.73 \pm 5.64}$ | $39.50 \pm 3.41$ | $\mathbf{40.30 \pm 5.23}$ | $39.37 \pm 3.55$ | $\mathbf{40.76 \pm 5.17}$ | $39.29 \pm 3.90$ | $\mathbf{41.26 \pm 5.14}$ | $39.31 \pm 4.35$ | $\mathbf{41.53 \pm 4.98}$ |
| **Quantile $q = 0.5$** | | | | | | | | | | | | |
| G-Net | $\mathbf{49.05 \pm 4.66}$ | $49.09 \pm 5.64$ | $47.81 \pm 3.51$ | $49.36 \pm 4.69$ | $47.71 \pm 3.99$ | $\mathbf{49.25 \pm 5.01}$ | $47.59 \pm 4.32$ | $\mathbf{49.41 \pm 5.24}$ | $47.37 \pm 4.70$ | $\mathbf{49.54 \pm 5.27}$ | $47.41 \pm 4.87$ | $\mathbf{49.59 \pm 5.31}$ |
| Transformer G-Net | $56.04 \pm 6.10$ | $55.32 \pm 3.34$ | $48.31 \pm 4.37$ | $\mathbf{50.45 \pm 3.43}$ | $46.95 \pm 5.74$ | $51.06 \pm 3.73$ | $46.42 \pm 6.71$ | $51.99 \pm 4.27$ | $46.48 \pm 7.08$ | $52.83 \pm 4.91$ | $46.67 \pm 7.58$ | $53.60 \pm 5.48$ |
| CT-CRPS | $44.20 \pm 4.79$ | $56.28 \pm 18.16$ | $44.36 \pm 1.99$ | $51.15 \pm 13.87$ | $43.81 \pm 2.15$ | $52.45 \pm 15.23$ | $43.75 \pm 2.68$ | $53.92 \pm 16.09$ | $44.01 \pm 2.50$ | $54.91 \pm 17.03$ | $44.45 \pm 2.75$ | $55.73 \pm 17.49$ |
| CT-Gaussian | $48.97 \pm 11.83$ | $53.52 \pm 6.59$ | $45.63 \pm 7.09$ | $52.16 \pm 6.32$ | $46.06 \pm 7.75$ | $53.13 \pm 7.08$ | $46.74 \pm 8.68$ | $54.13 \pm 7.68$ | $47.43 \pm 9.05$ | $55.10 \pm 7.93$ | $47.73 \pm 9.52$ | $55.99 \pm 8.29$ |
| D.S. G-VAE (Gaussian) | $47.05 \pm 6.83$ | $\mathbf{49.69 \pm 6.91}$ | $46.99 \pm 6.87$ | $48.96 \pm 4.58$ | $46.34 \pm 6.82$ | $47.74 \pm 3.90$ | $46.19 \pm 7.08$ | $47.40 \pm 3.49$ | $45.89 \pm 7.35$ | $47.05 \pm 3.54$ | $45.91 \pm 7.47$ | $46.66 \pm 3.64$ |
| D.S. G-VAE (CRPS) | $53.98 \pm 5.08$ | $48.64 \pm 4.45$ | $52.34 \pm 4.89$ | $49.09 \pm 6.81$ | $52.94 \pm 3.83$ | $48.23 \pm 7.47$ | $53.57 \pm 3.61$ | $48.14 \pm 8.39$ | $54.39 \pm 3.97$ | $48.22 \pm 9.23$ | $55.05 \pm 4.41$ | $48.18 \pm 10.11$ |
| G-Latent (Gaussian) | $47.85 \pm 3.78$ | $45.75 \pm 2.81$ | $47.79 \pm 2.33$ | $47.09 \pm 3.62$ | $47.81 \pm 1.50$ | $47.31 \pm 3.57$ | $47.70 \pm 0.97$ | $47.55 \pm 3.55$ | $47.65 \pm 0.91$ | $47.79 \pm 3.73$ | $47.58 \pm 1.11$ | $48.14 \pm 3.57$ |
| G-Latent (CRPS) | $51.93 \pm 4.23$ | $49.44 \pm 7.47$ | $\mathbf{51.21 \pm 3.54}$ | $51.36 \pm 6.28$ | $\mathbf{50.55 \pm 3.01}$ | $51.47 \pm 5.84$ | $\mathbf{50.01 \pm 3.08}$ | $51.51 \pm 5.14$ | $\mathbf{49.66 \pm 3.57}$ | $51.74 \pm 5.34$ | $\mathbf{49.39 \pm 3.99}$ | $51.83 \pm 5.15$ |
| **Quantile $q = 0.6$** | | | | | | | | | | | | |
| G-Net | $\mathbf{58.50 \pm 4.92}$ | $57.96 \pm 5.13$ | $56.38 \pm 3.19$ | $57.89 \pm 3.92$ | $55.67 \pm 3.46$ | $57.24 \pm 4.10$ | $55.21 \pm 3.83$ | $57.07 \pm 4.25$ | $54.84 \pm 4.17$ | $56.99 \pm 4.37$ | $54.63 \pm 4.25$ | $56.85 \pm 4.34$ |
| Transformer G-Net | $62.36 \pm 4.68$ | $\mathbf{61.25 \pm 2.91}$ | $\mathbf{57.87 \pm 3.91}$ | $\mathbf{58.71 \pm 3.07}$ | $56.41 \pm 5.32$ | $58.70 \pm 3.81$ | $55.08 \pm 4.48$ | $55.18 \pm 6.60$ | $55.10 \pm 7.05$ | $59.53 \pm 5.13$ | $\mathbf{60.03 \pm 5.81}$ | |
| CT-CRPS | $55.38 \pm 4.16$ | $66.33 \pm 17.93$ | $52.11 \pm 2.05$ | $58.22 \pm 13.95$ | $50.72 \pm 1.92$ | $58.58 \pm 15.48$ | $50.07 \pm 2.44$ | $\mathbf{59.27 \pm 16.52}$ | $50.06 \pm 2.28$ | $\mathbf{59.78 \pm 17.48}$ | $50.24 \pm 2.48$ | $\mathbf{60.29 \pm 18.09}$ |
| CT-Gaussian | $57.63 \pm 11.87$ | $62.93 \pm 6.43$ | $55.70 \pm 6.10$ | $62.24 \pm 6.96$ | $57.02 \pm 6.97$ | $62.12 \pm 7.91$ | $56.93 \pm 7.86$ | $62.19 \pm 8.70$ | $57.10 \pm 8.23$ | $62.62 \pm 9.04$ | $57.15 \pm 8.66$ | $63.11 \pm 9.57$ |
| D.S. G-VAE (Gaussian) | $79.08 \pm 3.99$ | $83.38 \pm 4.12$ | $74.80 \pm 5.67$ | $81.31 \pm 4.08$ | $73.20 \pm 5.82$ | $79.54 \pm 3.97$ | $71.92 \pm 6.25$ | $78.19 \pm 4.00$ | $70.81 \pm 6.75$ | $77.05 \pm 3.99$ | $70.09 \pm 6.94$ | $75.91 \pm 3.98$ |
| D.S. G-VAE (CRPS) | $69.93 \pm 4.47$ | $65.69 \pm 4.62$ | $65.18 \pm 4.32$ | $61.74 \pm 7.18$ | $64.13 \pm 2.11$ | $\mathbf{59.24 \pm 8.27}$ | $63.82 \pm 2.13$ | $58.26 \pm 9.15$ | $64.14 \pm 2.74$ | $57.68 \pm 9.91$ | $64.50 \pm 3.55$ | $57.24 \pm 10.77$ |
| G-Latent (Gaussian) | $74.39 \pm 2.97$ | $76.17 \pm 3.06$ | $70.89 \pm 2.35$ | $74.39 \pm 4.06$ | $68.58 \pm 2.04$ | $72.32 \pm 4.20$ | $66.97 \pm 1.74$ | $70.90 \pm 3.99$ | $65.89 \pm 1.69$ | $69.99 \pm 3.71$ | $65.11 \pm 1.74$ | $69.05 \pm 3.49$ |
| G-Latent (CRPS) | $64.07 \pm 4.57$ | $61.35 \pm 7.07$ | $62.69 \pm 3.09$ | $62.99 \pm 6.41$ | $\mathbf{61.62 \pm 2.49}$ | $62.44 \pm 5.76$ | $\mathbf{60.70 \pm 2.50}$ | $62.09 \pm 5.35$ | $\mathbf{60.15 \pm 2.92}$ | $62.01 \pm 5.05$ | $59.73 \pm 3.28$ | $61.94 \pm 4.84$ |
| **Quantile $q = 0.7$** | | | | | | | | | | | | |
| G-Net | $67.66 \pm 4.61$ | $67.06 \pm 4.23$ | $65.19 \pm 2.77$ | $66.61 \pm 2.84$ | $63.86 \pm 2.99$ | $65.57 \pm 2.96$ | $63.02 \pm 3.32$ | $65.03 \pm 2.99$ | $62.64 \pm 3.58$ | $64.78 \pm 3.15$ | $62.19 \pm 3.62$ | $64.58 \pm 3.04$ |
| Transformer G-Net | $\mathbf{68.92 \pm 3.21}$ | $67.59 \pm 2.49$ | $67.48 \pm 3.69$ | $67.19 \pm 2.94$ | $65.91 \pm 5.03$ | $66.41 \pm 4.03$ | $64.74 \pm 5.80$ | $66.24 \pm 4.78$ | $64.16 \pm 6.13$ | $66.33 \pm 5.34$ | $63.92 \pm 6.59$ | $66.74 \pm 6.05$ |
| CT-CRPS | $66.21 \pm 3.47$ | $75.05 \pm 16.02$ | $60.32 \pm 2.02$ | $65.44 \pm 13.29$ | $58.02 \pm 1.58$ | $64.62 \pm 15.15$ | $56.84 \pm 2.09$ | $64.61 \pm 16.45$ | $56.44 \pm 1.96$ | $64.72 \pm 17.45$ | $56.34 \pm 2.09$ | $64.85 \pm 18.15$ |
| CT-Gaussian | $66.36 \pm 11.39$ | $\mathbf{72.26 \pm 6.10}$ | $\mathbf{69.74 \pm 4.92}$ | $\mathbf{71.90 \pm 7.01}$ | $\mathbf{68.10 \pm 5.79}$ | $70.69 \pm 8.36$ | $67.31 \pm 6.39$ | $\mathbf{70.15 \pm 9.30}$ | $67.04 \pm 6.86$ | $\mathbf{70.03 \pm 9.88}$ | $66.82 \pm 7.26$ | $\mathbf{70.12 \pm 10.51}$ |
| D.S. G-VAE (Gaussian) | $94.82 \pm 1.20$ | $97.07 \pm 0.98$ | $91.28 \pm 2.46$ | $95.74 \pm 1.96$ | $90.17 \pm 3.18$ | $94.89 \pm 2.40$ | $89.14 \pm 3.48$ | $94.04 \pm 2.76$ | $88.31 \pm 3.68$ | $93.33 \pm 2.95$ | $87.64 \pm 3.89$ | $92.61 \pm 3.13$ |
| D.S. G-VAE (CRPS) | $82.24 \pm 3.60$ | $79.56 \pm 3.88$ | $76.71 \pm 2.46$ | $73.45 \pm 6.88$ | $74.71 \pm 1.25$ | $\mathbf{69.89 \pm 8.27}$ | $73.69 \pm 0.97$ | $67.99 \pm 9.02$ | $73.55 \pm 1.63$ | $66.99 \pm 9.70$ | $73.61 \pm 2.51$ | $66.20 \pm 10.62$ |
| G-Latent (Gaussian) | $90.77 \pm 1.55$ | $93.29 \pm 1.66$ | $87.00 \pm 2.07$ | $91.13 \pm 2.26$ | $84.29 \pm 2.21$ | $89.05 \pm 2.76$ | $82.30 \pm 2.28$ | $87.31 \pm 2.78$ | $80.87 \pm 2.22$ | $86.07 \pm 2.71$ | $79.79 \pm 2.23$ | $85.00 \pm 2.69$ |
| G-Latent (CRPS) | $75.16 \pm 4.62$ | $72.44 \pm 6.07$ | $73.58 \pm 2.59$ | $73.60 \pm 5.58$ | $72.25 \pm 2.08$ | $72.77 \pm 4.86$ | $\mathbf{71.20 \pm 1.99}$ | $72.09 \pm 4.49$ | $\mathbf{70.52 \pm 2.23}$ | $71.65 \pm 4.31$ | $\mathbf{69.84 \pm 2.48}$ | $71.42 \pm 4.10$ |
| **Quantile $q = 0.8$** | | | | | | | | | | | | |
| G-Net | $\mathbf{76.99 \pm 3.33}$ | $76.63 \pm 2.86$ | $74.31 \pm 2.24$ | $75.61 \pm 1.59$ | $72.59 \pm 2.49$ | $74.43 \pm 1.69$ | $71.52 \pm 2.72$ | $73.65 \pm 1.69$ | $70.91 \pm 3.00$ | $73.16 \pm 1.76$ | $70.41 \pm 3.07$ | $72.78 \pm 1.70$ |
| Transformer G-Net | $75.77 \pm 2.76$ | $74.56 \pm 2.34$ | $77.00 \pm 3.52$ | $75.89 \pm 2.95$ | $75.45 \pm 4.73$ | $74.53 \pm 4.08$ | $74.19 \pm 5.32$ | $73.82 \pm 4.78$ | $73.44 \pm 5.61$ | $73.67 \pm 5.34$ | $73.09 \pm 6.09$ | $73.74 \pm 5.89$ |
| CT-CRPS | $75.94 \pm 2.93$ | $82.19 \pm 13.12$ | $69.10 \pm 1.83$ | $72.84 \pm 11.82$ | $65.98 \pm 1.21$ | $70.88 \pm 14.18$ | $64.35 \pm 1.51$ | $70.28 \pm 15.69$ | $63.54 \pm 1.49$ | $69.91 \pm 16.72$ | $63.13 \pm 1.67$ | $69.66 \pm 17.57$ |
| CT-Gaussian | $75.46 \pm 10.17$ | $\mathbf{81.47 \pm 5.26}$ | $\mathbf{80.43 \pm 3.61}$ | $\mathbf{80.78 \pm 6.27}$ | $\mathbf{78.44 \pm 4.33}$ | $78.93 \pm 8.03$ | $77.41 \pm 4.86$ | $77.89 \pm 9.30$ | $76.96 \pm 5.26$ | $77.43 \pm 10.11$ | $76.50 \pm 5.66$ | $77.15 \pm 10.84$ |
| D.S. G-VAE (Gaussian) | $98.85 \pm 0.42$ | $99.57 \pm 0.20$ | $97.32 \pm 1.27$ | $99.15 \pm 0.64$ | $97.02 \pm 1.42$ | $98.96 \pm 0.91$ | $96.64 \pm 1.60$ | $98.76 \pm 1.17$ | $96.46 \pm 1.54$ | $98.56 \pm 1.37$ | $96.21 \pm 1.59$ | $98.36 \pm 1.57$ |
| D.S. G-VAE (CRPS) | $90.14 \pm 2.69$ | $88.91 \pm 2.53$ | $85.93 \pm 1.96$ | $83.43 \pm 5.73$ | $83.87 \pm 1.49$ | $\mathbf{79.69 \pm 7.18}$ | $82.61 \pm 0.96$ | $77.46 \pm 7.84$ | $82.35 \pm 1.14$ | $76.16 \pm 8.66$ | $82.23 \pm 1.74$ | $75.16 \pm 9.54$ |
| G-Latent (Gaussian) | $97.22 \pm 0.52$ | $98.62 \pm 0.50$ | $95.09 \pm 1.20$ | $97.62 \pm 0.90$ | $93.41 \pm 1.59$ | $96.68 \pm 1.20$ | $92.06 \pm 1.80$ | $95.76 \pm 1.41$ | $91.01 \pm 1.93$ | $94.97 \pm 1.49$ | $90.14 \pm 1.98$ | $94.32 \pm 1.56$ |
| G-Latent (CRPS) | $84.67 \pm 3.98$ | $82.21 \pm 4.59$ | $83.17 \pm 2.01$ | $82.96 \pm 4.00$ | $81.91 \pm 1.46$ | $81.98 \pm 3.39$ | $\mathbf{80.85 \pm 1.50}$ | $\mathbf{81.16 \pm 3.08}$ | $\mathbf{80.16 \pm 1.61}$ | $\mathbf{80.59 \pm 3.02}$ | $\mathbf{79.53 \pm 1.80}$ | $\mathbf{80.20 \pm 2.80}$ |
| **Quantile $q = 0.9$** | | | | | | | | | | | | |
| G-Net | $86.33 \pm 1.84$ | $86.56 \pm 1.15$ | $84.13 \pm 1.69$ | $85.37 \pm 0.94$ | $82.38 \pm 2.08$ | $84.15 \pm 1.12$ | $81.28 \pm 2.25$ | $83.28 \pm 1.34$ | $80.63 \pm 2.31$ | $82.60 \pm 1.30$ | $80.10 \pm 2.36$ | $82.04 \pm 1.31$ |
| Transformer G-Net | $83.13 \pm 2.79$ | $82.18 \pm 2.41$ | $86.47 \pm 3.01$ | $85.23 \pm 2.64$ | $85.15 \pm 4.03$ | $83.49 \pm 3.69$ | $84.05 \pm 4.32$ | $82.41 \pm 4.32$ | $83.36 \pm 4.55$ | $81.98 \pm 4.67$ | $82.99 \pm 4.97$ | $81.77 \pm 5.17$ |
| CT-CRPS | $84.04 \pm 2.49$ | $87.89 \pm 9.91$ | $78.28 \pm 1.50$ | $80.56 \pm 9.36$ | $74.99 \pm 0.86$ | $77.92 \pm 12.01$ | $73.22 \pm 0.82$ | $76.70 \pm 13.88$ | $72.09 \pm 0.89$ | $75.91 \pm 15.11$ | $71.42 \pm 1.15$ | $75.34 \pm 16.09$ |
| CT-Gaussian | $85.16 \pm 7.78$ | $\mathbf{90.35 \pm 3.74}$ | $\mathbf{89.36 \pm 1.94}$ | $89.02 \pm 4.70$ | $87.84 \pm 2.61$ | $87.00 \pm 6.78$ | $86.95 \pm 3.04$ | $85.62 \pm 8.21$ | $86.57 \pm 3.40$ | $84.91 \pm 9.17$ | $86.06 \pm 3.67$ | $84.40 \pm 9.94$ |
| D.S. G-VAE (Gaussian) | $99.67 \pm 0.16$ | $99.91 \pm 0.06$ | $99.14 \pm 0.50$ | $99.83 \pm 0.16$ | $99.18 \pm 0.59$ | $99.82 \pm 0.21$ | $99.19 \pm 0.62$ | $99.77 \pm 0.32$ | $99.15 \pm 0.62$ | $99.71 \pm 0.44$ | $99.15 \pm 0.62$ | $99.65 \pm 0.56$ |
| D.S. G-VAE (CRPS) | $94.58 \pm 1.80$ | $94.24 \pm 1.35$ | $92.95 \pm 1.22$ | $91.33 \pm 3.56$ | $91.43 \pm 1.35$ | $88.47 \pm 4.88$ | $\mathbf{90.46 \pm 1.23}$ | $86.56 \pm 5.55$ | $\mathbf{90.22 \pm 1.24}$ | $85.36 \pm 6.44$ | $\mathbf{90.13 \pm 1.44}$ | $84.44 \pm 7.30$ |
| G-Latent (Gaussian) | $99.24 \pm 0.18$ | $99.77 \pm 0.14$ | $98.37 \pm 0.68$ | $99.46 \pm 0.36$ | $97.75 \pm 0.90$ | $99.20 \pm 0.49$ | $97.24 \pm 1.07$ | $98.98 \pm 0.58$ | $96.80 \pm 1.14$ | $98.76 \pm 0.67$ | $96.41 \pm 1.23$ | $98.54 \pm 0.76$ |
| G-Latent (CRPS) | $\mathbf{92.32 \pm 2.62}$ | $90.52 \pm 2.77$ | $91.21 \pm 1.00$ | $\mathbf{90.80 \pm 1.91}$ | $\mathbf{90.25 \pm 0.70}$ | $\mathbf{90.01 \pm 1.51}$ | $89.52 \pm 0.68$ | $\mathbf{89.32 \pm 1.46}$ | $88.97 \pm 0.87$ | $\mathbf{88.79 \pm 1.53}$ | $88.63 \pm 1.14$ | $\mathbf{88.38 \pm 1.52}$ |

Table 14: Empirical coverage (%) by step and dimension for each quantile $q$. Steps $t' \in \{7, \ldots, 11\}$ and calibration across steps, two outcome dimensions.

| Model | Step $t'$ & Across Steps (Cal.) | | | | | | | | | | Across Steps (Cal.) | |
|---|---|---|---|---|---|---|---|---|---|---|---|---|
| | 7 | | 8 | | 9 | | 10 | | 11 | | | |
| | Dim 1 | Dim 2 | Dim 1 | Dim 2 | Dim 1 | Dim 2 | Dim 1 | Dim 2 | Dim 1 | Dim 2 | Dim 1 | Dim 2 |
| **Quantile $q = 0.1$** | | | | | | | | | | | | |
| G-Net | 17.62±6.07 | 20.20±5.19 | 17.90±6.30 | 20.53±5.11 | 17.95±6.32 | 20.51±4.96 | 18.03±6.28 | 20.46±4.85 | 18.11±6.38 | 20.44±4.75 | 16.94±5.62 | 19.24±4.93 |
| Transformer G-Net | 14.53±8.45 | 24.71±4.14 | 15.12±8.75 | 25.69±4.24 | 15.76±8.98 | 26.30±4.23 | 16.26±9.25 | 26.83±4.25 | 16.65±9.55 | 27.32±4.26 | 14.13±8.14 | 23.24±4.10 |
| CT-CRPS | 21.73±3.11 | 34.41±12.22 | 22.30±3.31 | 35.70±12.81 | 22.79±3.47 | 36.73±13.25 | 23.29±3.64 | 37.70±13.58 | 23.73±3.87 | 38.50±14.05 | 20.89±2.68 | 32.15±11.34 |
| CT-Gaussian | 14.86±6.16 | 24.59±4.66 | 15.51±6.63 | 25.93±4.95 | 16.19±7.25 | 27.03±5.11 | 16.95±7.80 | 28.03±5.46 | 17.54±8.18 | 28.84±5.66 | 14.17±5.60 | 22.86±4.32 |
| D.S. G-VAE (Gaussian) | 0.71±0.43 | 0.13±0.14 | 0.75±0.43 | 0.13±0.14 | 0.79±0.48 | 0.13±0.14 | 0.84±0.49 | 0.12±0.15 | 0.90±0.53 | 0.14±0.15 | 0.74±0.42 | 0.14±0.15 |
| D.S. G-VAE (CRPS) | 15.37±3.71 | 11.89±3.53 | 15.79±3.81 | 12.08±3.88 | 16.01±3.95 | 11.97±4.07 | 16.24±4.27 | 11.80±4.19 | 16.51±4.56 | 11.65±4.32 | 14.10±3.79 | 10.77±2.98 |
| G-Latent (Gaussian) | 3.21±1.30 | 1.28±1.04 | 3.41±1.40 | 1.40±1.05 | 3.57±1.45 | 1.55±1.14 | 3.71±1.54 | 1.64±1.11 | 3.85±1.66 | 1.77±1.11 | 2.92±1.19 | 1.16±0.90 |
| G-Latent (CRPS) | 11.69±2.11 | 13.19±4.05 | 11.89±2.20 | 13.60±4.14 | 12.09±2.36 | 13.80±4.20 | 12.23±2.45 | 14.00±4.14 | 12.40±2.57 | 14.15±4.13 | 11.40±2.05 | 12.59±3.72 |
| **Quantile $q = 0.2$** | | | | | | | | | | | | |
| G-Net | 25.81±6.25 | 28.45±5.88 | 26.06±6.41 | 28.67±5.70 | 26.19±6.64 | 28.61±5.60 | 26.27±6.68 | 28.51±5.54 | 26.36±6.84 | 28.51±5.43 | 25.25±5.85 | 27.54±5.60 |
| Transformer G-Net | 22.77±9.13 | 33.83±4.60 | 23.47±9.34 | 34.83±4.82 | 24.17±9.57 | 35.35±4.97 | 24.62±9.82 | 35.84±5.04 | 25.13±10.11 | 36.32±5.19 | 22.44±8.65 | 32.29±4.36 |
| CT-CRPS | 28.09±3.41 | 41.52±14.34 | 28.62±3.69 | 42.63±15.00 | 29.07±3.79 | 43.52±15.38 | 29.56±4.02 | 44.40±15.68 | 29.96±4.26 | 45.08±16.01 | 27.26±2.98 | 39.42±13.63 |
| CT-Gaussian | 23.06±8.24 | 33.78±4.86 | 23.76±8.72 | 34.97±5.01 | 24.54±9.22 | 35.91±5.29 | 25.25±9.86 | 36.89±5.68 | 25.96±10.26 | 37.67±5.92 | 22.18±7.63 | 32.04±4.75 |
| D.S. G-VAE (Gaussian) | 2.99±0.99 | 0.87±0.70 | 3.17±1.03 | 0.93±0.74 | 3.35±1.05 | 1.02±0.82 | 3.54±1.08 | 1.06±0.82 | 3.69±1.08 | 1.11±0.86 | 2.92±0.94 | 0.87±0.70 |
| D.S. G-VAE (CRPS) | 26.01±4.90 | 20.77±5.83 | 26.49±4.98 | 21.01±6.42 | 26.78±5.39 | 20.77±6.69 | 27.01±5.84 | 20.54±6.93 | 27.19±6.27 | 20.25±7.08 | 24.22±4.91 | 19.39±4.99 |
| G-Latent (Gaussian) | 8.57±2.05 | 4.77±2.38 | 9.05±2.14 | 5.35±2.46 | 9.36±2.24 | 5.74±2.57 | 9.60±2.30 | 6.11±2.61 | 9.87±2.45 | 6.48±2.67 | 7.85±1.89 | 4.35±2.16 |
| G-Latent (CRPS) | 20.49±3.34 | 22.31±4.32 | 20.73±3.58 | 22.83±4.40 | 20.82±3.57 | 22.83±4.46 | 20.94±3.72 | 22.92±4.46 | 21.06±3.77 | 22.94±4.45 | 20.12±3.12 | 21.51±4.12 |
| **Quantile $q = 0.3$** | | | | | | | | | | | | |
| G-Net | 33.22±5.95 | 35.69±6.01 | 33.39±6.13 | 35.89±5.84 | 33.55±6.41 | 35.77±5.72 | 33.65±6.53 | 35.62±5.65 | 33.71±6.71 | 35.57±5.61 | 32.87±5.62 | 35.00±5.75 |
| Transformer G-Net | 30.85±8.99 | 41.31±5.12 | 31.54±9.18 | 42.19±5.48 | 32.15±9.40 | 42.67±5.70 | 32.63±9.67 | 43.10±5.92 | 33.11±9.91 | 43.59±6.16 | 30.70±8.37 | 39.96±4.63 |
| CT-CRPS | 33.73±3.51 | 47.08±15.98 | 34.15±3.84 | 48.01±16.56 | 34.56±3.93 | 48.78±16.97 | 34.96±4.17 | 49.48±17.26 | 35.34±4.37 | 50.06±17.62 | 33.09±3.08 | 45.32±15.32 |
| CT-Gaussian | 31.16±9.58 | 41.88±5.58 | 31.74±10.04 | 42.87±5.94 | 32.55±10.46 | 43.70±6.20 | 33.27±10.97 | 44.50±6.56 | 33.90±11.31 | 45.17±6.83 | 30.27±8.99 | 40.34±5.54 |
| D.S. G-VAE (Gaussian) | 9.38±1.90 | 4.85±2.16 | 9.86±1.93 | 5.20±2.29 | 10.18±2.06 | 5.48±2.52 | 10.58±2.09 | 5.67±2.62 | 10.98±2.13 | 5.93±2.84 | 9.08±1.74 | 4.69±2.15 |
| D.S. G-VAE (CRPS) | 36.16±5.40 | 29.84±8.01 | 36.69±5.90 | 29.99±8.68 | 36.93±6.38 | 29.51±8.99 | 37.13±6.87 | 29.05±9.36 | 37.24±7.42 | 28.62±9.57 | 34.41±5.36 | 28.52±7.00 |
| G-Latent (Gaussian) | 17.66±2.38 | 13.29±3.36 | 18.25±2.44 | 14.22±3.49 | 18.60±2.52 | 14.78±3.56 | 18.89±2.55 | 15.29±3.60 | 19.21±2.71 | 15.78±3.59 | 16.55±2.27 | 12.09±3.19 |
| G-Latent (CRPS) | 29.63±4.28 | 31.95±4.66 | 29.86±4.41 | 32.49±4.56 | 29.91±4.52 | 32.29±4.63 | 29.96±4.54 | 32.22±4.72 | 29.93±4.56 | 32.07±4.76 | 29.48±3.85 | 31.18±4.56 |
| **Quantile $q = 0.4$** | | | | | | | | | | | | |
| G-Net | 40.31±5.57 | 42.66±5.80 | 40.44±5.76 | 42.81±5.64 | 40.53±6.01 | 42.64±5.51 | 40.62±6.23 | 42.44±5.37 | 40.67±6.48 | 42.33±5.37 | 40.20±5.24 | 42.20±5.53 |
| Transformer G-Net | 38.96±8.49 | 47.98±5.60 | 39.54±8.77 | 48.78±6.16 | 40.11±9.01 | 49.19±6.49 | 40.43±9.30 | 49.56±6.78 | 40.85±9.57 | 50.00±7.10 | 39.02±7.82 | 46.95±5.03 |
| CT-CRPS | 39.19±3.48 | 51.94±17.16 | 39.50±3.81 | 52.72±17.72 | 39.78±3.90 | 53.26±18.14 | 40.12±4.10 | 53.86±18.51 | 40.38±4.37 | 54.31±18.78 | 38.78±3.03 | 50.58±16.55 |
| CT-Gaussian | 39.56±10.15 | 49.43±6.99 | 40.09±10.54 | 50.21±7.29 | 40.78±10.83 | 50.81±7.57 | 41.35±11.31 | 51.53±7.91 | 41.98±11.64 | 51.99±8.10 | 38.85±9.63 | 48.21±6.73 |
| D.S. G-VAE (Gaussian) | 23.60±4.66 | 18.98±3.63 | 24.04±4.92 | 19.51±3.85 | 24.43±5.04 | 19.64±4.10 | 24.88±5.13 | 19.74±4.41 | 25.27±5.21 | 19.89±4.86 | 23.21±4.44 | 18.61±3.67 |
| D.S. G-VAE (CRPS) | 45.05±5.50 | 38.91±9.77 | 46.42±6.15 | 38.82±10.51 | 46.42±6.79 | 38.19±11.00 | 46.72±7.39 | 37.55±11.33 | 46.80±7.80 | 36.86±11.59 | 44.61±5.20 | 37.96±8.79 |
| G-Latent (Gaussian) | 31.04±2.06 | 28.12±3.67 | 31.46±2.11 | 29.33±3.79 | 31.65±2.14 | 29.59±3.74 | 31.81±2.32 | 29.90±3.88 | 31.88±2.41 | 30.20±3.90 | 30.07±1.94 | 26.95±3.49 |
| G-Latent (CRPS) | 39.24±4.48 | 41.99±4.90 | 39.33±4.64 | 42.32±4.71 | 39.26±4.75 | 42.06±4.72 | 39.22±4.87 | 41.73±4.86 | 39.07±4.95 | 41.47±4.96 | 39.33±4.02 | 41.31±4.91 |
| **Quantile $q = 0.5$** | | | | | | | | | | | | |
| G-Net | 47.43±5.13 | 49.67±5.29 | 47.42±5.37 | 49.73±5.14 | 47.42±5.55 | 49.52±4.94 | 47.48±5.84 | 49.28±4.83 | 47.50±6.08 | 49.13±4.78 | 47.51±4.78 | 49.45±4.96 |
| Transformer G-Net | 47.14±7.96 | 54.30±6.02 | 47.62±8.24 | 55.05±6.76 | 48.11±8.61 | 55.30±7.18 | 48.35±8.98 | 55.59±7.49 | 48.70±9.26 | 55.92±7.77 | 47.47±7.28 | 53.61±5.47 |
| CT-CRPS | 44.65±3.33 | 56.37±18.10 | 44.84±3.65 | 57.00±18.49 | 44.99±3.70 | 57.39±18.99 | 45.20±3.89 | 57.80±19.33 | 45.43±4.20 | 58.11±19.61 | 44.55±2.87 | 55.48±17.39 |
| CT-Gaussian | 48.39±9.99 | 56.62±8.55 | 48.75±10.28 | 57.18±8.84 | 49.27±10.61 | 57.62±9.03 | 49.71±10.99 | 58.09±9.32 | 50.22±11.30 | 58.45±9.49 | 47.99±9.48 | 55.85±8.06 |
| D.S. G-VAE (Gaussian) | 45.81±7.60 | 46.44±3.78 | 45.79±7.70 | 46.14±3.99 | 45.83±7.97 | 45.38±4.28 | 45.82±8.20 | 44.84±4.59 | 45.81±8.29 | 44.32±4.96 | 46.04±7.29 | 46.49±3.91 |
| D.S. G-VAE (CRPS) | 55.53±5.05 | 47.81±10.99 | 55.80±5.80 | 47.62±11.81 | 56.04±6.44 | 46.74±12.39 | 56.02±7.05 | 45.97±12.83 | 55.99±7.54 | 45.13±13.17 | 54.77±4.43 | 47.51±10.16 |
| G-Latent (Gaussian) | 47.55±1.36 | 48.42±3.72 | 47.48±1.46 | 48.76±3.76 | 47.33±1.61 | 48.33±3.71 | 47.17±1.82 | 48.11±3.58 | 46.91±1.88 | 47.84±3.61 | 47.50±1.05 | 47.93±3.58 |
| G-Latent (CRPS) | 49.23±4.20 | 52.00±4.94 | 49.18±4.43 | 52.21±4.84 | 49.01±4.53 | 51.81±4.81 | 48.66±4.63 | 51.29±4.86 | 48.48±4.81 | 50.93±5.01 | 49.54±3.69 | 51.62±5.12 |
| **Quantile $q = 0.6$** | | | | | | | | | | | | |
| G-Net | 54.67±4.55 | 56.87±4.28 | 54.54±4.81 | 56.89±4.18 | 54.50±5.10 | 56.58±4.02 | 54.46±5.39 | 56.32±3.94 | 54.48±5.61 | 56.16±3.86 | 54.94±4.25 | 56.89±4.02 |
| Transformer G-Net | 55.47±7.41 | 60.53±6.41 | 55.81±7.77 | 61.08±7.12 | 56.11±8.27 | 61.34±7.62 | 56.32±8.57 | 61.55±7.97 | 56.63±8.91 | 61.71±8.30 | 56.04±6.86 | 60.22±5.86 |
| CT-CRPS | 50.26±2.93 | 60.74±18.67 | 50.35±3.34 | 61.01±19.00 | 50.31±3.41 | 61.27±19.47 | 50.41±3.57 | 61.54±19.79 | 50.64±3.86 | 61.73±20.07 | 50.52±2.60 | 60.24±17.83 |
| CT-Gaussian | 57.55±9.08 | 63.41±9.86 | 57.75±9.38 | 63.81±10.17 | 58.10±9.63 | 64.03±10.36 | 58.32±9.93 | 64.38±10.69 | 58.65±10.13 | 64.59±10.83 | 57.63±8.55 | 63.25±9.26 |
| D.S. G-VAE (Gaussian) | 69.55±7.03 | 74.91±3.94 | 69.01±7.30 | 73.94±3.91 | 68.47±7.57 | 72.84±3.91 | 68.15±7.88 | 71.74±3.98 | 67.74±8.21 | 70.77±4.08 | 70.37±6.78 | 75.62±3.79 |
| D.S. G-VAE (CRPS) | 64.75±4.19 | 56.71±11.59 | 64.95±4.98 | 56.26±12.40 | 65.08±5.55 | 55.24±13.11 | 64.98±6.23 | 54.22±13.70 | 64.80±6.69 | 53.40±14.12 | 64.63±3.34 | 57.00±10.92 |
| G-Latent (Gaussian) | 64.48±1.80 | 68.44±3.46 | 63.87±1.84 | 67.99±3.53 | 63.48±1.94 | 67.07±3.17 | 62.91±2.01 | 66.31±3.03 | 62.43±2.05 | 65.65±2.88 | 65.46±1.61 | 69.21±3.47 |
| G-Latent (CRPS) | 59.38±3.57 | 61.85±4.63 | 59.15±3.75 | 61.85±4.59 | 58.85±3.93 | 61.23±4.67 | 58.53±4.13 | 60.66±4.76 | 58.13±4.33 | 60.16±4.92 | 59.89±3.06 | 61.72±4.96 |
| **Quantile $q = 0.7$** | | | | | | | | | | | | |
| G-Net | 62.08±3.83 | 64.38±3.04 | 61.95±4.23 | 64.30±2.94 | 61.80±4.56 | 64.00±2.90 | 61.78±4.83 | 63.68±2.85 | 61.71±4.98 | 63.45±2.79 | 62.62±3.68 | 64.64±2.79 |
| Transformer G-Net | 63.99±6.95 | 66.99±6.68 | 64.22±7.30 | 67.39±7.24 | 64.39±7.87 | 67.45±7.71 | 64.50±8.19 | 67.50±8.05 | 64.75±8.57 | 67.63±8.35 | 64.81±6.50 | 66.99±6.07 |
| CT-CRPS | 56.21±2.53 | 64.99±18.75 | 55.99±2.91 | 65.00±19.16 | 55.91±3.05 | 65.10±19.56 | 56.00±3.20 | 65.23±19.90 | 56.15±3.39 | 65.33±20.22 | 56.82±2.23 | 64.99±17.79 |
| CT-Gaussian | 66.95±7.68 | 70.14±10.80 | 66.93±7.90 | 70.26±11.22 | 67.12±8.11 | 70.25±11.41 | 67.22±8.26 | 70.39±11.72 | 67.44±8.42 | 70.52±11.99 | 67.47±7.08 | 70.45±10.11 |
| D.S. G-VAE (Gaussian) | 86.93±4.06 | 91.90±3.20 | 86.44±4.19 | 91.21±3.28 | 86.06±4.46 | 90.44±3.22 | 85.68±4.68 | 89.70±3.23 | 85.36±4.91 | 88.97±3.17 | 87.70±3.86 | 92.28±2.86 |
| D.S. G-VAE (CRPS) | 73.69±3.24 | 65.51±11.42 | 73.79±3.88 | 64.90±12.23 | 73.73±4.40 | 63.74±13.01 | 73.68±4.89 | 62.70±13.70 | 73.49±5.29 | 61.86±14.26 | 74.06±2.32 | 66.32±10.85 |
| G-Latent (Gaussian) | 79.00±2.21 | 84.42±2.58 | 78.23±2.29 | 83.87±1.84 | 77.61±2.26 | 82.38±2.32 | 77.07±2.22 | 81.47±2.08 | 76.59±2.31 | 80.83±2.01 | 80.28±2.08 | 85.05±2.41 |
| G-Latent (CRPS) | 69.40±2.81 | 71.15±3.92 | 69.10±3.02 | 70.92±3.89 | 68.73±3.23 | 70.33±3.94 | 68.34±3.52 | 69.77±4.13 | 68.04±3.76 | 69.19±4.36 | 70.10±2.40 | 71.29±4.20 |
| **Quantile $q = 0.8$** | | | | | | | | | | | | |
| G-Net | 70.11±3.12 | 72.51±1.72 | 69.94±3.38 | 72.34±1.74 | 69.81±3.74 | 71.94±1.77 | 69.75±4.00 | 71.62±1.92 | 69.73±4.09 | 71.34±1.99 | 70.91±2.99 | 72.94±1.50 |
| Transformer G-Net | 72.97±6.37 | 73.86±6.47 | 73.03±6.76 | 74.07±6.91 | 73.07±7.26 | 73.93±7.40 | 73.11±7.58 | 73.91±7.66 | 73.14±7.98 | 73.95±7.93 | 73.85±6.03 | 74.14±5.93 |
| CT-CRPS | 62.81±2.08 | 69.44±18.29 | 62.35±2.36 | 69.26±18.71 | 62.24±2.64 | 69.22±19.26 | 62.24±2.67 | 69.20±19.66 | 62.36±2.82 | 69.19±19.93 | 63.81±1.75 | 69.99±17.17 |
| CT-Gaussian | 76.45±5.86 | 76.88±11.24 | 76.30±6.16 | 76.70±11.69 | 76.38±6.20 | 76.53±11.97 | 76.36±6.34 | 76.50±12.28 | 76.42±6.44 | 76.47±12.62 | 77.17±5.38 | 77.53±10.36 |
| D.S. G-VAE (Gaussian) | 95.99±1.64 | 98.11±1.74 | 95.75±1.62 | 97.88±1.85 | 95.64±1.66 | 97.61±1.93 | 95.48±1.73 | 97.37±1.98 | 95.35±1.82 | 97.11±2.00 | 96.19±1.54 | 98.19±1.49 |
| D.S. G-VAE (CRPS) | 82.16±2.35 | 74.36±10.33 | 82.16±2.79 | 73.67±11.16 | 82.11±3.23 | 72.53±11.93 | 82.04±3.53 | 71.58±12.66 | 81.89±3.89 | 70.67±13.29 | 82.74±1.73 | 75.47±9.79 |
| G-Latent (Gaussian) | 89.49±1.92 | 93.71±1.54 | 88.92±1.97 | 93.15±1.58 | 88.51±2.03 | 92.59±1.51 | 88.14±2.01 | 91.98±1.46 | 87.79±2.01 | 91.53±1.40 | 90.46±1.76 | 94.23±1.37 |
| G-Latent (CRPS) | 79.11±2.08 | 79.81±2.78 | 78.75±2.26 | 79.52±2.76 | 78.42±2.55 | 78.98±2.85 | 78.11±2.83 | 78.50±3.07 | 77.81±3.06 | 78.00±3.43 | 79.78±1.79 | 80.17±2.95 |
| **Quantile $q = 0.9$** | | | | | | | | | | | | |
| G-Net | 79.76±2.44 | 81.61±1.44 | 79.49±2.51 | 81.35±1.57 | 79.39±2.66 | 80.93±1.63 | 79.38±2.73 | 80.65±1.80 | 79.31±2.91 | 80.38±2.03 | 80.59±2.21 | 82.24±1.21 |
| Transformer G-Net | 82.28±5.22 | 81.74±5.55 | 82.65±5.61 | 81.71±5.95 | 82.55±5.97 | 81.49±6.27 | 82.50±6.28 | 81.38±6.57 | 82.50±6.56 | 81.33±6.80 | 83.50±4.98 | 82.25±5.15 |
| CT-CRPS | 70.83±1.59 | 74.86±16.94 | 70.30±1.80 | 74.44±17.55 | 70.13±2.02 | 74.14±18.19 | 70.06±2.11 | 73.96±18.54 | 70.04±2.20 | 73.82±18.92 | 72.12±1.17 | 75.76±15.64 |
| CT-Gaussian | 85.96±3.83 | 83.92±10.48 | 85.80±4.03 | 83.63±11.06 | 85.89±4.09 | 83.28±11.50 | 85.80±4.08 | 83.01±11.87 | 85.74±4.12 | 82.87±12.17 | 86.60±3.40 | 84.77±9.54 |
| D.S. G-VAE (Gaussian) | 99.15±0.60 | 99.60±0.46 | 99.16±0.55 | 99.54±0.75 | 99.19±0.54 | 99.50±0.82 | 99.11±0.54 | 99.41±0.80 | 99.07±0.55 | 99.43±0.89 | 99.15±0.57 | 99.63±0.56 |
| D.S. G-VAE (CRPS) | 90.05±1.70 | 83.74±8.05 | 90.08±1.99 | 83.04±8.86 | 90.09±2.18 | 82.08±9.64 | 90.06±2.36 | 81.26±10.38 | 89.94±2.57 | 80.54±11.03 | 90.54±1.50 | 84.68±7.54 |
| G-Latent (Gaussian) | 96.12±1.23 | 98.35±0.78 | 95.91±1.22 | 98.14±0.81 | 95.68±1.27 | 97.90±0.86 | 95.53±1.26 | 97.72±0.84 | 95.32±1.25 | 97.47±0.84 | 96.51±1.07 | 98.45±0.69 |
| G-Latent (CRPS) | 88.34±1.27 | 88.04±1.59 | 88.02±1.49 | 87.81±1.57 | 87.85±1.73 | 87.34±1.81 | 87.66±1.92 | 86.93±2.09 | 87.43±2.11 | 86.57±2.29 | 88.79±1.08 | 88.40±1.56 |

## J.2 SEMI-SYNTHETIC DATASET (ORIGINAL VERSION)

In table 15, we show a summary of results for selected steps for the original semi-synthetic dataset with issues regarding the positivity assumption. In table 16, we show the Energy Scores. In tables 17, 18 and 19 we show the KDE-LL for bandwidths 0.2, 0.3 and 0.4, respectively. In table 20, we show the RMSE metrics.

Table 15: Results at selected steps $t' \in \{3, 5, 8, 11\}$ for the semi-synthetic dataset. Metrics: Energy Score (ES $\downarrow$) (per step and across steps), KDE-Loglikelihood (KDE-LL $\uparrow$), and RMSE $\downarrow$.

| Model | $t'=3$ | | | $t'=5$ | | | $t'=8$ | | | $t'=11$ | | | Global |
|---|---|---|---|---|---|---|---|---|---|---|---|---|---|
| | ES$\downarrow$ | KDE-LL$\uparrow$ | RMSE$\downarrow$ | ES$\downarrow$ | KDE-LL$\uparrow$ | RMSE$\downarrow$ | ES$\downarrow$ | KDE-LL$\uparrow$ | RMSE$\downarrow$ | ES$\downarrow$ | KDE-LL$\uparrow$ | RMSE$\downarrow$ | ES$\downarrow$ |
| G-Net | $0.65 \pm 0.08$ | $-2.22 \pm 0.39$ | $0.82 \pm 0.04$ | $0.99 \pm 0.11$ | $-3.54 \pm 0.43$ | $1.02 \pm 0.05$ | $1.27 \pm 0.11$ | $-4.65 \pm 0.44$ | $1.22 \pm 0.06$ | $1.41 \pm 0.14$ | $-5.15 \pm 0.47$ | $1.35 \pm 0.06$ | $3.57 \pm 0.43$ |
| Transformer G-Net | $0.49 \pm 0.08$ | $-1.49 \pm 0.32$ | $0.66 \pm 0.04$ | $0.74 \pm 0.11$ | $-2.42 \pm 0.43$ | $0.80 \pm 0.04$ | $1.10 \pm 0.11$ | $-3.69 \pm 0.39$ | $1.00 \pm 0.06$ | $1.31 \pm 0.14$ | $-4.14 \pm 0.35$ | $1.17 \pm 0.06$ | $2.92 \pm 0.38$ |
| CT (CRPS) | $0.41 \pm 0.06$ | $-1.40 \pm 0.20$ | $0.67 \pm 0.06$ | $0.53 \pm 0.06$ | $-1.86 \pm 0.25$ | $0.80 \pm 0.05$ | $0.65 \pm 0.06$ | $-2.29 \pm 0.24$ | $0.94 \pm 0.06$ | $0.73 \pm 0.05$ | $-2.60 \pm 0.22$ | $1.05 \pm 0.06$ | $1.85 \pm 0.22$ |
| CT (Gaussian) | $0.52 \pm 0.07$ | $-1.56 \pm 0.32$ | $0.64 \pm 0.06$ | $0.65 \pm 0.06$ | $-1.81 \pm 0.30$ | $0.78 \pm 0.05$ | $0.82 \pm 0.07$ | $-2.19 \pm 0.29$ | $0.91 \pm 0.05$ | $0.93 \pm 0.07$ | $-2.52 \pm 0.28$ | $1.03 \pm 0.06$ | $2.40 \pm 0.28$ |
| CT | ... | ... | $\mathbf{0.46 \pm 0.01}$ | ... | ... | $\mathbf{0.51 \pm 0.02}$ | ... | ... | $\mathbf{0.55 \pm 0.02}$ | ... | ... | $\mathbf{0.61 \pm 0.02}$ | ... |
| D.S. G-VAE (Gaussian) | $0.49 \pm 0.05$ | $-2.30 \pm 0.30$ | $0.69 \pm 0.05$ | $0.60 \pm 0.05$ | $-2.66 \pm 0.32$ | $0.88 \pm 0.07$ | $0.72 \pm 0.06$ | $-2.91 \pm 0.35$ | $1.18 \pm 0.08$ | $0.78 \pm 0.07$ | $-3.02 \pm 0.32$ | $1.35 \pm 0.08$ | $2.21 \pm 0.24$ |
| D.S. G-VAE (CRPS) | $0.44 \pm 0.06$ | $-1.57 \pm 0.25$ | $0.68 \pm 0.06$ | $0.51 \pm 0.05$ | $-1.80 \pm 0.22$ | $0.85 \pm 0.06$ | $0.58 \pm 0.07$ | $-2.04 \pm 0.24$ | $1.10 \pm 0.07$ | $0.65 \pm 0.08$ | $-2.24 \pm 0.21$ | $1.26 \pm 0.09$ | $1.85 \pm 0.21$ |
| G-Latent (Gaussian) | $0.40 \pm 0.04$ | $-1.48 \pm 0.31$ | $0.62 \pm 0.05$ | $\mathbf{0.46 \pm 0.04}$ | $-1.66 \pm 0.26$ | $0.70 \pm 0.05$ | $\mathbf{0.51 \pm 0.05}$ | $\mathbf{-1.81 \pm 0.24}$ | $0.78 \pm 0.05$ | $\mathbf{0.54 \pm 0.05}$ | $\mathbf{-1.91 \pm 0.24}$ | $0.83 \pm 0.07$ | $\mathbf{1.64 \pm 0.13}$ |
| G-Latent (CRPS) | $\mathbf{0.39 \pm 0.06}$ | $\mathbf{-1.32 \pm 0.15}$ | $0.65 \pm 0.06$ | $\mathbf{0.46 \pm 0.06}$ | $\mathbf{-1.59 \pm 0.16}$ | $0.77 \pm 0.06$ | $0.53 \pm 0.06$ | $-1.82 \pm 0.15$ | $0.88 \pm 0.04$ | $0.56 \pm 0.05$ | $-1.95 \pm 0.14$ | $0.94 \pm 0.03$ | $1.67 \pm 0.20$ |

Table 16: Energy Score per step $t'$ on semi-synthetic dataset. Rightmost column reports the Global Energy Score across steps. Best per column in **bold**.

| Model | $t'=2$ | $t'=3$ | $t'=4$ | $t'=5$ | $t'=6$ | $t'=7$ | $t'=8$ | $t'=9$ | $t'=10$ | $t'=11$ | Global |
|---|---|---|---|---|---|---|---|---|---|---|---|
| G-Net | $0.44 \pm 0.06$ | $0.65 \pm 0.08$ | $0.84 \pm 0.09$ | $0.99 \pm 0.11$ | $1.11 \pm 0.11$ | $1.20 \pm 0.13$ | $1.27 \pm 0.11$ | $1.33 \pm 0.13$ | $1.38 \pm 0.13$ | $1.41 \pm 0.14$ | $3.57 \pm 0.43$ |
| Transformer G-Net | $0.39 \pm 0.06$ | $0.49 \pm 0.08$ | $0.62 \pm 0.09$ | $0.74 \pm 0.11$ | $0.90 \pm 0.11$ | $1.03 \pm 0.13$ | $1.10 \pm 0.11$ | $1.19 \pm 0.13$ | $1.25 \pm 0.13$ | $1.31 \pm 0.14$ | $2.92 \pm 0.38$ |
| CT-CRPS | $0.35 \pm 0.05$ | $0.41 \pm 0.06$ | $0.49 \pm 0.06$ | $0.53 \pm 0.06$ | $0.58 \pm 0.06$ | $0.62 \pm 0.06$ | $0.65 \pm 0.06$ | $0.68 \pm 0.06$ | $0.71 \pm 0.06$ | $0.73 \pm 0.05$ | $1.85 \pm 0.22$ |
| CT-Gaussian | $0.42 \pm 0.05$ | $0.52 \pm 0.07$ | $0.58 \pm 0.07$ | $0.65 \pm 0.06$ | $0.72 \pm 0.06$ | $0.75 \pm 0.07$ | $0.82 \pm 0.07$ | $0.88 \pm 0.08$ | $0.91 \pm 0.07$ | $0.93 \pm 0.07$ | $2.40 \pm 0.28$ |
| D.S. G-VAE (Gaussian) | $0.40 \pm 0.05$ | $0.49 \pm 0.05$ | $0.54 \pm 0.04$ | $0.60 \pm 0.05$ | $0.67 \pm 0.06$ | $0.70 \pm 0.06$ | $0.72 \pm 0.06$ | $0.74 \pm 0.05$ | $0.76 \pm 0.07$ | $0.78 \pm 0.07$ | $2.21 \pm 0.24$ |
| D.S. G-VAE (CRPS) | $0.38 \pm 0.05$ | $0.44 \pm 0.06$ | $0.48 \pm 0.06$ | $0.51 \pm 0.05$ | $0.54 \pm 0.06$ | $0.56 \pm 0.06$ | $0.58 \pm 0.07$ | $0.61 \pm 0.06$ | $0.63 \pm 0.07$ | $0.65 \pm 0.08$ | $1.85 \pm 0.21$ |
| G-Latent (Gaussian) | $0.36 \pm 0.04$ | $0.40 \pm 0.04$ | $\mathbf{0.43 \pm 0.04}$ | $\mathbf{0.46 \pm 0.04}$ | $\mathbf{0.48 \pm 0.04}$ | $\mathbf{0.49 \pm 0.05}$ | $\mathbf{0.51 \pm 0.05}$ | $\mathbf{0.52 \pm 0.05}$ | $\mathbf{0.53 \pm 0.05}$ | $\mathbf{0.54 \pm 0.05}$ | $\mathbf{1.64 \pm 0.13}$ |
| G-Latent (CRPS) | $\mathbf{0.34 \pm 0.05}$ | $\mathbf{0.39 \pm 0.06}$ | $\mathbf{0.43 \pm 0.06}$ | $\mathbf{0.46 \pm 0.06}$ | $0.49 \pm 0.06$ | $0.51 \pm 0.06$ | $0.53 \pm 0.06$ | $0.54 \pm 0.06$ | $0.55 \pm 0.06$ | $0.56 \pm 0.05$ | $1.67 \pm 0.20$ |

Table 17: KDE Loglikelihood per step $t'$ on semi-synthetic dataset with bandwidth 0.2. Best per column in **bold**.

| Model | $t'=2$ | $t'=3$ | $t'=4$ | $t'=5$ | $t'=6$ | $t'=7$ | $t'=8$ | $t'=9$ | $t'=10$ | $t'=11$ |
|---|---|---|---|---|---|---|---|---|---|---|
| G-Net | $-1.68 \pm 0.38$ | $-3.09 \pm 0.45$ | $-4.43 \pm 0.41$ | $-5.71 \pm 0.61$ | $-6.83 \pm 0.56$ | $-7.72 \pm 0.40$ | $-8.41 \pm 0.73$ | $-8.99 \pm 0.60$ | $-9.38 \pm 0.69$ | $-9.69 \pm 0.71$ |
| Transformer G-Net | $-1.24 \pm 0.31$ | $-2.01 \pm 0.39$ | $-2.68 \pm 0.32$ | $-3.31 \pm 0.43$ | $-3.80 \pm 0.50$ | $-4.79 \pm 0.73$ | $-5.63 \pm 0.56$ | $-6.41 \pm 0.68$ | $-7.12 \pm 0.81$ | $-7.88 \pm 0.70$ |
| CT-CRPS | $-1.13 \pm 0.20$ | $-1.44 \pm 0.23$ | $-1.69 \pm 0.32$ | $-1.86 \pm 0.30$ | $-2.01 \pm 0.37$ | $-2.21 \pm 0.35$ | $-2.38 \pm 0.39$ | $-2.53 \pm 0.45$ | $-2.67 \pm 0.31$ | $-2.80 \pm 0.29$ |
| CT-Gaussian | $-1.25 \pm 0.22$ | $-1.67 \pm 0.27$ | $-1.81 \pm 0.33$ | $-1.95 \pm 0.37$ | $-2.10 \pm 0.29$ | $-2.29 \pm 0.33$ | $-2.52 \pm 0.25$ | $-2.72 \pm 0.30$ | $-2.89 \pm 0.36$ | $-3.02 \pm 0.41$ |
| D.S. G-VAE (Gaussian) | $-1.90 \pm 0.34$ | $-2.27 \pm 0.31$ | $-2.51 \pm 0.45$ | $-2.67 \pm 0.39$ | $-2.79 \pm 0.31$ | $-2.87 \pm 0.44$ | $-2.94 \pm 0.41$ | $-2.99 \pm 0.48$ | $-3.04 \pm 0.51$ | $-3.07 \pm 0.43$ |
| D.S. G-VAE (CRPS) | $-1.26 \pm 0.25$ | $-1.51 \pm 0.32$ | $-1.70 \pm 0.21$ | $-1.82 \pm 0.27$ | $-1.95 \pm 0.31$ | $-2.06 \pm 0.30$ | $-2.16 \pm 0.39$ | $-2.25 \pm 0.35$ | $-2.33 \pm 0.42$ | $-2.40 \pm 0.29$ |
| G-Latent (Gaussian) | $-1.30 \pm 0.29$ | $-1.53 \pm 0.18$ | $-1.71 \pm 0.13$ | $-1.83 \pm 0.11$ | $-1.93 \pm 0.14$ | $-2.02 \pm 0.19$ | $-2.09 \pm 0.25$ | $-2.15 \pm 0.30$ | $-2.20 \pm 0.36$ | $-2.26 \pm 0.41$ |
| G-Latent (CRPS) | $\mathbf{-0.97 \pm 0.24}$ | $\mathbf{-1.31 \pm 0.27}$ | $\mathbf{-1.55 \pm 0.29}$ | $\mathbf{-1.72 \pm 0.30}$ | $\mathbf{-1.87 \pm 0.31}$ | $\mathbf{-1.98 \pm 0.31}$ | $\mathbf{-2.07 \pm 0.31}$ | $\mathbf{-2.14 \pm 0.30}$ | $\mathbf{-2.19 \pm 0.30}$ | $\mathbf{-2.25 \pm 0.29}$ |

Table 18: KDE Loglikelihood per step $t'$ on semi-synthetic dataset with bandwidth 0.3 Best per column in **bold**.

| Model | $t'=2$ | $t'=3$ | $t'=4$ | $t'=5$ | $t'=6$ | $t'=7$ | $t'=8$ | $t'=9$ | $t'=10$ | $t'=11$ |
|---|---|---|---|---|---|---|---|---|---|---|
| G-Net | $-1.41 \pm 0.32$ | $-2.47 \pm 0.39$ | $-3.37 \pm 0.34$ | $-4.15 \pm 0.46$ | $-4.80 \pm 0.49$ | $-5.29 \pm 0.54$ | $-5.68 \pm 0.40$ | $-5.99 \pm 0.46$ | $-6.21 \pm 0.41$ | $-6.37 \pm 0.51$ |
| Transformer G-Net | $-1.23 \pm 0.30$ | $-1.83 \pm 0.39$ | $-2.37 \pm 0.39$ | $-3.12 \pm 0.41$ | $-3.81 \pm 0.37$ | $-4.43 \pm 0.41$ | $-4.44 \pm 0.39$ | $-4.68 \pm 0.38$ | $-5.25 \pm 0.32$ | $-5.77 \pm 0.34$ |
| CT-CRPS | $-1.02 \pm 0.22$ | $-1.29 \pm 0.18$ | $-1.52 \pm 0.24$ | $-1.71 \pm 0.25$ | $-1.88 \pm 0.24$ | $-2.01 \pm 0.21$ | $-2.14 \pm 0.22$ | $-2.22 \pm 0.24$ | $-2.26 \pm 0.22$ | $-2.30 \pm 0.23$ |
| CT-Gaussian | $-1.29 \pm 0.31$ | $-1.49 \pm 0.38$ | $-1.77 \pm 0.32$ | $-1.98 \pm 0.39$ | $-2.17 \pm 0.30$ | $-2.30 \pm 0.37$ | $-2.46 \pm 0.34$ | $-2.59 \pm 0.29$ | $-2.72 \pm 0.28$ | $-2.85 \pm 0.31$ |
| D.S. G-VAE (Gaussian) | $-1.92 \pm 0.29$ | $-2.27 \pm 0.31$ | $-2.49 \pm 0.34$ | $-2.65 \pm 0.31$ | $-2.76 \pm 0.37$ | $-2.85 \pm 0.29$ | $-2.91 \pm 0.32$ | $-2.96 \pm 0.34$ | $-3.00 \pm 0.35$ | $-3.03 \pm 0.38$ |
| D.S. G-VAE (CRPS) | $-1.29 \pm 0.20$ | $-1.50 \pm 0.25$ | $-1.66 \pm 0.23$ | $-1.77 \pm 0.29$ | $-1.87 \pm 0.22$ | $-1.97 \pm 0.24$ | $-2.05 \pm 0.25$ | $-2.12 \pm 0.31$ | $-2.19 \pm 0.27$ | $-2.26 \pm 0.27$ |
| G-Latent (Gaussian) | $-1.25 \pm 0.37$ | $-1.43 \pm 0.30$ | $-1.56 \pm 0.25$ | $-1.65 \pm 0.22$ | $-1.73 \pm 0.21$ | $-1.80 \pm 0.20$ | $-1.85 \pm 0.20$ | $-1.89 \pm 0.20$ | $-1.93 \pm 0.21$ | $-1.97 \pm 0.23$ |
| G-Latent (CRPS) | $\mathbf{-0.98 \pm 0.18}$ | $\mathbf{-1.25 \pm 0.20}$ | $\mathbf{-1.44 \pm 0.21}$ | $\mathbf{-1.58 \pm 0.21}$ | $\mathbf{-1.69 \pm 0.21}$ | $\mathbf{-1.78 \pm 0.21}$ | $\mathbf{-1.85 \pm 0.20}$ | $\mathbf{-1.91 \pm 0.20}$ | $\mathbf{-1.96 \pm 0.19}$ | $\mathbf{-2.00 \pm 0.19}$ |

Table 19: KDE Loglikelihood per step $t'$ on semi-synthetic dataset with bandwidth 0.4. Best per column in **bold**.

| Model | $t'=2$ | $t'=3$ | $t'=4$ | $t'=5$ | $t'=6$ | $t'=7$ | $t'=8$ | $t'=9$ | $t'=10$ | $t'=11$ |
|---|---|---|---|---|---|---|---|---|---|---|
| G-Net | $-1.37 \pm 0.32$ | $-2.22 \pm 0.39$ | $-2.94 \pm 0.39$ | $-3.54 \pm 0.43$ | $-4.01 \pm 0.44$ | $-4.38 \pm 0.42$ | $-4.65 \pm 0.44$ | $-4.88 \pm 0.45$ | $-5.04 \pm 0.40$ | $-5.15 \pm 0.47$ |
| Transformer G-Net | $-1.21 \pm 0.30$ | $-1.49 \pm 0.32$ | $-1.89 \pm 0.40$ | $-2.42 \pm 0.43$ | $-2.91 \pm 0.44$ | $-3.38 \pm 0.45$ | $-3.69 \pm 0.39$ | $-3.85 \pm 0.37$ | $-4.01 \pm 0.40$ | $-4.14 \pm 0.35$ |
| CT-CRPS | $-1.15 \pm 0.19$ | $-1.40 \pm 0.20$ | $-1.62 \pm 0.24$ | $-1.86 \pm 0.25$ | $-1.99 \pm 0.26$ | $-2.14 \pm 0.24$ | $-2.29 \pm 0.24$ | $-2.35 \pm 0.22$ | $-2.48 \pm 0.23$ | $-2.60 \pm 0.22$ |
| CT-Gaussian | $-1.41 \pm 0.31$ | $-1.56 \pm 0.32$ | $-1.68 \pm 0.34$ | $-1.81 \pm 0.30$ | $-1.95 \pm 0.29$ | $-2.06 \pm 0.31$ | $-2.19 \pm 0.29$ | $-2.33 \pm 0.28$ | $-2.38 \pm 0.28$ | $-2.52 \pm 0.28$ |
| D.S. G-VAE (Gaussian) | $-1.97 \pm 0.28$ | $-2.30 \pm 0.30$ | $-2.51 \pm 0.28$ | $-2.66 \pm 0.32$ | $-2.77 \pm 0.31$ | $-2.85 \pm 0.31$ | $-2.91 \pm 0.35$ | $-2.96 \pm 0.33$ | $-2.99 \pm 0.32$ | $-3.02 \pm 0.32$ |
| D.S. G-VAE (CRPS) | $-1.39 \pm 0.22$ | $-1.57 \pm 0.25$ | $-1.70 \pm 0.22$ | $-1.80 \pm 0.22$ | $-1.89 \pm 0.22$ | $-1.97 \pm 0.25$ | $-2.04 \pm 0.24$ | $-2.11 \pm 0.22$ | $-2.18 \pm 0.21$ | $-2.24 \pm 0.21$ |
| G-Latent (Gaussian) | $-1.34 \pm 0.36$ | $-1.48 \pm 0.31$ | $-1.59 \pm 0.28$ | $-1.66 \pm 0.26$ | $-1.72 \pm 0.25$ | $-1.77 \pm 0.24$ | $-1.81 \pm 0.24$ | $-1.85 \pm 0.24$ | $-1.88 \pm 0.24$ | $-1.91 \pm 0.24$ |
| G-Latent (CRPS) | $\mathbf{-1.10 \pm 0.14}$ | $\mathbf{-1.32 \pm 0.15}$ | $\mathbf{-1.48 \pm 0.16}$ | $\mathbf{-1.59 \pm 0.16}$ | $\mathbf{-1.69 \pm 0.16}$ | $\mathbf{-1.76 \pm 0.16}$ | $\mathbf{-1.82 \pm 0.15}$ | $\mathbf{-1.87 \pm 0.15}$ | $\mathbf{-1.91 \pm 0.14}$ | $\mathbf{-1.95 \pm 0.14}$ |

Table 20: RMSE per step $t'$ on semi-synthetic dataset. Best per column in **bold**.

| Model | $t'=2$ | $t'=3$ | $t'=4$ | $t'=5$ | $t'=6$ | $t'=7$ | $t'=8$ | $t'=9$ | $t'=10$ | $t'=11$ |
|---|---|---|---|---|---|---|---|---|---|---|
| G-Net | $0.67 \pm 0.03$ | $0.82 \pm 0.04$ | $0.96 \pm 0.04$ | $1.02 \pm 0.05$ | $1.09 \pm 0.05$ | $1.18 \pm 0.05$ | $1.22 \pm 0.06$ | $1.25 \pm 0.06$ | $1.29 \pm 0.06$ | $1.35 \pm 0.06$ |
| Transformer G-Net | $0.59 \pm 0.03$ | $0.66 \pm 0.04$ | $0.73 \pm 0.04$ | $0.80 \pm 0.04$ | $0.86 \pm 0.05$ | $0.92 \pm 0.05$ | $1.00 \pm 0.06$ | $1.06 \pm 0.06$ | $1.11 \pm 0.06$ | $1.17 \pm 0.06$ |
| CT-Gaussian | $0.54 \pm 0.05$ | $0.64 \pm 0.06$ | $0.72 \pm 0.06$ | $0.78 \pm 0.05$ | $0.84 \pm 0.05$ | $0.88 \pm 0.06$ | $0.91 \pm 0.05$ | $0.95 \pm 0.05$ | $0.99 \pm 0.06$ | $1.03 \pm 0.06$ |
| CT-CRPS | $0.55 \pm 0.05$ | $0.67 \pm 0.06$ | $0.76 \pm 0.06$ | $0.80 \pm 0.05$ | $0.85 \pm 0.05$ | $0.91 \pm 0.06$ | $0.94 \pm 0.06$ | $0.97 \pm 0.06$ | $1.02 \pm 0.05$ | $1.05 \pm 0.06$ |
| CT | $\mathbf{0.37 \pm 0.01}$ | $\mathbf{0.46 \pm 0.01}$ | $\mathbf{0.49 \pm 0.01}$ | $\mathbf{0.51 \pm 0.02}$ | $\mathbf{0.53 \pm 0.02}$ | $\mathbf{0.54 \pm 0.02}$ | $\mathbf{0.55 \pm 0.02}$ | $\mathbf{0.58 \pm 0.02}$ | $\mathbf{0.60 \pm 0.02}$ | $\mathbf{0.61 \pm 0.02}$ |
| D.S. G-VAE (Gaussian) | $0.56 \pm 0.06$ | $0.69 \pm 0.05$ | $0.79 \pm 0.06$ | $0.88 \pm 0.07$ | $0.97 \pm 0.07$ | $1.09 \pm 0.07$ | $1.18 \pm 0.08$ | $1.24 \pm 0.09$ | $1.30 \pm 0.08$ | $1.35 \pm 0.08$ |
| D.S. G-VAE (CRPS) | $0.57 \pm 0.05$ | $0.68 \pm 0.06$ | $0.77 \pm 0.06$ | $0.85 \pm 0.06$ | $0.93 \pm 0.08$ | $1.02 \pm 0.06$ | $1.10 \pm 0.07$ | $1.16 \pm 0.09$ | $1.22 \pm 0.08$ | $1.26 \pm 0.09$ |
| G-Latent (Gaussian) | $0.54 \pm 0.05$ | $0.62 \pm 0.05$ | $0.67 \pm 0.05$ | $0.70 \pm 0.05$ | $0.73 \pm 0.05$ | $0.76 \pm 0.05$ | $0.78 \pm 0.05$ | $0.79 \pm 0.06$ | $0.81 \pm 0.07$ | $0.83 \pm 0.07$ |
| G-Latent (CRPS) | $0.56 \pm 0.06$ | $0.65 \pm 0.06$ | $0.72 \pm 0.06$ | $0.77 \pm 0.06$ | $0.81 \pm 0.05$ | $0.85 \pm 0.05$ | $0.88 \pm 0.04$ | $0.90 \pm 0.04$ | $0.92 \pm 0.03$ | $0.94 \pm 0.03$ |

## J.3 REAL WORLD DATASET

In table 21, we show the Energy Scores. In table 22, we show the KDE-LL metric for bandwidth 3.6. In table 23, we show the RMSE metrics.

Table 21: Energy Score per step $t'$ on real-world dataset. Rightmost column reports the Global Energy Score across steps. Best per column in **bold**.

| Model | $t'=2$ | $t'=3$ | $t'=4$ | $t'=5$ | $t'=6$ | Global |
|---|---|---|---|---|---|---|
| G-Net | $5.32 \pm 0.08$ | $5.82 \pm 0.08$ | $6.29 \pm 0.08$ | $6.98 \pm 0.09$ | $7.44 \pm 0.11$ | $18.35 \pm 0.33$ |
| Transformer G-Net | $5.28 \pm 0.06$ | $5.84 \pm 0.08$ | $6.17 \pm 0.09$ | $6.47 \pm 0.08$ | $6.90 \pm 0.08$ | $16.70 \pm 0.23$ |
| CT-CRPS | $4.92 \pm 0.06$ | $5.39 \pm 0.08$ | $5.60 \pm 0.07$ | $5.77 \pm 0.08$ | $5.86 \pm 0.07$ | $14.61 \pm 0.27$ |
| CT-Gaussian | $5.25 \pm 0.06$ | $5.71 \pm 0.08$ | $5.99 \pm 0.08$ | $6.15 \pm 0.07$ | $6.34 \pm 0.08$ | $15.55 \pm 0.23$ |
| D.S. G-VAE (Gaussian) | $5.51 \pm 0.08$ | $5.99 \pm 0.08$ | $6.21 \pm 0.10$ | $6.34 \pm 0.06$ | $6.44 \pm 0.07$ | $15.98 \pm 0.23$ |
| D.S. G-VAE (CRPS) | $4.89 \pm 0.08$ | $5.36 \pm 0.08$ | $5.56 \pm 0.09$ | $5.70 \pm 0.07$ | $5.82 \pm 0.06$ | $14.38 \pm 0.19$ |
| G-Latent (Gaussian) | $5.27 \pm 0.06$ | $5.64 \pm 0.08$ | $5.84 \pm 0.09$ | $5.96 \pm 0.07$ | $6.07 \pm 0.07$ | $15.21 \pm 0.26$ |
| G-Latent (CRPS) | $\mathbf{4.85 \pm 0.05}$ | $\mathbf{5.25 \pm 0.08}$ | $\mathbf{5.47 \pm 0.06}$ | $\mathbf{5.60 \pm 0.09}$ | $\mathbf{5.72 \pm 0.06}$ | $\mathbf{14.23 \pm 0.23}$ |

Table 22: KDE Loglikelihood per step $t'$ on real-world dataset with bandwidth 3.6. Best per column in **bold**.

| Model | $t'=2$ | $t'=3$ | $t'=4$ | $t'=5$ | $t'=6$ |
|---|---|---|---|---|---|
| G-Net | $-3.92 \pm 0.05$ | $-4.11 \pm 0.05$ | $-4.29 \pm 0.06$ | $-4.55 \pm 0.07$ | $-4.83 \pm 0.04$ |
| Transformer G-Net | $-3.89 \pm 0.06$ | $-4.06 \pm 0.08$ | $-4.16 \pm 0.06$ | $-4.30 \pm 0.06$ | $-4.48 \pm 0.04$ |
| CT-CRPS | $-3.81 \pm 0.06$ | $-3.94 \pm 0.06$ | $-3.99 \pm 0.07$ | $-4.08 \pm 0.04$ | $-4.19 \pm 0.06$ |
| CT-Gaussian | $-3.92 \pm 0.06$ | $-4.04 \pm 0.07$ | $-4.09 \pm 0.06$ | $-4.18 \pm 0.06$ | $-4.24 \pm 0.07$ |
| D.S. G-VAE (Gaussian) | $-3.90 \pm 0.06$ | $-3.98 \pm 0.06$ | $-4.01 \pm 0.05$ | $-4.03 \pm 0.05$ | $-4.04 \pm 0.05$ |
| D.S. G-VAE (CRPS) | $-3.82 \pm 0.06$ | $-3.92 \pm 0.05$ | $-3.94 \pm 0.05$ | $-3.99 \pm 0.06$ | $-4.04 \pm 0.06$ |
| G-Latent (Gaussian) | $-3.85 \pm 0.06$ | $-3.89 \pm 0.06$ | $-3.92 \pm 0.05$ | $\mathbf{-3.94 \pm 0.04}$ | $\mathbf{-3.95 \pm 0.06}$ |
| G-Latent (CRPS) | $\mathbf{-3.79 \pm 0.06}$ | $\mathbf{-3.88 \pm 0.05}$ | $\mathbf{-3.91 \pm 0.05}$ | $\mathbf{-3.94 \pm 0.05}$ | $-3.96 \pm 0.06$ |

Table 23: RMSE per step $t'$ on real-world dataset. Best per column in **bold**.

| Model | $t'=2$ | $t'=3$ | $t'=4$ | $t'=5$ | $t'=6$ |
|---|---|---|---|---|---|
| G-Net | $11.84 \pm 0.24$ | $12.83 \pm 0.29$ | $13.54 \pm 0.33$ | $14.05 \pm 0.30$ | $14.23 \pm 0.29$ |
| Transformer G-Net | $10.90 \pm 0.30$ | $11.67 \pm 0.26$ | $12.39 \pm 0.38$ | $12.96 \pm 0.32$ | $13.21 \pm 0.29$ |
| CT-CRPS | $9.34 \pm 0.25$ | $10.10 \pm 0.29$ | $10.53 \pm 0.26$ | $10.75 \pm 0.29$ | $10.91 \pm 0.28$ |
| CT-Gaussian | $9.63 \pm 0.25$ | $10.41 \pm 0.29$ | $10.74 \pm 0.29$ | $11.01 \pm 0.34$ | $11.25 \pm 0.30$ |
| CT | $\mathbf{9.00 \pm 0.23}$ | $\mathbf{9.57 \pm 0.24}$ | $\mathbf{9.90 \pm 0.25}$ | $\mathbf{10.16 \pm 0.27}$ | $\mathbf{10.35 \pm 0.31}$ |
| D.S. G-VAE (Gaussian) | $9.58 \pm 0.25$ | $10.29 \pm 0.22$ | $10.66 \pm 0.29$ | $10.88 \pm 0.26$ | $11.04 \pm 0.29$ |
| D.S. G-VAE (CRPS) | $9.40 \pm 0.22$ | $10.09 \pm 0.25$ | $10.41 \pm 0.23$ | $10.63 \pm 0.29$ | $10.79 \pm 0.30$ |
| G-Latent (Gaussian) | $9.42 \pm 0.23$ | $10.09 \pm 0.23$ | $10.43 \pm 0.25$ | $10.64 \pm 0.19$ | $10.80 \pm 0.25$ |
| G-Latent (CRPS) | $9.23 \pm 0.20$ | $9.79 \pm 0.24$ | $10.14 \pm 0.23$ | $10.36 \pm 0.29$ | $10.55 \pm 0.28$ |

# K  LLMS USAGE

We used LLMs for diverse tasks in the production of this work. Mainly, for text and math reviewing and correction. To a lesser extent, for discussing ideas.

