# OpenReview forum: "Latent g-Computation for Potential Outcomes Distributional Estimation under Time-Varying Treatments"
_ICLR.cc/2026/Conference — Submitted to ICLR 2026_

### Official Review · Reviewer_35oK · 2025-10-15

**Soundness:** 4
**Presentation:** 2
**Contribution:** 2
**Rating:** 4
**Confidence:** 2

**Summary:**

The paper introduces G-Latent, an ML framework for time-varying causal inference. The main architectural novelty lies in the latent rollout technique used, which enables more computationally efficient and (potentially) accurate inference. Theoretical and empirical results were provided.

**Strengths:**

The topic of the paper addresses the timely topic of integrating deep machine learning models into time-varying causal inference.

The paper is technically sound, and the modeling approach was analyzed in depth.

The experiment results are clean and fair. It shows that the proposed framework can make accurate counterfactual distribution estimation and enjoys good computational efficiency.

I appreciate the efforts of the authors in identifying potential limitations of their proposed approach.

**Weaknesses:**

Presentation: The paper is heavily loaded with technical details but lacks explanatory text. This decreases the readability of the paper. Theoretical results lack intuitive interpretations. Some notations (e.g., scope) appear without a formal definition.

Limited architecture novelty: the core novelty of the framework seems to lie in replacing the original data-space autoregression with latent GRU updates. This seems to be a variant of existing VAE-based / representation learning-based causal inference methods, rather than an architecture innovation.

Assumption restrictiveness: The paper's theoretical contribution is somewhat limited, as the analysis relies on the "latent factorization and context sufficiency" assumption, which postulates that the learned embeddings provide good approximations. This assumption is rather heuristic and was not validated via theoretical derivation or numerical experiments. Its informal presentation (without being stated as a numbered assumption) makes the exposition appear somewhat evasive.

Limited experiment scope: The two datasets used in the experiments are both ICU datasets. The method's application potential to other settings and other data types (e.g., images) remains unknown. Additionally, the code is not provided.

Jargon inconsistency: Some jargon used in the paper is not consistent. For example, "A" was referred to as treatment, decision, and plan.

**Questions:**

What is the definition of scope?

Why is the latent rollout idea novel?

Why was the ICU dataset chosen in the experiments? How would the framework generalize to other types of datasets? Can the modeling assumption be empirically validated?

---

> ### Author Response · Authors · 2025-11-18
>
> We thank the reviewer for the careful reading of our paper and for the positive assessment of its technical soundness and of the cleanliness and fairness of the experimental results. We are grateful for the constructive suggestions and address each concern below, in particular the scope of the empirical evaluation.
>
> **Novelty**:
> We apologize if our writing suggested that the contribution is mainly architectural. As detailed in our general comment on novelty, our main contribution is at the estimator level, not just at the module level. Briefly: (i) we introduce a **latent-space g-computation estimator** for individualized potential outcome trajectories, and show that, under the standard identification assumptions and latent factorization/context-sufficiency assumption, it targets the same interventional distribution as the standard discrete-time g-formula while operating entirely in latent space; (ii) we derive a **tighter total-variation error-propagation bound** for the latent rollout than for data-space autoregressive rollouts, which explains the improved performance we observe at longer horizons; and (iii) beyond this estimator-level contribution, we add several practically useful design choices: a conditional VAE head that improves G-Net’s residual-pool mechanism, an ALD-based mixture head for better-calibrated distributional predictions, and a decoupled encoder with a lightweight latent GRU rollout and selective decoding that significantly speeds up Monte Carlo inference.
>
> While we agree that (iii) consists of a set of practically important but incremental improvements, we believe that (i) and (ii) constitute substantive estimator-level advances that are suitable for ICLR.
>
> We updated our paper to make our contributions clearer.
>
> **Assumptions**:
>
> We appreciate this concern and clarify it next.
>
> **What the assumption actually says.**
>
> The ``latent factorization and context sufficiency'' condition (Eq.(7), previously Eq.(8)) states that the true one-step conditional $p^\star(\ell_{t+t'} \mid \bar h_{t+t'-1}, a_{t+t'-1})$ can be represented as a latent mixture with respect to a fixed prior $p_0(z_{t,t'})$, with mixture components parameterized by our decoder given context $c_{t,t'}(r_t, s_{t,t'-1}, a_{t+t'-1}, t')$.
> Intuitively, this is the standard modeling assumption made by conditional VAEs: the decoder family is rich enough to approximate the true conditional density given a sufficiently informative context. It is not an additional causal identifiability assumption beyond the usual sequential ignorability / positivity / consistency, which we inherit from the standard g-formula; it is a function-class approximation assumption about the VAE.
>
> **Why this is not “evasive”.**
>
> Theorem 5.1 makes the dependence on this assumption explicit: under the latent factorization and context sufficiency condition, the latent g-computation rollout targets the same interventional law as data-space g-computation. An analogous approximation assumption is implicitly made by any VAE-based data-space rollout, which also relies on its decoder family to approximate the relevant one-step conditional distributions. One of our contributions is to show that, given whatever local approximation error is present, the error-propagation bound (Prop. 5.3) implies that latent rollouts are less prone to compounding these errors than autoregressive data-space rollouts.
>
> **Empirical support.**
>
> While such an approximation assumption cannot be fully verified from finite data alone, we provide diagnostics that are consistent with it in our setting: we compare our latent g-computation rollout to data-space variants and other baselines and observe strong performance in distributional metrics, especially at longer horizons. Furthermore, we checked KL capacity and made sure there was no posterior collapse.
>
> **Presentation**
>
> Thank you for pointing this out. We agree that the paper is highly technical and that, given space restrictions, intuitive comments are not as extensive as would be desirable. In the present revision, we have started addressing this by clarifying some definitions and cleaning up the notation. In a further revision, we are considering restructuring some aspects of the paper to make it easier to follow.
>
> **Definition of “scope”.**
>     -  In our original submission, “scope” was defined in Sec. 5.2 (“scope and selective decoding”) and Alg. 1. In the updated version, we define it in line 261. scope ∈ {all,last} controls **which horizons are decoded** given a fixed latent rollout. With scope = all, we decode / return the full path $y_{t+1:t+τ}$; with scope = last, we decode only the final horizon $y_{t+τ}$. The latent trajectory and state updates are identical in both cases; only decoding is selective. It is also possible to decode any other subset of steps selectively, although in the paper we focus on these two options.

---

> > ### Author Response · Authors · 2025-11-18
> >
> > **Jargon and notation consistency.**
> >     - We have standardized the nomenclature to use **“treatment”** for At or **“treatment plan”** for $ā_{t:t+τ−1}$. We avoid using “decision” or “plan” interchangeably with A.
> >
> > **Why ICU / MIMIC-based datasets?**
> >
> > We chose the semi-synthetic and real-world ICU datasets for three reasons.
> >
> > (1) Standard benchmark and comparability.
> >
> > Following the Causal Transformer (CT) line of work, there are three widely used benchmarks for time-varying treatment effect estimation: (i) a synthetic tumor-growth dataset, (ii) a semi-synthetic MIMIC-III dataset, and (iii) a real-world MIMIC-III dataset. Since CT [1] introduced the semi-synthetic MIMIC-III, methods for potential outcomes estimation under time varying treatments have routinely been evaluated on these benchmarks (e.g., [2,3]). In our experiments, we adopt the semi-synthetic and real MIMIC-based benchmarks, which allows a controlled comparison to strong baselines under matched preprocessing and confounding structure. We chose not to use the tumor-growth dataset because it has a one-dimensional outcome and essentially no covariates, whereas our model is designed for scenarios with time-varying covariates that may act as confounders; in such a setting G-Latent’s main advantages (rich covariate modeling and latent rollout) are not meaningfully exercised.
> >
> > (2) High-dimensional, long-range trajectories.
> >
> > ICU time series feature many covariates, treatments, and relatively long horizons, forming exactly the kind of setting where data-space g-computation becomes computationally heavy and error-compounding.
> >
> > (3) Availability of ground-truth potential outcomes.
> >
> > The semi-synthetic setup built from MIMIC-EXTRACT allows us to control confounding and know ground-truth potential outcomes, which is essential to evaluate distributional metrics under interventions. Few datasets provide ground-truth counterfactuals in the high-dimensional, time-varying treatment setting we consider.
> >
> > **Generalization to other domains and data types.**
> >
> > The proposed framework is domain-agnostic at the level of its causal interface: it only requires sequences of covariates \(X\), treatments \(A\), outcomes \(Y\), and static covariates \(V\). Nothing in Algorithm 1 or the theory assumes ICU-specific structure.
> >
> > - For other tabular or time-series domains (e.g., marketing, education), one can use the same multi-input transformer or a simpler recurrent encoder for \((\bar X_t, \bar Y_t, \bar A_{t-1}, V)\).
> > - For image or text covariates, the history encoder \(f_\omega\) can be augmented with a CNN or transformer that maps high-dimensional inputs at each time step into vector embeddings; these embeddings are then processed identically by the temporal core and VAE. Latent rollout and the g-computation equivalence do not depend on the modality of \(X\).
> >
> > While incorporating image-valued covariates is therefore straightforward at the architectural level, the current time-varying treatment literature (e.g., CT, G-Net and follow-ups) predominantly evaluates on tabular / time-series benchmarks such as the above MIMIC-based datasets rather than on raw images. Our experimental design follows this standard evaluation protocol. Exploring applications of G-Latent in settings with image covariates is an interesting direction for future work but is beyond the scope of this paper.
> >
> > **Code availability.**
> >
> > In the paper, we state that the code will be released upon acceptance. However, we are currently preparing an anonymized version of the codebase and expect to make it available during the rebuttal period.
> >
> > We thank the reviewer again for the thoughtful feedback, which we believe will help us substantially improve the clarity and impact of the paper.
> >
> > [1] Melnychuk, V., Frauen, D., & Feuerriegel, S. (2022, June). Causal transformer for estimating counterfactual outcomes. In International conference on machine learning (pp. 15293-15329). PMLR.
> >
> > [2] El Bouchattaoui, M., Tami, M., Lepetit, B., & Cournède, P. H. (2024). Causal contrastive learning for counterfactual regression over time. Advances in Neural Information Processing Systems, 37, 1333-1369.
> >
> > [3] Wang, H., Li, H., Zou, H., Chi, H., Lan, L., Huang, W., & Yang, W. (2025). Effective and Efficient Time-Varying Counterfactual Prediction with State-Space Models. In The Thirteenth International Conference on Learning Representations.

---

> > > ### Comment · Reviewer_35oK · 2025-11-19
> > >
> > > I appreciate the author for their response, and I am interested in further discussions.
> > >
> > > For the generative model assumption, I still suggest giving it a formal label (e.g., Assumption X.X). Its importance is arguably parallel to the three causal identification assumptions.
> > >
> > > Can the author provide an intuitive explanation (e.g., a toy example) of why "latent rollouts are less prone to compounding these errors than autoregressive data-space rollouts"?
> > >
> > > I also wonder what the author's thoughts are on IPTW's capacity to generalize to estimate conditional densities. I understand that g-computation is used here because it serves well for conditional estimates, whereas IPTW is for marginal estimates. However, the reverse tradeoff is that IPTW is much more computationally efficient to execute than the g-formula. Is there any possibility that IPTW can be extended to estimate the conditional estimate in the paper without the need for autoregression/rollouts?
> > >
> > > Adding to the previous point, I noticed that the authors did not include IPTW/MSM-based models for comparison. I believe it is helpful to include some SOTAs, such as Wu et al. (2024), in the paper for experimental comparison. This is to show how marginal estimates can be significantly off on conditional objectives, which could further strengthen the paper.
> > >
> > > Small issue: line 084, the use of "---" with "," looks weird.

---

> > > > ### Author Response · Authors · 2025-11-22
> > > >
> > > > We appreciate the follow-up by the reviewer.
> > > >
> > > > Following the suggestions, we have updated again the paper. We give a formal label to the Latent factorization and context sufficiency assumption, and we also add the following intuitive explanation at the end of the Theoretical Insights section, after proposition 4.4:
> > > >
> > > > Our model inevitably makes small one-step errors in the conditional distributions. The key difference is how these local errors are propagated. In the latent g-computation rollout, once the factual history is encoded, all future evolution happens in latent space and decoded predictions are never fed back; mathematically, the subsequent latent transitions are Markov and non-expansive in total
> > > > variation, so each local error contributes at most additively to the final discrepancy. In a data-space autoregressive rollout, every decoded prediction is fed back through a powerful encoder to form the next context, and these encode–decode maps can enlarge discrepancies, so a small local error at a given time step can be amplified at later steps. Proposition 5.4 formalizes exactly this: both approaches share the same local approximation errors, but only the data-space rollout has this additional error-amplification channel, which explains its worse long-horizon behavior.
> > > >
> > > > We also solved the small issue pointed out.
> > > >
> > > > We agree that IPTW is often computationally attractive, and it can be adapted beyond purely marginal estimands. However, in the time-varying, high-dimensional setting we study it comes with trade-offs that differ from ours.
> > > >
> > > > First, an IPTW-based approach requires a separate propensity model for the treatment process. Misspecification of this model directly propagates into the counterfactual estimator, in addition to whatever approximation error the outcome/generative model has. In contrast, G-Latent trains a single generative model for the outcomes and covariates  and does not rely on an explicit treatment model.
> > > >
> > > > Second, variance of the weights can be substantial in longitudinal settings. As a simple illustration, consider a 10-step horizon with binary treatment and a per-step propensity of 1/2 for a given trajectory; the joint probability of a particular treatment plan is then 2^{-10}, so the corresponding inverse probability weight is 2^{10}. In realistic, imbalanced settings, propensities can be much smaller and the weights much larger. Stabilized and truncated IPTW can mitigate this, but they do not completely remove the fundamental variance issue, especially when conditioning on rich histories.
> > > >
> > > > Third, extending IPTW to continuous treatments typically requires modeling a full conditional density rather than a simple multinomial propensity. This often entails using flexible density estimators such as conditional normalizing flows or related models, which adds a non-trivial amount of modeling and training complexity. In contrast, while our experiments use categorical treatments, the G-Latent architecture itself is agnostic to whether treatments are discrete or continuous: treatments enter as inputs to the history encoder and latent transition, and no explicit treatment density model is required.
> > > >
> > > > Regarding computational efficiency, we see the trade-offs as somewhat different from the classical “IPTW vs. g-formula” comparison. The standard criticism of g-computation is that it must model and sample all covariates, which is expensive. Our estimator indeed models covariates during training, but at inference time it can perform lightweight rollouts that only decode the outcomes, without ever regenerating covariates, and can even decode only a subset of the steps. This makes the per-query cost comparable to that of any conditional generative model for potential outcomes. In contrast, an IPTW-based approach would typically train two models (a propensity model plus an outcome/generative model), where the latter may be architecturally simpler (it only models outcomes, not covariates) but the former can be quite heavy, especially in the continuous-treatment case.

---

> > > > > ### Author Response · Authors · 2025-11-22
> > > > >
> > > > > Wu et al. (2024) propose a marginal structural generative framework where the conditional generator takes the entire treatment history as input and outputs an outcome at the target time. Their target is the population-level counterfactual distribution for each treatment plan, rather than individualized distributions conditional on rich time-varying histories. Moreover, the generator is a one-shot mapping from the whole treatment plan to the outcome, with no explicit latent state that evolves over clinical time. This design does not, by construction, encode the causal constraint that treatments at a given time step should only affect outcomes at later time steps, so there is no built-in mechanism preventing later treatments from influencing predictions for earlier outcomes; enforcing the correct temporal ordering would require introducing an explicit temporal factorization or state evolution. In contrast, G-Latent models the joint trajectory with a latent Markov rollout and directly targets individualized trajectory distributions, where the forward-in-time causal structure is built into the latent dynamics.
> > > > >
> > > > > Adapting Wu et al. (2024) to our setting would therefore require, at minimum, several substantial changes: (i) changing the estimand from population-level outcome distributions to individualized trajectory distributions, so that the model conditions on rich time-varying histories rather than only on treatment plans; (ii) modifying the generator architecture to introduce an explicit temporal factorization or latent state that evolves over clinical time, so that treatments at a given time step can only influence outcomes at later time steos; and, in the continuous-treatment case, (iii) extending the propensity model from a classifier to a flexible conditional density estimator (e.g., normalizing-flow–based). We fully agree that constructing and comparing against such an IPTW/MSM-style generative baseline, tailored to distributional, individualized potential outcomes, would strengthen the empirical picture and is an important direction for follow-up work. At the same time, we believe it is reasonable, in this paper, to focus on (a) introducing G-Latent as a practical estimator for individualized distributional potential outcomes under time-varying treatments, and (b) providing a theoretical and empirical comparison between latent and data-space g-computation rollouts for such outcome models.
> > > > >
> > > > > We have added this paragraph in the conclusions:
> > > > > ”We restrict attention to g-computation–based estimators rather than IPTW/MSM-style generative baselines (e.g., Wu et al. 2024). In principle, IPTW could be adapted to our conditional, trajectory-level estimands, but would require high-dimensional propensity models (or conditional treatment densities for continuous treatments) and weighted conditional density estimation, which can lead to unstable importance weights in long-horizon, high-dimensional settings. Designing and evaluating IPTW/MSM-style generative models for individualized distributional potential outcomes remains an interesting direction for future work.”
> > > > >
> > > > > We have also eliminated the Background section to gain space, and integrated the g-formula presentation in the Problem Section.
> > > > >
> > > > > We thank the reviewer again for these suggestions, which helped us clarify both our assumptions and the positioning of G-Latent. We would be happy to further discuss any of these points during the discussion period.

---

### Official Review · Reviewer_pxge · 2025-10-17

**Soundness:** 3
**Presentation:** 2
**Contribution:** 2
**Rating:** 4
**Confidence:** 3

**Summary:**

The paper proposes a method for estimating distributions of potential outcomes over time based on variational autoencoders. For this purpose, the authors extend recently proposed methods based on G-computation (G-Net) and move the G-computation steps into the latent space. The method is validated on synthetic and real-world datasets.

**Strengths:**

- Uncertainty quantification for treatment effects is a relevant and important topic, particularly in the time series setting
- The proposed architecture is compelling and yields several practical advantages
- Experimental results look promising

**Weaknesses:**

- Novelty: The key ingredients (e.g., G-computation, VAEs) are known. The contribution is mainly a novel backbone/ model architecture tailored to distributional G-computation
- In the Appendix, the authors describe that they used the same shared and fixed hyperparameters for all transformer architectures (reasonable). However, for G-Latent, additional hyperparameters are tuned in a data-driven manner. I think this might give an unfair advantage to the G-Latent method as compared to the baselines
- The authors evaluate using a point error metric (RMSE) and two distributional metrics (energy and KL). However, the distributional metrics appear to be based on assumptions about the data (e.g., KL is compared to a Gaussian kernel density estimator). I think it may be valuable to use an additional assumption-agostic distributional distance (e.g., Wasserstein distance to the empirical data distribution).
- Missing baselines: There are existing baselines not compared with, that can learn the aleatoric uncertainty of potential outcomes over time: E.g., "Counterfactual Generative Models for Time-Varying Treatments" (Uses IPW instead of G-computation). Additionally, the authors claim that "Bayesian Neural Controlled Differential Equations for Treatment Effect Estimation" only models epistemic aleatoric uncertainty, which seems incorrect, as the paper models the posterior predictive distribution

**Questions:**

- Why are the results reported only for selected time steps t? How was this selection made?
- How does the method perform when the strength of time-varying confounding is varied?

---

> ### Author Response · Authors · 2025-11-18
>
> We thank the reviewer for carefully reading and reviewing our paper, and for recognizing several aspects of our work.
>
> We agree that our building blocks (g-computation, VAEs, transformers/GRUs) are standard, and we do not claim novelty at the level of these components. We apologize if our writing suggested that the contribution is mainly architectural. As also detailed in our **general rebuttal (“On novelty”)**, our contribution is at the estimator level rather than at the level of individual architectural components.
>
> Concretely:
>
> 1) **Latent-space g-computation estimator.** We introduce a discrete-time g-computation estimator that performs rollouts entirely in the **latent space** of a conditional VAE with GRU updates. Under a latent factorization / context-sufficiency assumption, we prove that this estimator targets the **same interventional distribution** as the standard g-formula, while never autoregressing covariates in observation space (Thm. 5.1 / Cor. 5.2).
>
> 2) **Error-propagation guarantee.** We show that, given any local one-step approximation error, the latent rollout enjoys a **tighter total-variation error-propagation bound** than data-space autoregressive g-computation (Prop. 5.3), which theoretically explains the improved stability we observe at longer horizons.
>
> 3) **Distributional POs without residual pools in discrete time.** To our knowledge, this is the first **discrete-time** method that estimates **individualized distributions** of potential outcomes without global residual pools, and that comes with both identifiability (same target as the g-formula) and error-propagation guarantees. We also add architectural choices (ALD mixture decoder, decoupled transformer encoder with lightweight latent GRU and selective decoding) that are intended as **important but incremental refinements** that make this estimator effective and efficient in practice, rather than as standalone sources of novelty.
>
> **Hyperparameters**:  the transformer hyperparameters that we use are the ones [1] selected after an optimization process in a data specific way. They are used for all the models that have a transformer, including ours (they are the same across baselines, not across datasets). For our model, we do an additional optimization process for some specific hyperparameters, and for G-Net we take the hyperparameters that also [1] selected in a data specific way. Thus, we believe that our comparison is fair.
>
> We apologize that the current text was unclear. We use:
>
> - The **same multi-input transformer hyperparameters for all transformer-based models** (CT, CT-CRPS, CT-Gaussian, Transformer G-Net, and G-Latent). These are taken from [1] and were selected via dataset-specific tuning there.
> - For **G-Net**, we use the hyperparameters from the public implementation of Melnychuk et al., again tuned in a data-specific way by the original authors.
> - For **G-Latent**, the **only** additional tuning we perform concerns VAE-specific hyperparameters that have no analogue in the baselines (KL weight, latent dimension, reconstruction weights, etc.). These are selected via a lightweight search on factual validation sets.
>
> So all transformer-based models share the same transformer backbone, and baseline configurations are taken from (or directly aligned with) hyperparameters that were already tuned in prior work for each dataset. We will clarify this more explicitly in the appendix to avoid the impression that G-Latent received disproportionate tuning.
>
> **Evaluation**: We do not use KL divergence as an evaluation metric. The “LL-KDE” metric in the tables is the log-likelihood of the observed outcome under a **Gaussian kernel density estimator** fitted to our MC samples. This can be applied to any set of MC samples, regardless of their true distribution, though it will indeed behave best when the data are near-Gaussian.
> Importantly, we already use a **fully distribution-agnostic proper scoring rule**: the **Energy Score (ES)**, which is a standard way to evaluate multivariate predictive distributions from MC samples. ES does not assume any particular parametric form and precisely measures how well a single observation is embedded in the predicted distribution.
> We chose not to use Wasserstein distance because it is more natural for comparing **two empirical distributions**, while in our evaluation we compare **a single test observation** to a predictive distribution represented by MC samples. Treating the observation as a Dirac measure would essentially reduce this to a different loss that is less well-aligned with common probabilistic forecasting practice. For our setting (predictive distribution vs. single realized outcome), ES is the standard and more appropriate choice.

---

> ### Author Response · Authors · 2025-11-18
>
> **Baselines**: We do not compare against Counterfactual Generative Models for Time-Varying Treatments [2] because it estimates **population-level** distributions of potential outcomes under time-varying treatments (through IPTW), whereas our focus is on **individualized** potential outcome distributions conditional on rich patient history. Extending their approach to individualized distributions in our setting would require non-trivial modifications. For this reason we treated it as complementary rather than a direct baseline; we will make this distinction clearer.
>
> Regarding [3], the reviewer says that: the authors claim that "Bayesian Neural Controlled Differential Equations for Treatment Effect Estimation" only models epistemic aleatoric uncertainty. In the related work, we mention that it models both epistemic and aleatoric uncertainty. On the other hand, as we mention in the Baselines (within Section 6), we exclude the former because it introduces a heavy machinery for epistemic uncertainty and continuous time processing (our setting is discrete and our model addresses aleatoric uncertainty) that makes it very expensive to train, while its way to handle aleatoric uncertainty is just a gaussian head, which is already covered by CT-gaussian. The authors of [3] report a training time of 34 hours for dataset much simpler than ours. Running this method fairly with multiple random seeds and hyperparameter configurations is a disproportionate compute effort taking into account that the big complexity of this model is due to adaptations to continuous time and estimation of epistemic uncertainty and not to aleatoric uncertainty quantification.
>
> **Q1**. We show the results for a representative subset of the steps in the main text due to space limitations. The complete results are in appendix J.
>
> **Q2**. We compare our model against the baselines in (i) the original semi-synthetic dataset that has practical violations of the positivity assumption, and also limited confounding, (ii) our new version of this semi-synthetic dataset, with a strong level of confounding verified via treatment propensities, and (iii) a real world dataset, where we expect substantial time-varying confounding, although we can only evaluate on factual treatments . Across these regimes, G-Latent tends to outperform the baselines on distributional metrics, and our
> theoretical analysis shows that the latent g-computation targets the same estimand as the standard g-formula under the usual identification assumptions (sequential ignorability, positivity, consistency). A systematic sweep over confounding strength is an excellent suggestion; due to space and compute constraints we did not include such an ablation here, but we plan to explore it in follow-up work.
>
> We thank the reviewer again for the thoughtful feedback, which we believe will help us substantially improve the clarity and impact of the paper.
>
> [1] Melnychuk, V., Frauen, D., & Feuerriegel, S. (2022, June). Causal transformer for estimating counterfactual outcomes. In International conference on machine learning (pp. 15293-15329). PMLR.
>
> [2] Wu, Shenghao, et al. "Counterfactual generative models for time-varying treatments." Proceedings of the 30th ACM SIGKDD Conference on Knowledge Discovery and Data Mining. 2024.
>
> [3] Hess, K., Melnychuk, V., Frauen, D., & Feuerriegel, S. (2023). Bayesian neural controlled differential equations for treatment effect estimation. arXiv preprint arXiv:2310.17463.

---

> > ### Comment · Reviewer_pxge · 2025-11-28
> >
> > I thank the authors for their rebuttal and clarifications. The rewriting of the paper definitely highlights the contribution better by placing less emphasis on the proposed architecture.
> >
> > However, I do not remain entirely convinced of the benefits of latent G-computation. Intuitively, moving G-computation into latent space should only have advantages if the latent dimension/ distributional complexity is somehow "simpler" then modeling the conditional distributions of the observed covariates. This could be reflected in modeling choices. However, it is not really possible to test this, and reducing dimensions too much will likely introduce bias. I feel like this point is not yet sufficiently taken into account.
> >
> > Additionally, I still feel that the contribution (i.e., moving G-computation into latent space and the error composition) is somewhat marginal. Regarding the experiments. The authors mention that they tune their hyperparameters, but reuse hyperparameters for baselines from existing papers. I think that this is only a fair comparison if the exact same experiments are performed (exactly the same DGPs, evaluation, sample splits, etc). Is this really ensured here?

---

> > > ### Author Response · Authors · 2025-11-28
> > >
> > > We thank the reviewer for their thoughtful follow-up and for engaging with the revised version of the paper. Below, we clarify the benefits of latent g-computation beyond latent dimensionality, discuss the choice of latent dimension, explain our hyperparameter setup, and further articulate the nature of our contribution.
> > > First, we would like to clarify that latent g-computation avoids decoding at intermediate steps **for any latent dimensionality**. As we show in Proposition 4.4, this reduces error accumulation independently of the dimension of the latent space, and it also saves substantial compute whenever we do not need to estimate outcomes at all intermediate steps. Note that a data-space g-computation model typically also has internal latent variables in its generative process, so this is not something that our model adds; what we do is to exploit these latents as the state on which g-computation operates, and we prove that this is sufficient for estimating distributional potential outcomes. Intuitively, the latent rollout reduces accumulation error because it never feeds previously decoded covariates back into the model: it removes decoding–re-encoding as a source of compounding error, regardless of the size of the latent space. The computational gain for only last step decoding vs. full trajectory decoding models is indeed relatively larger when the latent dimension is smaller, since latent-only rollouts become cheaper while the cost of decoding all steps is roughly constant.
> > >
> > > We also agree that choosing the latent dimension too small can introduce bias. In practice, we treat the latent dimension as a regular hyperparameter and select it using factual validation performance, ensuring that we are in a regime where increasing the dimension further does not yield improvements in predictive or distributional metrics. This typically gives us a latent space that is small enough to keep the model efficient, but not so small that factual performance degrades. Importantly, the benefits of latent g-computation—both in terms of reduced error accumulation and compute—are not restricted to very low-dimensional latents. We will clarify this in a revised version.
> > >
> > > Regarding the hyperparameters, we completely understand the concern of the reviewer, and agree that it is fundamental that the exact same experiments were performed. We indeed performed exactly the same experiments as Causal Transformer: we reused their code for generating the datasets and their configuration values, so dataset parameters, sample splits, etc. were all the same. We will further clarify this in our paper.
> > >
> > > Finally, regarding novelty, we would like to highlight that, beyond bringing g-computation to the latent space, we (i) formally define the corresponding latent g-computation estimator, and theoretically demonstrate its validity —which is not trivial a priori— by proving that it targets the same interventional distribution as the standard g-formula, (ii) establish a theoretical error-propagation advantage relative to data-space g-computation, which is also confirmed empirically, and (iii) develop a set of incremental but practically important improvements that allow to get proper, well calibrated and computationally efficient distributional estimates, and that altogether make our approach overcome by an important margin previous g-computation based models.
> > >
> > > We hope these clarifications address the reviewer’s remaining concerns about the benefits and novelty of latent g-computation and the fairness of our comparisons.

---

### Official Review · Reviewer_oFrA · 2025-10-20

**Soundness:** 2
**Presentation:** 2
**Contribution:** 1
**Rating:** 2
**Confidence:** 4

**Summary:**

This paper introduces G-Latent, a neural method for estimating distributions of potential outcomes in discrete time.
The authors leverage a transformer-VAE based architecture with a GRU decoder component to approximate the distributional version of G-computation.
Further, the advantages of latent representations over standard autoregressive rollouts are discussed.
Finally, the method is benchmarked on a semi-synthetic dataset and a real world dataset.

**Strengths:**

- Complete piece of work: The paper provides (almost) all necessary notation, assumptions and proofs, and is therefore self-contained.

- Sensible approach: The approach for G-computation chosen by the authors seems to be a sensible choice. Performing distributional G-computation with a VAE approach rather than with residual hold-outs as in G-Net is certainly a step forward regarding implementation.

- Important topic: Estimating potential outcomes over time has recently received more attention, and incorporating uncertainty quantification is an important part of it.

**Weaknesses:**

**Major:**

- **Very limited novelty:** Overall, the novelty of this work is very limited, which is my main concern: From a theoretical perspective, distributional G-computation with neural networks **has been established by G-Net** [1].
From an implementation perspective, while using transformers and VAEs is an improvement over LSTMs with residual hold-outs, the architecture presented is just a **simple combination of the multi-input transformer by [2] with the asymmetric VAE in [3]**.

- **Training objective:** The training objective looks a bit off to me. Specifically, I cannot make sense of the auxilliary one-step prediction loss. There is **no technical reason** why the history representation has to be (necessarily) predictive of a one-step ahead factual outcome.
This component is not grounded theoretically; it is further striking to me how a balancing parameter $\lambda$ should be tuned in practice.

- **Limited evaluation:** there is only a **single dataset** (semi-synthetic MIMIC III) on which the authors actually benchmark their method for potential outcome estimation, which is not think this is sufficient. Further, I do not like the reasons for which [4] and [5] were not chosen as baselines.

- **Point estimation:** Proposition 5.3 argues that computing latent representations is superior to autoregressive rollouts.
This is an interesting insight, but the theorem only covers distributional estimation, and oversees that baselines that are designed for computing point estimates of the potential outcomes should be analyzed with different metrics other than the total variation.
Specifically, while computing distributions of latent representations may be superior to computing distributions of autoregressive rollouts, it may not be superior to computing point estimates of autoregressive rollouts when the goal is to estimate a point estimate of the potential outcome. Computing distributions is statistically more difficult than computing point estimates, and if the goal is to compute a point estimate of the potential outcome, there is no reason to use G-Latent (which is even validated by the experiments).
There is nothing wrong with the Proposition itself: instead, what is want to point out is that the lemma does not apply to atuoregressive point estimation methods such as CT and, therefore, the **insight is rather limited.**

**Minor:**

- Many of the cited works are already published by conferences and not preprints on arxiv any longer. I would advise the authors to double-check their references.

- The total variation distance should be introduced somewhere in the paper.

- I think the paper could be presented a little better. There is a lot of text that could, perhaps, be shortened and summarized in an additional figure. But this is, of course, a matter of personal taste.

- In Proposition 5.3, the assumption made in the implied Lemma from the appendix should be clearly stated.



____
[1] Rui Li, Stephanie Hu, Mingyu Lu, Yuria Utsumi, Prithwish Chakraborty, Daby M Sow, Piyush Madan, Jun Li, Mohamed Ghalwash, Zach Shahn, et al. G-net: a recurrent network approach to g-computation for counterfactual prediction under a dynamic treatment regime. In Machine Learning for Health, pp. 282–299. PMLR, 2021.

[2] Valentyn Melnychuk, Dennis Frauen, and Stefan Feuerriegel. Causal transformer for estimating counterfactual outcomes. In International conference on machine learning, pp. 15293–15329. PMLR, 2022.

[3] Seunghwan An and Jong-June Jeon. Distributional learning of variational autoencoder: Application to synthetic data generation. Advances in Neural Information Processing Systems, 36:57825–57851, 2023.

[4] Konstantin Hess, Valentyn Melnychuk, Dennis Frauen, and Stefan Feuerriegel. Bayesian neural controlled differential equations for treatment effect estimation. International Conference on Learning Representations, 2024.

[5] Wenhao Mu, Zhi Cao, Mehmed Uludag, and Alexander Rodr´ıguez. Counterfactual probabilistic diffusion with expert models. arXiv preprint arXiv:2508.13355, 2025.

**Questions:**

- Can the authors clearly point their technical novelty?

- What is the reason for the auxilliary one-step prediction loss?

- What is the authors' take on point estimation of potential outcomes with their method?

---

> ### Author Response · Authors · 2025-11-18
>
> We thank the reviewer for their time in carefully reading and reviewing our work.
>
> Next, we address the weaknesses pointed out by the reviewer, and answer their questions.
>
> **Novelty:**
>
> We agree that replacing the RNN in G-Net with a transformer is an incremental change (we do not list it as a contribution in our paper), and that using a VAE instead of residual noise alone would still be a recombination of existing components. Our main novelty is different: we propose a **latent g-computation estimator** that performs g-computation in the latent space of a conditional VAE whose latent state is updated by an RNN/GRU. A priori it is not guaranteed that such a construction preserves the standard g-formula target or behaves better than data-space rollouts; in this work we (i) show theoretically that our latent rollout corresponds to a valid g-computation estimator for the *same* interventional distribution as the standard g-formula (Thm. 5.1), (ii) prove that it exhibits more favorable error propagation than data-space autoregressive rollouts (Prop. 5.3), (iii) demonstrate empirically that it outperforms data-space rollouts and strong baselines in distributional metrics, and (iv) analyze, both theoretically and empirically, its advantages in terms of inference-time compute.
>
> In addition, we introduce incremental but practically important design choices that make a VAE-based g-computation model work well in practice, such as using an ALD mixture head for flexible outcome distributions and decoupling sequence encoding from latent rollout for fast inference with transformer-quality representations. Finally, to the best of our knowledge, ours is the first **discrete-time** model for individualized *distributional* potential outcomes **without** relying on global residual pools: prior work that estimates full PO distributions without residual pools either focuses on continuous time or targets population-level distributions rather than individualized trajectories. We refer the reviewer to the general rebuttal comment for a more extended discussion on the novelty and to the new version of the manuscript, where we have restructured several parts of our work to make the novelty clearer. We apologize that novelty was not sufficiently well communicated in the original submission.
>
> **Training objective:**
>
> The transformer encoder we use is a high-capacity architecture with many parameters, and in contrast to point predictors, the VAE loss alone can be minimized substantially even if the history representation carries little predictive signal (the decoder can partly *ignore* the transformer representation). When we train the transformer and the VAE jointly from scratch, there is therefore a risk that the encoder does not learn useful representations of the history and that training converges to a degenerate configuration: a reasonable VAE objective value but representations that are unsuitable for sampling conditioned on individual history.
>
> The purpose of the auxiliary one-step prediction loss is to **regularize and guide** the history representation by enforcing that it is predictive of the next factual outcome as a proxy task. As mentioned in the implementation details in Sec. 5.1, we apply a warm start where only this auxiliary loss is trained. This helps the transformer learn meaningful initial features of the history, which then serve as a good starting point for conditioning the VAE. As these features are already informative, the VAE is encouraged to make use of them from the beginning instead of ignoring them. We completely agree that this loss is not grounded in the causal theory and that other proxy objectives could be used; we will clarify in the paper that it is a *technical* device to stabilize training and improve representations, and that it does not alter the well-grounded VAE objective or the definition of the estimator.
>
> The hyperparameter that weights this auxiliary loss is tuned like other hyperparameters: by monitoring empirical performance on a held-out factual validation set.

---

> > ### Author Response · Authors · 2025-11-18
> >
> > **Limited evaluation:** We evaluate on two datasets: (i) the real-world MIMIC-III dataset and (ii) the semi-synthetic dataset built from MIMIC-III as proposed by [2]. While both are based on MIMIC-III, the semi-synthetic dataset differs substantially from the real-world one in terms of outcome-generating mechanism and treatment policy (we detail this in Appendix F). In line with prior work on potential outcomes under time-varying treatments, the three commonly used benchmarks are: real-world MIMIC-III, semi-synthetic MIMIC-III, and the tumor-growth synthetic dataset [2]. We chose not to use the tumor-growth dataset because it has a one-dimensional outcome and essentially no covariates, whereas our model is designed for scenarios with time-varying covariates that may act as confounders; in such a setting G-Latent’s main advantages (rich covariate modeling and latent rollout) are not meaningfully exercised.
> >
> > As for the comment on the baselines, in the case of [5], the preprint appeared very close to the submission deadline and no code was provided. Given the complexity of the method, together with the timing and lack of reference implementation, it would in practice be very difficult for any group to produce a faithful and well-tuned implementation within the same submission cycle. We therefore regard [5] as concurrent and complementary work rather than a baseline that can be fairly included in this evaluation. Furthermore, the setting it addresses (continuous time and domain-specific expert ODE models) is different from ours: in our work, we do not assume access to any mechanistic model; we only observe high-dimensional covariates, treatments, and outcomes in discrete time.
> >
> > With regards to [4], according to the authors, it took 34 hours to train their model on a dataset much simpler than ours. Running this method fairly with multiple random seeds and hyperparameter configurations is a disproportionate compute effort taking into account that the big complexity of this model is due to the fact that (i) it is designed to handle continuous time  (our setting is discrete, so it should not have any advantage for that), and (ii) it introduces a heavy Bayesian machinery for epistemic uncertainty modeling (we address aleatoric uncertainty, so again it should not have any advantage for that). On the other hand, as we mention in the paper, its way to handle aleatoric uncertainty is a Gaussian head, which we already cover with our Gaussian baselines.
> >
> > **Point estimation**: We fully agree that computing full predictive distributions is statistically more demanding than computing point estimates, and that methods specifically designed for point estimation (such as CT) should be evaluated with point metrics (e.g., RMSE). Our primary goal in this work is to estimate **individualized distributions** of potential outcomes over time and to quantify aleatoric uncertainty; accordingly, Proposition 5.3 is stated in terms of total variation distance between *distributions* and does not claim superiority for point estimation.
> >
> > In the experiments, we therefore treat CT and related methods as **strong point-estimation baselines** and evaluate G-Latent both on point metrics (RMSE) and distributional metrics (Energy Score, log-likelihood). While G-Latent is competitive in RMSE, point-focused methods like CT perform better—which we see as consistent with the fact that they optimize a simpler target. Our claim is not that G-Latent should replace point estimators when the goal is purely point prediction, but that when **distributional potential outcomes and uncertainty quantification are required**, latent g-computation offers (i) improved theoretical error propagation for the distribution (Prop. 5.3) and (ii) empirically strong performance on distributional metrics.
> >
> > Proposition 5.3 is intended as a statement about *distributional* estimation under g-computation and does not apply directly to other autoregressive **point** estimators that do not follow g-computation, such as CT. We changed the title of the paragraph where proposition 5.3 is stated from “Error propagation: latent vs. autoregressive rollouts” to “Error propagation: latent vs. data-space g-computation rollouts” to avoid confusion.
> >
> > Minor Aspects:
> >
> > We thank the reviewer for pointing out several minor aspects of our paper to improve. We have changed the references for the published works, added a total variation distance definition, and clearly stated the assumption in proposition 5.3. Also, we will think in additional ways to make the paper clearer and will consider adding new figures.
> >
> > We thank the reviewer again for the thoughtful feedback, which we believe will help us substantially improve the clarity and quality of the paper.

---

> > > ### Comment · Reviewer_oFrA · 2025-11-26
> > >
> > > **Limited novelty:**
> > >
> > > Thank you. Regarding theory: I do not know which theorem 5.1 the authors are referring to, I assume they mean 4.2. This theorem relies on assumption E.1, which is a strong assumption and is practically hard to guarantee (it makes the proof work). Further, Assumption E.7 for Proposition 4.4 (which the authors refer to as 5.3?) looks a bit arbitrary to me. Is this assumption informed by literature?
> > >
> > > **Training objective:**
> > >
> > > > The purpose of the auxiliary one-step prediction loss is to regularize and guide the history representation by enforcing that it is predictive of the next factual outcome as a proxy task.
> > >
> > > This has no theoretical justification; unfortunately, your response did not resolve my concern, but reinforced it. Why should enforcing prediction of the next factual outcome be a proxy for a counterfactual treatment sequence? As I see it, there is no theory supporting this claim.
> > >
> > > > The hyperparameter that weights this auxiliary loss is tuned like other hyperparameters: by monitoring empirical performance on a held-out factual validation set.
> > >
> > > This is exactly my problem; you introduce another hyperparemeter, that has **significant** impact on the overall performance of the model. The only way to heuristically tune this hyperparameter is to use factual outcomes; hence, you tune your model to be predictive of the factuals, and the hyperparameter favors factual outcomes for no theoretically valid reason.
> > >
> > > **Limited evaluation:**
> > >
> > > Thank you for clarifying. Since you target the counterfactuals, evaluating on real-world data sets is basically meaningless, as there is no need for adjustment strategies like G-computation. Any backdoor-adjustment / standard regression suffices. Hence, the only relevant dataset is the semi-synthetic one.
> > >
> > > However, I fully agree that regarding computing time, the proposed method appears much more sensible than SDE-based alternative or the hold-out sampling approach.
> > >
> > > **Point estimation:**
> > >
> > > Thank you for clarifying.
> > >
> > > **Minor** (just noticed this):
> > >
> > > In case of a resubmission / camera-ready version, please consider adding the years to all the references. Some references appear without year.
> > >
> > > ____
> > >
> > > Overall, I think this paper is a fair attempt to develop a neural network with some smaller improvements over existing works. However, I have concerns about i) the **training objective** (especially how a factual outcome balancing loss should be tuned in practice and why this makes sense at all), ii) **limited evaluation** (only one relevant dataset), iii) **strong assumptions** regarding the claimed main contribution of the paper, i.e., identification through latent G-computation.
> > >
> > > I believe the authors that their method has some advantages, such as shorter computing times compared to the baselines. However, I strongly believe that iv) the **contribution is very limited**.
> > >
> > > *It is not a bad paper, but I do **not** think it is good enough for a top conference like ICLR. I will maintain my score.*

---

> > > > ### Author Response · Authors · 2025-11-26
> > > >
> > > > We thank the reviewer again for the detailed follow-up and for engaging carefully with both the paper and our previous response. We address the new comments point by point.
> > > >
> > > > We apologize for the mismatch in theorem numbering. Our earlier response referred to the numbering in an intermediate revision. After the latest update (following the discussion with reviewer 35oK), the numbering changed: Theorem 5.1 is now Theorem 4.2, and Proposition 5.3 is now Proposition 4.4.
> > > >
> > > > With respect to the latent factorization and context sufficiency (in the current version, Assumption 4.1 or E.1 in the appendix), we note that this is the standard modeling assumption in conditional VAEs: the decoder family is rich enough to approximate the true conditional density given a sufficiently informative context. It is not an additional causal identifiability assumption beyond the usual sequential ignorability / positivity / consistency, which we inherit from the standard g-formula; it is a function-class approximation assumption about the VAE, and it is shared by any VAE-based model, including one that performs standard data-space g-computation, so it does not add more assumptions than a VAE-based data-space model. While such an approximation assumption cannot be fully verified from finite data alone, we provide diagnostics that are consistent with it in our setting: we compare our latent g-computation rollout to data-space variants and other baselines and observe strong performance in distributional metrics, especially at longer horizons. Furthermore, we checked KL capacity and made sure there was no posterior collapse.
> > > >
> > > > Regarding assumption E.7, it requires that the single-step autoregressive (AR) tail operator, which maps the law of the decoded quantity at time j to the law at time j+1 via re-encoding and decoding, is Lipschitz-continuous in total variation with factor $(1+\lambda_j)$. We introduce this assumption to isolate a clean, interpretable condition under which error amplification is controlled. This is not an ad-hoc condition: it is a standard type of stability assumption for Markov kernels and neural generative models. Any finite composition of affine layers and Lipschitz activation functions is itself Lipschitz on bounded subsets of the input space. In Appendix E (below Assumption E.7), we show that if (i) the “context update” map that re-encodes decoded outputs into histories is Lipschitz, and (ii) the decoder-induced one-step kernel family is Lipschitz in its context, then the induced AR tail operator on probability measures is Lipschitz in total variation with constant $(1+\lambda_j)$, where $\lambda_j$ is controlled by the product of these layer-wise Lipschitz constants. This connects Assumption E.7 directly to standard Lipschitz properties of neural networks, rather than to any special causal structure. We refer the reviewer to [3] for an analysis on Lipschitz regularity of deep neural networks. We will include this reference in the next revision of the paper.
> > > >
> > > > With respect to the auxiliary loss, we fail to see the reason for the criticism. We already clarified that it is a regularization / representation-learning mechanism, not part of the causal definition of the estimator: the auxiliary head does not change the estimand; it only affects how the encoder parameters are trained in practice.
> > > >
> > > > Importantly, when we refer to it as a “proxy task,” we do not mean a proxy for the *causal* objective (the target interventional distribution), but a proxy for learning a useful history representation. In our architecture, the transformer representation is used to condition the VAE in order to approximate $p(L_{t+1}∣\bar{H}_t,A_t)$. Encouraging it to predict the next factual outcome is therefore a practical way to avoid degenerate solutions where the decoder largely ignores the history representation.
> > > >
> > > > While, as we mentioned in our rebuttal, the auxiliary loss is not theoretically motivated, it does not alter the well-grounded VAE objective or the definition of the estimator. We see our warm-start with this loss as a pre-training for the transformer, so it already captures some meaningful features (it is quite clear to us that a history representation that is predictive of the next step will contain some meaningful features for the VAE optimization task, and that at least will be better than a random initialization). Above all, empirically, we found this design improves training stability and performance; since it does not affect the estimator or our theoretical results, we include it as an implementation detail.
> > > >
> > > > **Very succinctly: we include the auxiliary loss component because it empirically has proved helpful. This implementation detail is not grounded theoretically and we do not claim so, but it does not affect the theoretically grounded core components of our model.**

---

> > > > > ### Author Response · Authors · 2025-11-26
> > > > >
> > > > > As for the criticism on how the hyperparameter is found, the reviewer says:  “The only way to heuristically tune this hyperparameter is to use factual outcomes; hence, you tune your model to be predictive of the factuals, and the hyperparameter favors factual outcomes for no theoretically valid reason”. This is true, but it is exactly the same that happens with any other hyperparameter, in our work and in any other related work: we do not see the non factual potential outcomes during training, so we train and optimize hyperparameters based on factual data. For example, [1] or [2] find the hyperparameters that regulate the trade-off between the predictive and the balancing components of their respective models with factual data also.
> > > > >
> > > > > With respect to the evaluation comment, including real world datasets is a standard practice in literature (for example, in the same works cited above [1] and [2]) to test applicability in real world settings. Although it is true that it does not have the same relevance as datasets where we have ground truths, we believe that a good performance in the real world dataset of latent g-computation with respect to data space g-computation is a good indicator. We agree with the reviewer, however, that in future work it would be interesting to test our approach in additional synthetic datasets.
> > > > >
> > > > > As for the minor comment on the years in the references, thanks for noticing it, we will include the years in our next revision.
> > > > >
> > > > > Finally, we would like to note that, in the initial review, the main criticism was the lack of novelty, and that this was supported by a claim that our main contribution was adding a multi-input transformer with an asymmetric VAE, without mentioning the g-latent estimator. After the clarification of our estimator-level contribution and restructuring of the paper, the main novelty critique seems to have shifted towards the assumptions we make. For this reason, we hope the reviewer will consider the fact, mentioned above, that (i) assumption 4.1 is the standard modeling assumption of VAEs rather than an additional causal assumption, and that it is shared also with a VAE-based data space models, and that (ii) E.7 comes directly from standard Lipschitz properties of neural networks.
> > > > >
> > > > > [1] Melnychuk, V., Frauen, D., & Feuerriegel, S. (2022, June). Causal transformer for estimating counterfactual outcomes. In International conference on machine learning (pp. 15293-15329). PMLR.
> > > > >
> > > > > [2] El Bouchattaoui, M., Tami, M., Lepetit, B., & Cournède, P. H. (2024). Causal contrastive learning for counterfactual regression over time. Advances in Neural Information Processing Systems, 37, 1333-1369.
> > > > >
> > > > >  [3] Virmaux, A., & Scaman, K. (2018). Lipschitz regularity of deep neural networks: analysis and efficient estimation. *Advances in Neural Information Processing Systems*, *31*.

---

> > > > > > ### Comment · Reviewer_oFrA · 2025-11-27
> > > > > >
> > > > > > Thank you for your responses.
> > > > > >
> > > > > > **Regarding the assumptions:** I think that everything is clear now. I would appreciate if the assumptions were highlighted in the main paper along with the references above (to avoid confusion).
> > > > > >
> > > > > > **Regarding the balancing objective:** I understand using a loss that partly predicts factual outcomes, partly counterfactuals, can increase performance empirically. The issue is, still, how to tune such a loss when the counterfactuals are not available for validation. The authors say that they tuned their hyperparameters (including the balancing $\lambda$) on factual data -- which makes sense --, but their statement that this balancing objective improved their performance empirically implies that they have validated their method on unseen counterfactuals as well.
> > > > > >
> > > > > > My issue is: the parameter $\lambda$ will have a much larger impact on the overall performance of the method than, for example, the number of weights in some hidden layer. If it is only tuned on factual data, then the method will always be biased towards factual outcomes, and this bias is not quantifiable in any way. This is why methods that target an unbiased estimand would always be more reliable for counterfactual inference; at least, they are guaranteed to target the correct estimand as opposed to the proposed method.
> > > > > >
> > > > > > **Regarding experiments:** I agree that several papers propose such sanity checks on real-world data, and I do not see this as a minus for the proposed work. However, it is not a proper way to validate the method on counterfactuals, and hence, there only remains a single valid dataset in the paper where the method is **correctly** validated, i.e., on the task that it is supposed to solve.
> > > > > >
> > > > > > Overall, I appreciate the authors' clarifications on the assumptions. However, I am not convinced by the heuristic training objective, and I do not think that the experimental validation is sufficient.
> > > > > >
> > > > > > *I will increase my score to 4*, but I strongly feel the work will need a significant revision for another submission round.

---

> > > > > > > ### Author Response · Authors · 2025-11-28
> > > > > > >
> > > > > > > We are grateful for the reconsideration of your evaluation in light of our clarification of the novelty and the assumptions of our work.
> > > > > > >
> > > > > > > As for the comment on the training objective, you write: “I understand using a loss that partly predicts factual outcomes, partly counterfactuals, can increase performance empirically”. We would like to clarify some fundamental aspects regarding this sentence. Our training loss, as the loss of any other g-computation method, does not target counterfactuals. We minimize a VAE objective defined on factual outcomes only. What makes a g-computation estimator target an unbiased interventional distribution is the *inference procedure*, where we generate trajectories by composing a sequence of observational (factual) conditionals according to the g-formula. In our model, this process takes place in the latent space, but the logic is the same. Thus, the method targets the same unbiased counterfactual distribution even though it is trained to approximate factual outcomes.
> > > > > > > Hyperparameter selection is a different issue: all hyperparameters (including the auxiliary-loss weight) are necessarily tuned using factual validation data, because counterfactuals are never observed. In that sense, the model is “optimized” on factual outcomes (we prefer this term to “biased” to avoid confusion with estimator bias). This is true for our work and for essentially all related methods in this area: the performance gap between a model whose hyperparameters are tuned on factual data and a hypothetical model tuned on counterfactual data is intrinsically unquantifiable unless counterfactual ground truth is available.
> > > > > > >
> > > > > > > We would like to emphasize that the unbiasedness of our model depends on (i) the inference process and (ii) the VAE aiming at the correct factual distributions. (i) is unaffected by our auxiliary loss, while (ii) is improved by this implementation detail. In other words, it is a representation-learning / regularization device that influences how well the factual distributions are modeled in finite samples, but it does not change the estimand or the form of the estimator.
> > > > > > >
> > > > > > > When we wrote that the auxiliary loss improves performance empirically, we were referring to performance on factual data, not to validation on counterfactual outcomes.
> > > > > > >
> > > > > > > An important takeaway is that g-computation models in general aim at an interventional unbiased estimand by properly modeling and sampling from factual distributions (in our case, we do not sample in data-space because the process takes place in the latent space, but we still achieve counterfactuals at inference while modeling factual outcomes at training). Therefore, properly modelling factual distributions is a fundamental part of the process. What we do with the auxiliary loss is improve the way the factual distributions are modeled.
> > > > > > >
> > > > > > > We hope to have clarified your remaining concerns about the auxiliary loss, and are open to any further discussion if necessary.

---

### Official Review · Reviewer_zHLR · 2025-11-04

**Soundness:** 1
**Presentation:** 2
**Contribution:** 2
**Rating:** 2
**Confidence:** 3

**Summary:**

The paper proposes a new method for potential outcomes prediction in a time-varying potential outcomes framework, namely, G-Latent. Specifically, the proposed G-Latent estimates the distribution of multi-step-ahead potential outcomes with a tailored time-varying variational auto-encoder (VAE). The model is based on (i) transformer-encoded representations, (ii) a light-weight GRU for multi-step-ahead prediction, and (iii) infinite mixtures VAE parametrization to enhance expressivity.  Importantly, the authors suggested using a roll-out in the latent space so that the time-consuming auto-regressive prediction can be omitted. Finally, the  G-Latent is evaluated on several (semi-)synthetic benchmarks.

**Strengths:**

The paper studies a relatively understudied area of the causal ML, time-varying generative modelling of potential outcomes. The proposed method is original.

**Weaknesses:**

I have several major concerns regarding the core idea of the method: Replacing the full g-computation with the roll-out in time in the latent space.
- I agree that under certain conditions, we don’t need to auto-regressively feed the generated original covariates and can rather operate in the latent space. However, I don’t see how we can circumvent the exponential (in special cases polynomial) complexity of g-compution wrt. to the prediction horizon $\tau$ (i.e., it can be seen as an instance of the nested MC-estimator [1]). Therefore, in my opinion, the latent rollout (Algorithm 1) does not achieve the full **consistency** of the method: I would expect some sort of marginalization in the latent space $z_t$ over time steps $t, \dots, t + \tau -1$.  This would then have the same sampling speed wrt. $\tau$ as the original G-Net (as we would require $M^\tau$ samples to sample $M$ instances of $Y_{t+\tau}[a_{t}:a_{t+\tau-1}] for consistency [1]).
- I carefully checked the theoretical results Sec 5.3 (regarding the issue of consistency): Even under the assumed factorization from Eq. 8, I don’t think we can simply skip the marginalization over time-steps $t, \dots, t + \tau -1$. Equivalently, it is not clear to me how “The inner integrals over $l_{t+t’}$ evaluate to 1 for $t′ = 1, \dots, \tau − 1$” (line 1016). This would be true for the conditional distributions of outcomes but not for the distributions of potential outcomes (otherwise, our estimator yields runtime confounding [2]).

If the authors can clarify this crucial detail of their method (and all the other questions), I am open to raising my score. I understand that many existing methods in the related work (e.g., CT or G-Net) do not properly discuss the consistency of the multi-step-ahead prediction. However, the issue of consistency is crucial for the reliable application of the method.

Also, I found some small mistakes:
- Lines 138-139. “epistemic … [? and aleatoric ?]“, “bayesian” (= wrong capitalization).
- Lines 196-197. “(Variants such as βVAE scale the KL term by a factor β.)” This is not a proper sentence.
- Line 258. “ In this work, we extend DistVAE, introduced”: Reference to DistVAE is missing.
- I found the notation (upper and lower indices) inconsistent in App. E. 2.
- I think Assumption A.3 is not correct (see the correct version here [3]).

References:
- [1] Rainforth, Tom, et al. "On nesting Monte Carlo estimators." International Conference on Machine Learning. PMLR, 2018.
- [2] Coston, Amanda, Edward Kennedy, and Alexandra Chouldechova. "Counterfactual predictions under runtime confounding." Advances in neural information processing systems 33 (2020): 4150-4162.
- [3] Frauen, Dennis, Konstantin Hess, and Stefan Feuerriegel. "Model-agnostic meta-learners for estimating heterogeneous treatment effects over time." arXiv preprint arXiv:2407.05287 (2024).

**Questions:**

- Are you sure a method from [1] cannot be adapted to the current setting? The experiments in [1] also include discrete-time benchmarks.
- Why are negative log-likelihoods for $y$ and $x$ different in the reconstruction loss (Eq. 6)?
- I wonder how the proposed method compares to the full auto-regressive version (where we reconstruct all the time-varying covariates). I encourage the authors to provide such an ablation.

References:
- [1] Mu, Wenhao, et al. "Counterfactual probabilistic diffusion with expert models." arXiv preprint arXiv:2508.13355 (2025).

---

> ### Author Response · Authors · 2025-11-18
>
> We thank the reviewer for their time in carefully reading and reviewing our work, and for recognizing its originality.
>
> Next, we address the weaknesses and questions.
>
> **W1**. We do not use a nested Monte Carlo estimator, so there is no exponential complexity in the prediction horizon. We perform ancestral sampling of full latent paths: we draw \(M\) Monte Carlo latent trajectories (one chain per path across $t,\ldots,t+\tau$). This yields linear cost $O(\tau M)$, as is also the case for G-Net / transformer-based rollouts.
>
> Regarding marginalization: the concern suggests we should integrate over the latents across steps. This is exactly what our estimator does—stochastic marginalization over the latent path by sampling full trajectories.
>
> By Corollary 5.2, selective decoding does not change the target: if we decode only at $t{+}\tau$ (the ``last'' horizon in Alg. 1), earlier steps are correctly integrated out and we still sample from the marginal $p^{\bar{\mathbf{a}}}(\mathbf{y}_{t+\tau}\mid \bar{\mathbf{h}}_t)$ without decoding intermediate $\mathbf{l}$'s or running inner loops. This has the same limit law as decoding at every step, at lower computational cost.
>
> Finally, as with data-space rollouts, complexity is linear in $\tau$; the latent rollout further reduces the per-step constant by avoiding the decode$\to$re-encode loop and (by Prop.~5.3) mitigates error accumulation when rolling forward. Our pseudocode makes the ``one chain per path, no inner MC loops'' structure explicit.
>
> **W2**. We do not skip marginalization over the time steps $t,\ldots,t+\tau-1$. We perform this marginalization in latent space via **stochastic marginalization**: we draw $M$ Monte Carlo latent trajectories (ancestral sampling). The “equals 1” statement applies only to the integrals over the decoded data variables $l_{t+t'}$ for $t'=1,\ldots,\tau-1$, because each conditional $p_\theta(l_{t+t'} \mid z_{t,t'},, c_{t,t'},, a_{t+t'-1})$ is a normalized density and the context $c_{t,t'}$ does not depend on decoded $L$. In contrast, we do not drop the integrals over the latents $z_{t,t'}$ ($c_{t,t'}$ does depend on them); those are integrated by sampling full latent paths. This yields an unbiased pathwise estimator with linear cost in the horizon $O(\tau M)$ and is not a nested Monte Carlo scheme.
>
> We have slightly reformulated the appendix paragraph that was creating confusion (see line 1045 in the new version).
>
> **Q1**. We do not claim that this work could not be adapted to our setting. However, it targets a different setting relying on continuous-time domain-specific expert ODE models, which are not available in our benchmarks. In our benchmarks, we *deliberately* do not assume access to any mechanistic model; we only observe high-dimensional covariates, treatments, and outcomes in discrete time. There is no obvious expert ODE we could plug into ODE-Diff without first designing a new mechanistic model for each dataset, which would effectively change the problem we are studying. ODE-Diff couples a diffusion model with classifier-style guidance and a neural/expert ODE, requiring multiple denoising steps plus ODE solves at each step. Our focus in G-Latent is on making multi-step g-computation efficient via latent rollouts in discrete time, in settings where no mechanistic simulator is available. Even if adapted, ODE-Diff would therefore have a rather different computational trade-off from our method.
> From a *practical* standpoint, the timing also played a role. The preprint [1] appeared roughly one month before our submission deadline, without available code at that moment.
>
> **Q2**. The outcomes Y are modeled with an ALD/CRPS-type decoder (distributional head) while the covariates X use Gaussian heads; thus the NLLs differ by design. ALD/CRPS-type decoders increase expressivity and improve results (in the tables, we compare against Gaussian-heads for both covariates and outcomes). However, they also increase complexity and compute, and while there are clear benefits in using them for modelling the outcomes, using them to model the covariates doesn’t pay off.
>
> **Q3**. We indeed provide this comparison: it is the Data Space (D.S.) G-VAE, present in all the tables.

---

> > ### Author Response · Authors · 2025-11-18
> >
> > **Minor comments**:
> >
> > Thank you for the detailed suggestions. We have (i) standardized capitalization and terminology—we follow standard usage and keep epistemic and aleatoric lowercase as they are common adjectives, and we corrected "Bayesian" to uppercase throughout—(ii) added the missing DistVAE citation at its first local mention, (iii) removed sentence of line 196, (iv) checked notation across the paper and appendix, (v) corrected assumption A.3 to match the standard sequential form.
> >
> > We thank the reviewer again for the thoughtful feedback, which we believe will help us substantially improve the clarity and quality of the paper. We hope that these clarifications resolve the concerns regarding consistency and complexity of the latent rollout, and help in reassessing the soundness of our approach.
> >
> > [1] Mu, Wenhao, et al. "Counterfactual probabilistic diffusion with expert models." arXiv preprint arXiv:2508.13355 (2025).

---

### Author Response · Authors · 2025-11-18
**General comment on novelty and the changes of the new uploaded version**

We thank the reviewers for their careful reading and insightful comments.

We first clarify the estimator-level novelty of our work, and then summarize several additional analyses and experiments added in the revision.

**On novelty**

We agree that our building blocks (transformers, VAEs, GRUs) are standard, and we regret that our writing may have made it seem as if the paper’s novelty is architectural only. Our main contribution is instead at the estimator level: we introduce a latent-space implementation of discrete-time g-computation for individualized potential outcome trajectories. Under a latent factorization/context sufficiency assumption, we prove that the proposed latent rollout is equivalent to the standard g-formula estimator, i.e., it targets the same interventional distribution even though it never autoregresses covariates in the observation space. We further derive a bound (Prop. 5.3) showing that this latent rollout has more favorable error propagation in total variation than conventional data-space autoregressive rollouts, providing a theoretical explanation for the improved stability we observe at longer horizons. Finally, in contrast to G-Net’s residual hold-out scheme, which samples from a global error pool, our conditional VAE head learns individualized, history- and treatment-conditioned outcome distributions at each time step, enabling richer uncertainty quantification. To our knowledge, this is the first discrete-time method that estimates individualized distributions of potential outcomes without global residual pools. Moreover, by decoupling the transformer encoder from the lightweight latent GRU rollout and decoding only outcomes for selected horizons, the estimator supports fast Monte Carlo sampling and substantially reduced inference-time compute compared to data-space g-computation. We view these architectural choices and the ALD mixture parameterization as practically important but incremental, supporting the estimator-level novelty rather than constituting standalone contributions.

To better reflect this estimator-level focus, we have **restructured the manuscript**. In the introduction, we rewrote the contributions paragraph to explicitly state that the core novelty is the latent-space g-computation estimator and its theoretical guarantees (equivalence to the g-formula and improved error propagation), with the VAE/transformer instantiation presented as one concrete realization. In the main text, Section 3–5 now proceed from (i) the problem formulation and intervention targets, to (ii) the generic g-computation estimator we implement, and only then to (iii) the specific neural parameterization and implementation details. Within Section 5, we separate the estimator-level description from the architectural implementation, and reorder the section to present first this description and the theoretical insights. We hope this reorganization makes it clearer that the primary contribution is an estimator with provable properties, supported by but not limited to our neural architecture.

In this new version, we have also addressed most of the concerns and comments of the reviewers. Furthermore, we have incorporated several additions to strengthen the empirical validation of our model:

**1. Calibration of predictive distributions.**

Because a central goal of our work is to estimate full **predictive distributions** of potential outcomes (rather than just point estimates), we extended the evaluation to explicitly assess **calibration** of these distributions. In the revised version we now report, in addition to the Energy Score and log-likelihood already present in the original submission:

- **Coverage of nominal prediction intervals / quantiles** over time and outcomes, and
- A **scalar calibration error** summarizing the average deviation between nominal and empirical coverage across horizons.

These metrics directly evaluate whether the uncertainty quantified by the model matches the observed frequencies in the data. Across both datasets, G-Latent (in particular with the CRPS/ALD head) exhibits substantially better calibration than the baselines, supporting our claim that the latent g-computation estimator is well-suited for individualized distributional potential outcomes. We show the scalar calibration error for the semi-synthetic dataset in table 1, and the full coverage tables in appendix J.1 (tables 13 and 14).

---

> ### Author Response · Authors · 2025-11-18
>
> **2. Semi-synthetic MIMIC-III: standard benchmark, positivity analysis, and corrected dataset.**
>
> The MIMIC-III–based semi-synthetic benchmark that we use follows the setup introduced by prior work (e.g., Causal Transformer) and has since become a **standard evaluation benchmark** for methods on time-varying treatment effects. While revisiting this setup after the submission, we discovered that the **standard treatment policy can severely violate positivity** for some regions of the covariate space (i.e., propensities very close to 0 or 1). This affects *all* methods evaluated on this benchmark that are meant to be interpreted under the usual longitudinal causal assumptions (sequential ignorability, consistency, positivity). In other words, the data-generating process departs from the standard identification conditions that underlie most causal estimators in this setting, independently of their functional form.
>
> In the revised manuscript we therefore:
>
> - Make this issue explicit and illustrate the resulting lack of overlap in the original semi-synthetic setup (appendix F.1.1).
> - Propose a **modified treatment policy** that enforces overlap (while preserving the original confounding structure and outcome-generating process) and define a **corrected semi-synthetic dataset** based on this policy (appendix F.2).
> - Re-run **all methods** on the corrected semi-synthetic datasets and report results. In the main text, we give now a summary of results for the corrected semi-synthetic dataset, while the complete results for both the corrected and uncorrected semi-synthetic datasets are in the appendix.
>
> The **relative performance ranking of the methods is mostly unchanged**, and G-Latent remains strongest on distributional metrics. At the same time, this analysis and correction improve the reliability of a widely used benchmark. We see this as an additional contribution of our work to the community: clarifying and correcting a widely used benchmark so that methods evaluated on it satisfy the usual causal identification conditions.
>
> **3. inference-time efficiency and fair runtime comparison.**
>
> One of the practical advantages we highlight for latent g-computation is reduced **inference-time cost**, especially for Monte Carlo rollouts over multiple horizons. In the revised version, we make this more concrete by adding a dedicated runtime comparison. Specifically:
>
> - We **tensorized and cached** the computation for **all** models, so that each baseline benefits from similar optimizations to G-Latent.
> - We report a new table with **wall-clock inference times** for multi-step counterfactual rollouts under matched hardware and implementation conditions.
>
> These results show that G-Latent’s latent rollout with selective decoding achieves **substantially lower per-trajectory inference time** than data-space g-computation variants and than non decoupled transformer-based models. This empirically supports our claims about the computational benefits of performing g-computation in latent space.
>
> We thank the reviewers again and hope that the clarification of the novelty of our work, together with the rebuttal comments and our revised update, will allow them to reconsider their evaluation.

---

### Meta-Review · Area_Chair_X9TN · 2026-01-06

**Summary:**

This paper proposes an architecture and estimation procedure for estimating potential outcomes in settings with time-varying treatments. Specifically, the key idea is to evolve the latent state of a conditional VAE over time using a GRU, while decoupling the Transformer from the rollout at inference time. This design aims to enable fast Monte Carlo sampling while mitigating error accumulation over long horizons.

Reviewer zHLR acknowledged the originality of the work but raised serious concerns about whether the latent-space rollout genuinely avoids the computational complexity and consistency issues inherent to g-computation. The authors argued that, under their interpretation of the sampling procedure, computational complexity is reduced and that no inconsistency arises once the roles of the marginalized variables are properly clarified. However, these responses do not directly or convincingly address the reviewer’s concern regarding potential lack of consistency, nor do they provide clear theoretical justification to rule it out. From a reviewer’s perspective, it would be difficult to conclude that all concerns have been fully resolved based on the rebuttal alone.

Reviewer oFrA expressed strong skepticism regarding the technical novelty of the method and pointed out perceived design unnaturalness and over-strong claims, particularly concerning superiority over point-estimation methods without sufficient additional validation. The authors responded by emphasizing that the novelty lies at the level of the estimator rather than the architecture and by explaining the claimed advantages. Although extensive discussion took place during the rebuttal phase and the reviewer slightly improved their evaluation, they continued to view parts of the design as heuristic and the empirical validation as insufficient, maintaining an overall skeptical stance. While the discussion could not proceed further, the authors’ rebuttal did address some of the reviewer’s concerns, and the criticism may appear somewhat overly strict. Nevertheless, the fact that extensive rebuttal was required to clarify the contribution, together with the need for substantial revisions, suggests that the paper would benefit from a major revision and a new round of review.

Reviewer pxge acknowledged the practical advantages of the proposed method but raised concerns about the limited novelty and the fairness of hyperparameter tuning across methods. The authors responded by stressing the primacy of the estimation approach over architectural choices and by clarifying the experimental setup to argue for fairness. While the reviewer accepted some of these clarifications, they remained concerned that latent g-computation could still introduce bias. The authors countered this by discussing computational costs and empirical advantages, but the reviewer’s core concern regarding potential bias does not appear to have been adequately addressed.

Reviewer 35oK pointed out insufficiently explicit presentation of theoretical assumptions, a lack of intuitive explanation for why error accumulation is reduced, and shortcomings in the experimental comparisons. The authors provided detailed explanations and clarifications on each of these points, and the reviewer appeared to accept these responses to a reasonable extent.

In summary, the paper initially failed to clearly articulate its core contribution, methodological advantages, and design rationale, requiring extensive discussion to partially convince the reviewers. This is fundamentally a presentation issue, and it is unlikely to be fully remedied through incremental rebuttal-stage improvements alone. A substantial revision followed by a new round of review would be more appropriate. Moreover, some technical responses were occasionally imprecise, and the authors did not fully resolve several reviewers’ technical concerns.

**Reviewer Concerns:**

See above.

**Reviewer Scores:**

See above.

---

### Decision · Program_Chairs · 2026-01-26

Reject